# Alternative splicing of latrophilin-3 controls synapse formation

Shuai Wang[1,2✉], Chelsea DeLeon[3], Wenfei Sun[1,4,5], Stephen R. Quake[4,5,6], Bryan L. Roth[3] & Thomas C. Südhof[1,2✉]

The assembly and specification of synapses in the brain is incompletely understood[1–3]. Latrophilin-3 (encoded by *Adgrl3*, also known as *Lphn3*)—a postsynaptic adhesion G-protein-coupled receptor—mediates synapse formation in the hippocampus[4] but the mechanisms involved remain unclear. Here we show in mice that LPHN3 organizes synapses through a convergent dual-pathway mechanism: activation of $G\alpha_s$ signalling and recruitment of phase-separated postsynaptic protein scaffolds. We found that cell-type-specific alternative splicing of *Lphn3* controls the LPHN3 G-protein-coupling mode, resulting in LPHN3 variants that predominantly signal through $G\alpha_s$ or $G\alpha_{12/13}$. CRISPR-mediated manipulation of *Lphn3* alternative splicing that shifts LPHN3 from a $G\alpha_s$- to a $G\alpha_{12/13}$-coupled mode impaired synaptic connectivity as severely as the overall deletion of *Lphn3*, suggesting that $G\alpha_s$ signalling by LPHN3 splice variants mediates synapse formation. Notably, $G\alpha_s$-coupled, but not $G\alpha_{12/13}$-coupled, splice variants of LPHN3 also recruit phase-transitioned postsynaptic protein scaffold condensates, such that these condensates are clustered by binding of presynaptic teneurin and FLRT ligands to LPHN3. Moreover, neuronal activity promotes alternative splicing of the synaptogenic $G\alpha_s$-coupled variant of LPHN3. Together, these data suggest that activity-dependent alternative splicing of a key synaptic adhesion molecule controls synapse formation by parallel activation of two convergent pathways: $G\alpha_s$ signalling and clustered phase separation of postsynaptic protein scaffolds.

Synapse formation is central to the assembly of neural circuits in brain. Synapse formation is controlled, at least in part, by trans-synaptic complexes between adhesion molecules that organize pre- and post-synaptic specializations[1–3]. Multiple adhesion molecules are known to localize pre- or postsynaptically, but no coherent concept exists on how synaptic adhesion molecules assemble synapses. Among synaptic adhesion molecules, LPHN3 has a prominent role in establishing Schaffer-collateral synapses formed by CA3-region axons on CA1-region pyramidal neurons in the hippocampus[4]. LPHN3 is a postsynaptic adhesion G-protein-coupled receptor (GPCR) that binds to presynaptic teneurin and FLRT adhesion molecules[5–7]. The function of LPHN3 in synapse formation is known to require both its extracellular FLRT- and teneurin-binding sequences and its intracellular regions, including its Gα protein-binding sequences[4,8], but it is unclear how LPHN3 functions in synapse formation as a key synaptic adhesion molecule that is also a GPCR. In cell-signalling assays, multiple Gα proteins were reported to couple to LPHN3[9–12], as confirmed by cryo-electron microscopy (cryo-EM) structures[13,14]. However, it is unclear which Gα protein physiologically mediates LPHN3-dependent synapse assembly, whether Gα protein signalling on its own constitutes the core mechanism of LPHN3-induced synapse formation and how presynaptic ligand binding to postsynaptic LPHN3 induces synapse formation. Moreover,

synaptic scaffold proteins have well-established functions in synapse organization through the formation of phase-separated condensates[15], but their relation to trans-synaptic adhesion complexes remains poorly understood.

Here we show that *Lphn3* transcripts undergo extensive alternative splicing. The resulting protein variants couple to different Gα proteins. Of these, the variant coupling to $G\alpha_s$ that induces cAMP production is the predominant splice variant in the brain that is essential for synapse formation in the hippocampus. Increasing neuronal activity switches alternative splicing of *Lphn3* towards the synapse-forming $G\alpha_s$-coupled variant. Notably, only the $G\alpha_s$-coupled LPHN3 splice variant recruits postsynaptic scaffold proteins to trans-synaptic junctions. This recruitment requires the integration of the LPHN3 cytoplasmic tail onto the surface of phase-transitioned postsynaptic protein scaffolds. Presynaptic teneurin and FLRT ligands synergistically promote clustering of LPHN3-containing postsynaptic scaffold protein condensates, explaining how trans-synaptic interactions could induce the assembly of postsynaptic specializations. Both the synaptic function of LPHN3 and its ability to recruit phase-transitioned postsynaptic protein scaffolds require the C-terminal LPHN3 PDZ-domain-binding motif (PBM) that interacts with SHANK scaffold proteins. Our data outline a mechanistic pathway

[1]Department of Molecular and Cellular Physiology, Stanford University, Stanford, CA, USA. [2]Howard Hughes Medical Institute, Stanford University, Stanford, CA, USA. [3]Department of Pharmacology, UNC Chapel Hill School of Medicine, Chapel Hill, NC, USA. [4]Department of Applied Physics, Stanford University, Stanford, CA, USA. [5]Department of Bioengineering, Stanford University, Stanford, CA, USA. [6]The Chan Zuckerberg Initiative, Redwood City, CA, USA. ✉e-mail: swang9@stanford.edu; tcs1@stanford.edu

of synapse formation in which Gα_s signalling, phase separation and trans-synaptic ligand binding synergize in the assembly of postsynaptic specializations. This pathway is controlled by the alternative splicing of *Lphn3*, enabling precise regulation of synapse formation by neuronal activity.

## Extensive alternative splicing of *Lphn3*

To comprehensively profile the alternative splicing pattern of *Lphn3*, we analysed full-length mRNA transcripts from the mouse retina and cortex[16]. We identified five principal sites of *Lphn3* alternative splicing (Fig. 1a and Extended Data Fig. 1a). Among these, alternatively spliced exons 6 and 9 (E6 and E9) encode extracellular sequences that are known to regulate binding to the presynaptic ligand teneurin[7,17], whereas E15 encodes a 13-amino-acid sequence within the extracellular GAIN domain. On the cytoplasmic side, E24 encodes a sequence in the third intracellular loop of the seven-transmembrane-region GPCR region of LPHN3. The most extensive alternative splicing of *Lphn3* is observed in the C-terminal sequence, which is encoded by E28–32. Following the constitutive E27, all *Lphn3* transcripts contain either E31 or E32. E28–30 are variably included in E31-containing but not in E32-containing transcripts. As a result, *Lphn3* transcripts encode three distinct C-terminal sequences (Fig. 1b and Extended Data Fig. 1b). Transcriptome analyses of *Adgrl1* (also known as *Lphn1*) and *Adgrl2* (also known as *Lphn2*) revealed that all latrophilins are alternatively spliced at sites corresponding to E9 and E31/E32 of *Lphn3*, and that *Lphn2* but not *Lphn1* is also alternatively spliced at sites corresponding to E15 and E24 of *Lphn3* (Extended Data Fig. 2a–d).

To assess the cell type specificity and relative abundance of various *Lphn3* transcripts, we analysed RNA-sequencing (RNA-seq) data obtained using ribosome-bound mRNAs that were isolated from different types of neurons[18]. We found that mRNAs containing E31 were more abundant (60–80% total) than mRNAs containing E32 (20–40%), with fewer mRNAs containing E30b (20–25%) (Fig. 1c and Extended Data Fig. 1c). Alternative splicing was cell type specific, such that inhibitory neurons had a higher prevalence of mRNAs containing E31 and E30b compared with excitatory neurons (Fig. 1c and Extended Data Fig. 1c–e). Some sites of alternative splicing exhibited developmental regulation (Extended Data Fig. 3).

## *Lphn3* splicing controls Gα_s coupling

As the alternatively spliced *Lphn3* sequences at the cytoplasmic sides are proximal to its G-protein interaction site[13,14], we systematically analysed G-protein coupling of principal *Lphn3* splice variants using TRUPATH[19] (Fig. 1d (left)). The alternative splicing pattern of E24, E30b, E31 and E32 produces six principal variants. Substantial differences emerged between *Lphn3* splice variants in their G-protein-coupling preferences (Fig. 1d (middle)). The most abundant *Lphn3* splice variant in the hippocampus (E24⁺E30b⁻E31⁺E32⁻; Extended Data Figs. 1 and 3) preferentially couples to Gα_s and less strongly to Gα_{12/13}. If E31 is replaced by E32 (E24⁺E30b⁻E31⁻E32⁺), LPHN3 predominantly couples to Gα_{12/13}. Inclusion of E30b or exclusion of E24 also shifts LPHN3 Gα coupling from Gα_s to Gα_{12/13} (Fig. 1d and Extended Data Fig. 4). The role of E24 in the third cytoplasmic loop of LPHN3 is consistent with recent studies revealing the importance of this sequence in controlling G-protein coupling[20]. However, the effect of the C-terminal alternative splicing of LPHN3 on Gα-protein coupling is surprising given that the sequences involved start 81 residues downstream of the last transmembrane region. These C-terminal sequences are not resolved in cryo-EM structures of LPHN3 complexed to G proteins[13,14].

As an orthogonal approach to confirm the TRUPATH data, we measured the effect of *Lphn3* splice variants on cAMP levels in HEK293 cells.

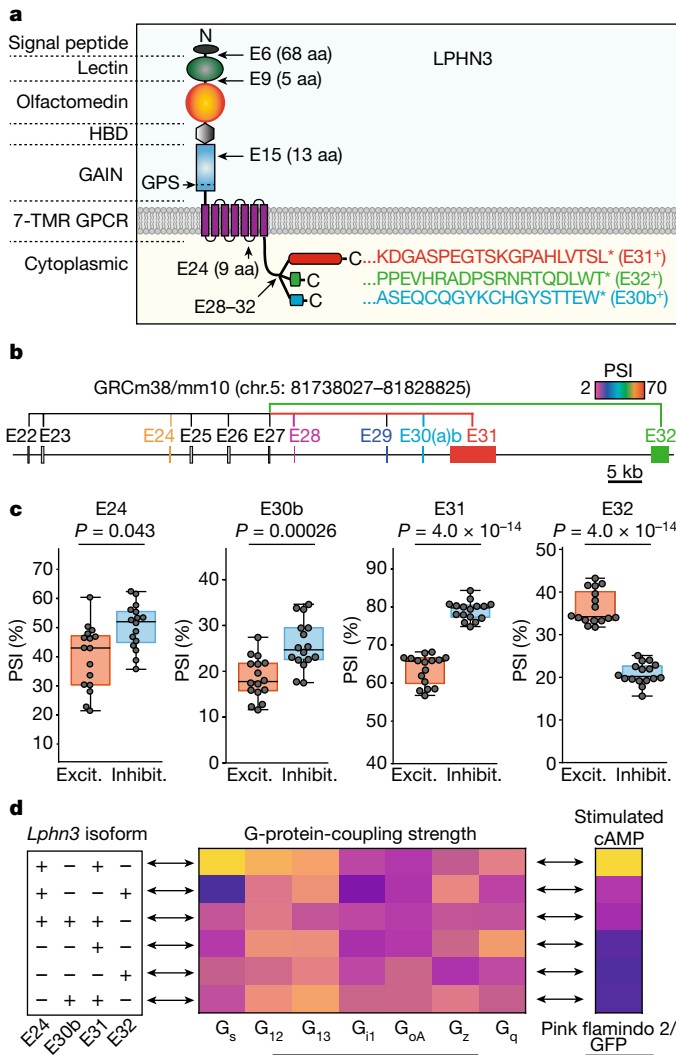

**Fig. 1 | Differentially expressed *Lphn3* splice variants couple to different G proteins. a**, Schematic of *Lphn3* alternative splicing. The asterisks indicate the stop codon. aa, amino acids; HBD, hormone-binding domain; TMR, transmembrane region. **b**, Genomic organization of the 3′ alternatively spliced exons of the *Lphn3* gene. Alternative exons are colour coded on the basis of the percentage spliced in (PSI) in the hippocampus (Extended Data Fig. 1c), with constitutive exons coloured grey. **c**, Cell-type-specific splicing of *Lphn3*. Raw data from ribosome-associated transcriptome analyses[18] were analysed to calculate the PSI of each exon for excitatory (excit.) and inhibitory (inhibit.) neurons (subtype-specific data are shown in Extended Data Fig. 1c). Statistical analysis for *n* = 16 biologically independent replicates was performed using two-sided *t*-tests; *P* values are shown in the figure. For the box plots, the whiskers extend to the minimum and maximum values, the centre line shows the median value and the box limits show the interquartile range (25th to 75th percentile). **d**, G-protein coupling and stimulated cAMP levels associated with *Lphn3* splice variants. Left, the splice variants. Middle, the G-protein-coupling signal (bioluminescence resonance energy transfer (BRET) signal) from TRUPATH assays. Right, cAMP stimulated by *Lphn3* splice variant expression in HEK293 cells. Detailed data are shown in Extended Data Figs. 4 and 5a.

Co-expression of the cAMP reporter pink flamindo 2[21] with *Lphn3* splice variants revealed that the Gα_s-coupled E31 variant of *Lphn3*, but not the E32 variant, induced high cAMP levels that were quenched by co-expressed PDE7b, a cAMP-phosphodiesterase (Fig. 1d and Extended Data Fig. 5a,b). We conclude that alternative splicing of *Lphn3* controls

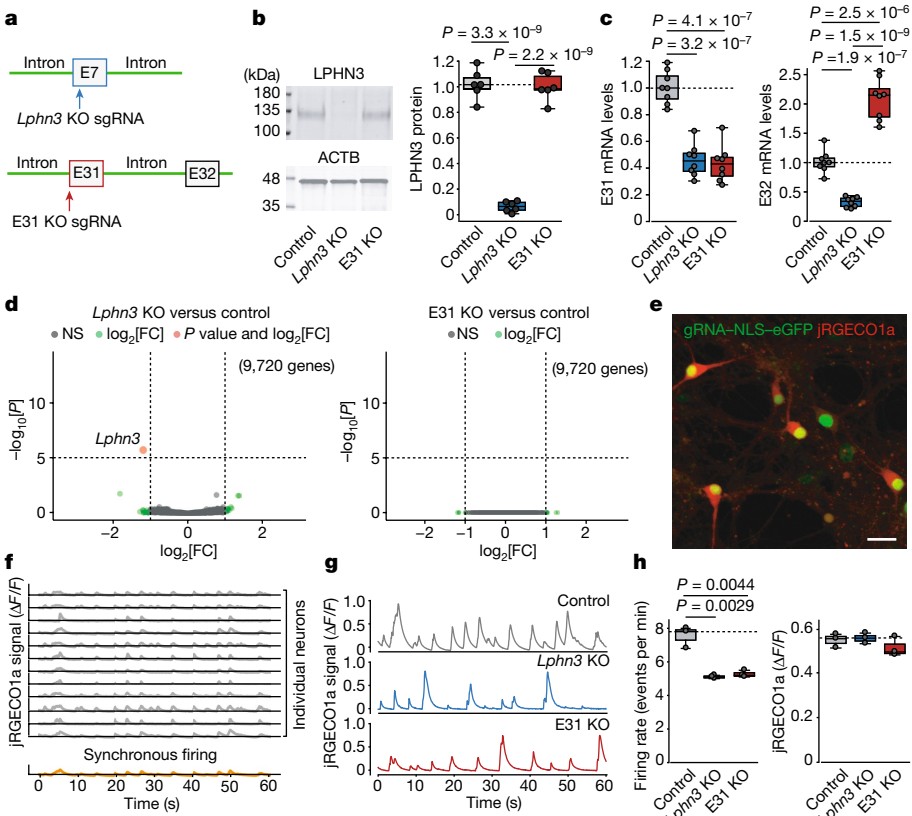

**Fig. 2 | CRISPR-mediated conversion of *Lphn3* alternative splicing from E31 to E32 impairs neuronal network activity. a**, The CRISPR strategy to produce either an acute deletion of *Lphn3* expression (*Lphn3* knockout (KO)) or a selective deletion of *Lphn3* E31 (E31 KO) of *Lphn3*. **b**, LPHN3 immunoblots from hippocampal neurons showing that the *Lphn3*-E31-specific KO does not change LPHN3 protein levels, whereas *Lphn3* KO ablates LPHN3 expression. Statistical analysis from *n* = 6 independent cultures was performed using two-sided *t*-tests. **c**, RT–qPCR analysis demonstrating that the *Lphn3*-E31 KO and the *Lphn3* KO similarly ablate the expression of E31-containing *Lphn3* mRNAs but have the opposite effect on E32-containing *Lphn3* mRNAs. Statistical analysis from *n* = 8 independent cultures was performed using two-sided *t*-tests. **d**, RNA-seq analyses of differentially expressed genes, comparing *Lphn3* KO with the control (left) or *Lphn3*-E31-specific KO with the control (right). Statistical

analysis from *n* = 3 biologically independent cultures was performed using two-sided Wald tests (in DESeq2); *P* values are shown. **e**, Representative Ca²⁺ imaging experiment of hippocampal neurons expressing gRNA, eGFP and jRGECO1a[38]. Cells expressing the gRNA with nuclear eGFP are shown in green. Red fluorescence from the soma corresponds to the jRGECO1a peak signal. Scale bar, 20 μm. **f**, Representative illustration of the extraction of jRGECO1a signals (Ca²⁺ imaging traces) from individual neurons (grey, top) of which the average is the synchronous firing trace for one field of view (orange, bottom). **g**, Representative traces of synchronous firing in control, *Lphn3*-KO and *Lphn3*-E31-KO neurons. **h**, Quantification of the synchronous firing rate (left) and amplitude (right). Statistical analysis from *n* = 3 independent cultures was performed using two-sided *t*-tests. For **b**, **c** and **h**, statistically significant *P* values are shown (*P* < 0.05).

its Gα specificity, with the most abundantly expressed LPHN3 variants in the hippocampus preferentially coupling to Gα_s and stimulating cAMP production.

LPHN1 and LPHN2 have also been associated with different Gα proteins in previous studies[8,22,23], prompting us to additionally study their Gα-protein-coupling modes. For these latrophilins we also observed preferential coupling to Gα_s for the most abundant splice variants (Extended Data Fig. 2e). Viewed together, these data reveal that alternative splicing regulates the Gα protein preference of latrophilins, with the more abundant latrophilin variants coupling to Gα_s.

## Genetic manipulation of *Lphn3* splicing

To understand which *Lphn3* splice variants might promote synapse formation, we focused on the two most abundant alternatively spliced *Lphn3* exons in the hippocampus: E31 and E32, which regulate the coupling of LPHN3 to different Gα proteins (Extended Data Fig. 1a). We controlled the expression of these two exons from the endogenous *Lphn3* gene using an acute CRISPR–Cas9 gene manipulation approach whereby we selectively deleted the alternatively spliced E31 with a

guide RNA (gRNA), using a non-targeting gRNA as a negative control and an E7-targeting gRNA to ablate all *Lphn3* expression as a positive control (Fig. 2a).

In primary hippocampal cultures, acute CRISPR-mediated total *Lphn3* deletion rendered LPHN3 protein undetectable by immunoblotting, whereas the E31-only *Lphn3* deletion or the control gRNA had no apparent effect on LPHN3 protein levels (Fig. 2b). Quantitative PCR with reverse transcription (RT–qPCR) and RNA-seq analyses of neurons with a targeted E31-specific deletion showed around a 60% decrease in the levels of E31-containing mRNAs and an approximately 100% increase in the levels of E32-containing mRNAs (Fig. 2c and Extended Data Fig. 5c). When LPHN3 protein was deleted by targeting E7, we observed a decrease of around 60% in mRNAs containing E31 or E32, presumably due to nonsense-mediated mRNA decay of the mutant mRNAs[24]. Transcriptomic analyses detected no off-target effects by either genetic manipulation (Fig. 2d and Extended Data Fig. 5d). These results validate the efficiency and specificity of the CRISPR manipulations, with E7 targeting causing a complete loss of LPHN3 protein, whereas E31 targeting induced a selective switch from E31-containing to E32-containing mRNAs.

## Synapse connection requires Gα_s–LPHN3

We used three approaches to test whether deletion of E31—and, therefore, elimination of Gα_s coupling—affects the function of LPHN3 in synapse formation. First, we measured the network activity of cultured hippocampal neurons using $Ca^{2+}$ imaging (Fig. 2e). Neurons exhibit regular spiking owing to spontaneous activity that can be averaged to produces a 'synchronous firing' trace (Fig. 2f), which reflects the strength of the synaptic network[25,26]. Quantifications of the synchronous firing of cultured neurons showed that the global loss of LPHN3 caused a significant decrease (~40%) in the firing rate without altering the signal amplitude (Fig. 2g,h). Notably, the E31-specific deletion produced a decrease in neuronal firing rate similar to that observed for the global loss of LPHN3 proteins.

Second, we examined whether the decrease in firing rate results, at least in part, from a decrease in excitatory synapse numbers. We quantified the excitatory synapse density in cultured hippocampal neurons after deletion of either all LPHN3 protein or *Lphn3* transcripts containing E31. Both the complete loss of LPHN3 and the E31-specific deletion produced a significant decrease in synapse density (Fig. 3a,b).

Third, we tested the function of E31 in vivo. We performed monosynaptic retrograde tracing using pseudotyped rabies virus to map the connectivity of genetically manipulated starter neurons in the hippocampal CA1 region (Fig. 3c). Again, both the loss of all *Lphn3* expression and the switching from E31 to E32 caused a large decrease (~60%) in synaptic inputs to CA1 pyramidal neurons from the ipsilateral and contralateral CA3 region (Fig. 3d,f). Inputs from the entorhinal cortex were unchanged as LPHN3 mediates the formation of CA3 to CA1 Schaffer-collateral but not of entorhinal cortex-to-CA1 synapses[4] (Fig. 3e,f).

Thus, using three independent methods, these data demonstrate that the E31-containing LPHN3 isoform coupled to Gα_s is essential for LPHN3-mediated synaptic connectivity.

## Assembly of synaptic complexes by LPHN3

Given the importance of the C-terminal sequence of LPHN3 encoded by E31 in synapse formation, we examined whether E31 performs additional functions other than regulating G protein coupling. We noticed that only the E31-containing *Lphn3* transcripts encode a PBM that interacts with SHANK proteins[27,28] and that is conserved in all latrophilins (Fig. 1a and Extended Data Fig. 2a,b). We therefore sought to test biochemically whether full-length LPHN3 could form a complex with postsynaptic scaffold protein networks that are composed of GKAP, HOMER, PSD95 and SHANK, which are known to form phase-separated protein assemblies[15] (Fig. 4a).

We purified recombinant GKAP, HOMER, PSD95 and SHANK proteins, which are highly soluble individually (Extended Data Fig. 6). When mixed, these proteins formed a postsynaptic density complex (the GHPS complex) through phase separation[15] that was detected as a sedimented pellet by centrifugation (Fig. 4b) and as droplet-like structures by imaging (Fig. 4d). When we added purified recombinant full-length LPHN3 to the GHPS complex, only LPHN3 containing E31 robustly co-sedimented with the GHPS complex, whereas LPHN3 containing E32 did not (Fig. 4b,c and Extended Data Fig. 7a,b). Moreover, when we truncated the last three amino acids of the PBM from LPHN3 containing E31 (hereafter, ΔPBM), a mutation that is known to disrupt SHANK binding[27], co-sedimentation of LPHN3 was also impaired (Fig. 4b,c). Imaging showed that LPHN3 containing E31, but not E32 or E31(ΔPBM), was fully assembled on the GHPS complex droplets representing phase-separated condensates (Fig. 4d,e and Extended Data Fig. 7c,d). Notably, E31-containing LPHN3 was highly enriched at the periphery of the droplets, suggesting that detergent-solubilized LPHN3 formed a layer on top of the postsynaptic

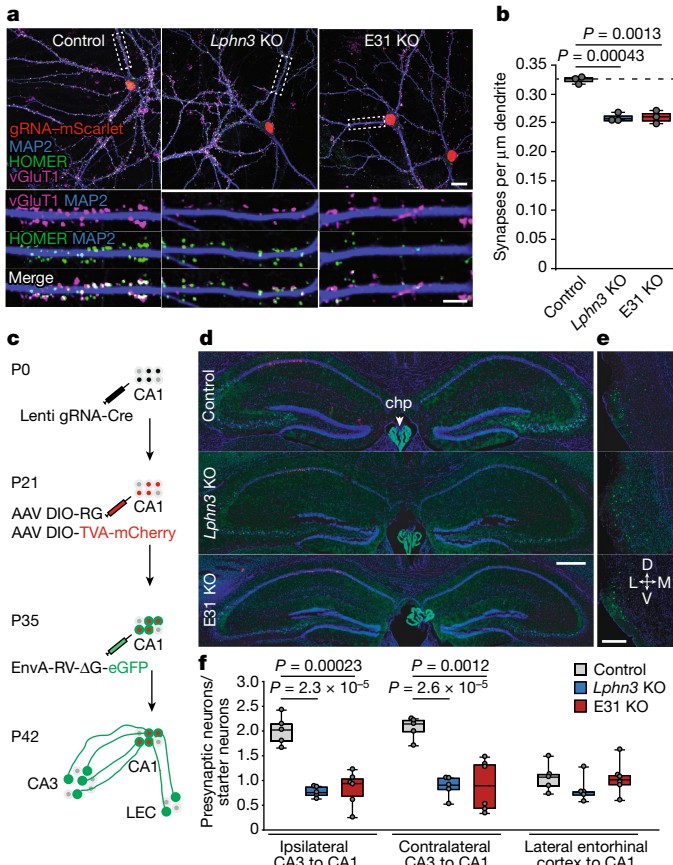

**Fig. 3 | Switching LPHN3 G-protein coupling from Gα_s to Gα_{12/13} by deleting E31 suppresses synaptic connectivity of hippocampal neurons. a,b**, Selective deletion of *Lphn3* E31 decreases the excitatory synapse density similarly to the entire deletion of *Lphn3*. **a**, Representative images of excitatory neuron staining in cultured hippocampal neurons that were stained with antibodies against vGluT1, HOMER1 and MAP2. Scale bars, 20 μm (top) and 10 μm (bottom). **b**, Summary graph of the density of puncta positive for both vGluT1 and HOMER1. Statistical analysis from *n* = 3 independent cultures (*n* = 3) was performed using two-sided *t*-tests. **c**, The experimental strategy for the retrograde tracing of monosynaptic connections using pseudotyped rabies virus[4] in CA1 neurons with acute CRISPR-mediated in vivo deletions of *Lphn3* or *Lphn3* E31. P0, postnatal day 0. **d,e**, Representative images of pseudotyped rabies tracing experiments in the hippocampal region (**d**) and lateral entorhinal cortex (LEC) (**e**). Note a weak GFP signal from the Cas9 mice was observed in the dentate gyrus granule cells and choroid plexus (chp), but not in CA3 pyramidal cells (Extended Data Fig. 5e). Scale bars 500 μm (**d**) and 200 μm (**e**). D, dorsal; L, lateral; M, medial; V, ventral. **f**, Acute CRISPR-mediated in vivo deletion of E31 of *Lphn3* impairs the number of CA3-region input synapses into the CA1-region neurons to a similar extent to the overall deletion of *Lphn3*. The box plots show the number of presynaptic neurons (ipsilateral CA3, contralateral CA3 and ipsilateral LEC) normalized to the starter neuron number. Statistical analysis from independent animals (*n* = 5 (control and *Lphn3* KO) and *n* = 6 (*Lphn3*-E31 KO)) was performed using two-sided *t*-tests. Note that *Lphn3*-E31 KO and the *Lphn3* KO do not affect entorhinal cortex input synapses, which depend on LPHN2 instead of LPHN3[4,39]. For **b** and **f**, statistically significant *P* values are shown (*P* < 0.05).

scaffold network (Fig. 4d,e and Extended Data Fig. 4c,d). As both SHANK3 and PSD95 contain PDZ domains that could interact with the PBM of LPHN3 containing E31 (Fig. 4a) and postsynaptic scaffold phase separation critically depends on SHANK3 but only weakly on PSD95 (Extended Data Fig. 7h), we conclude that the alternative splicing of LPHN3 at the C terminus determines its ability to recruit postsynaptic scaffold proteins and that such recruitment requires the

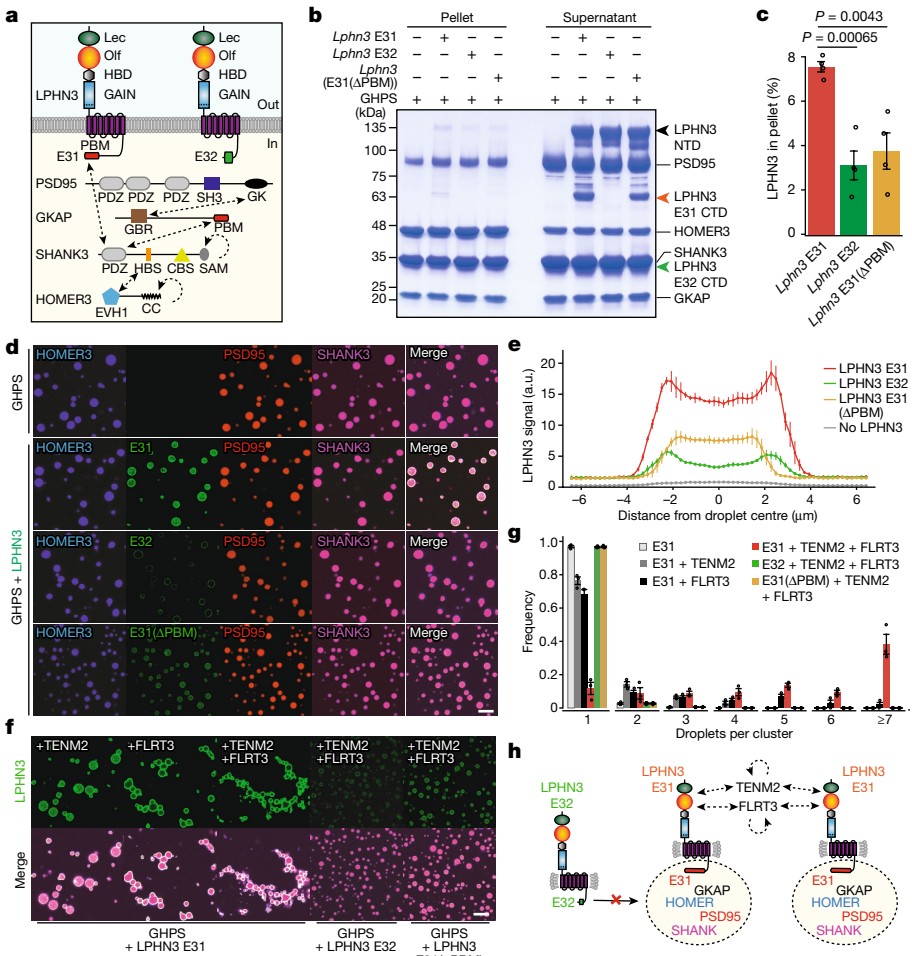

**Fig. 4 | The alternatively spliced LPHN3 E31 variant assembles phase-separated postsynaptic scaffold protein condensates. a**, Schematic of proteins. The dashed arrows show interactions. Lec, lectin-like domain; Olf, olfactomedin-like domain. GBR, GK domain-binding repeats; HBS, Homer-binding sequence; CBS, cortactin-binding sequence; SAM, sterile alpha motif; CC, coiled-coil domain. **b**, Sedimentation assay of phase-transitioned complexes. The scaffold protein mixture containing GKAP, HOMER3, PSD95 and SHANK3 (GHPS) was incubated with the indicated *Lphn3*. The pellet and supernatant were separated by centrifugation and analysed using SDS–PAGE. LPHN3 is autocleaved at the GPS site to produce the N-terminal (NTD, black arrow) and C-terminal (CTD, red and green arrow) domains[40]. The LPHN3 E32 CTD fragments migrate similarly to SHANK3 (Extended Data Fig. 6). **c**, Quantification of the LPHN3 pellet. Data are mean ± s.e.m. *n* = 4 independent experiments. Statistical analysis was performed using two-sided *t*-tests; statistically significant *P* values are shown (*P* < 0.05). **d**, Imaging of

phase-transitioned complexes. HOMER3, LPHN3 (E31, E32 and E31(ΔPBM)), PSD95 and SHANK3 were labelled with NHS-ester fluorophore 405, 488, 546 and 647, respectively, and GKAP was unlabelled. Scale bar, 5 μm. **e**, Quantification of LPHN3 across the phase-separated GHPS droplet illustrating the surface localization of *Lphn3* E31 on the droplet. Data are mean ± s.e.m. *n* = 3 independent experiments. See also Extended Data Figs. 7d and 8a. a.u., arbitrary units. **f**, Representative images of phase-transitioned postsynaptic GHPS scaffold-protein complexes containing LPHN3 E31 that were clustered by presynaptic ligands TENM2 and FLRT3. Scale bar, 5 μm. **g**, Quantification of the clustering effect of presynaptic TENM2 and FLRT3 ligands on LPHN3-E31-coated, phase-transitioned postsynaptic GHPS scaffolding protein complexes. Data are mean ± s.e.m. *n* = 3 independent experiments. **h**, Schematic of the localization of LPHN3 E31 but not of LPHN3 E32 on the surface of phase-transitioned droplets formed by postsynaptic scaffold proteins, and the clustering of droplets by the LPHN3 ligands TENM2 and FLRT3.

interaction between the PBM in E31 of LPHN3 and the PDZ domain in SHANK3 or/and PSD95.

Teneurins (TENMs)[5] and FLRTs[6] are single-transmembrane-region-containing adhesion molecules that bind to the extracellular region of LPHN3. Their binding to LPHN3 is thought to mediate the trans-synaptic interaction between axon terminals and postsynaptic spines[4]. We examined how TENM2 and FLRT3 might affect the morphology of the phase-separated postsynaptic scaffold protein complex containing LPHN3. When we added the purified extracellular region of TENM2 to the phase-separated GHPS complex, we observed partial clustering of monomeric droplets into dimers and trimers (Fig. 4f,g and Extended Data Figs. 7f,g and 8b). The addition of FLRT3 clustered the droplets into higher-order oligomers, presumably partly due to the higher affinity of FLRT3 ($K_d$ ≈ 15 nM)[6] compared

with TENM2 ($K_d$ ≈ 500 nM)[29] for LPHN3. TENM2 and FLRT3 can bind to LPHN3 simultaneously[29,30] and acted synergistically in promoting the clustering of phase-transitioned droplets. The clustering effect was not observed in E32-containing LPHN3 or E31(ΔPBM)-containing LPHN3. As TENM2 is an obligatory dimer through disulfide bonds between EGF repeat domains[31] and FLRT3 forms dimers through its leucine-rich repeat domain[32,33], we posit that the dimerization of ligands promoted the intermolecular interaction of LPHN3 in adjacent droplets, resulting in the formation of clustered LPHN3-coated postsynaptic scaffold protein condensates (Fig. 4h). The formation of reconstituted LPHN3-bound scaffold protein condensates is robust at various scaffold protein concentrations (Fig. 4 and Extended Data Figs. 7 and 8a–c) and stoichiometries (Extended Data Fig. 8d) and at physiological salt concentrations (Extended Data Fig. 8e).

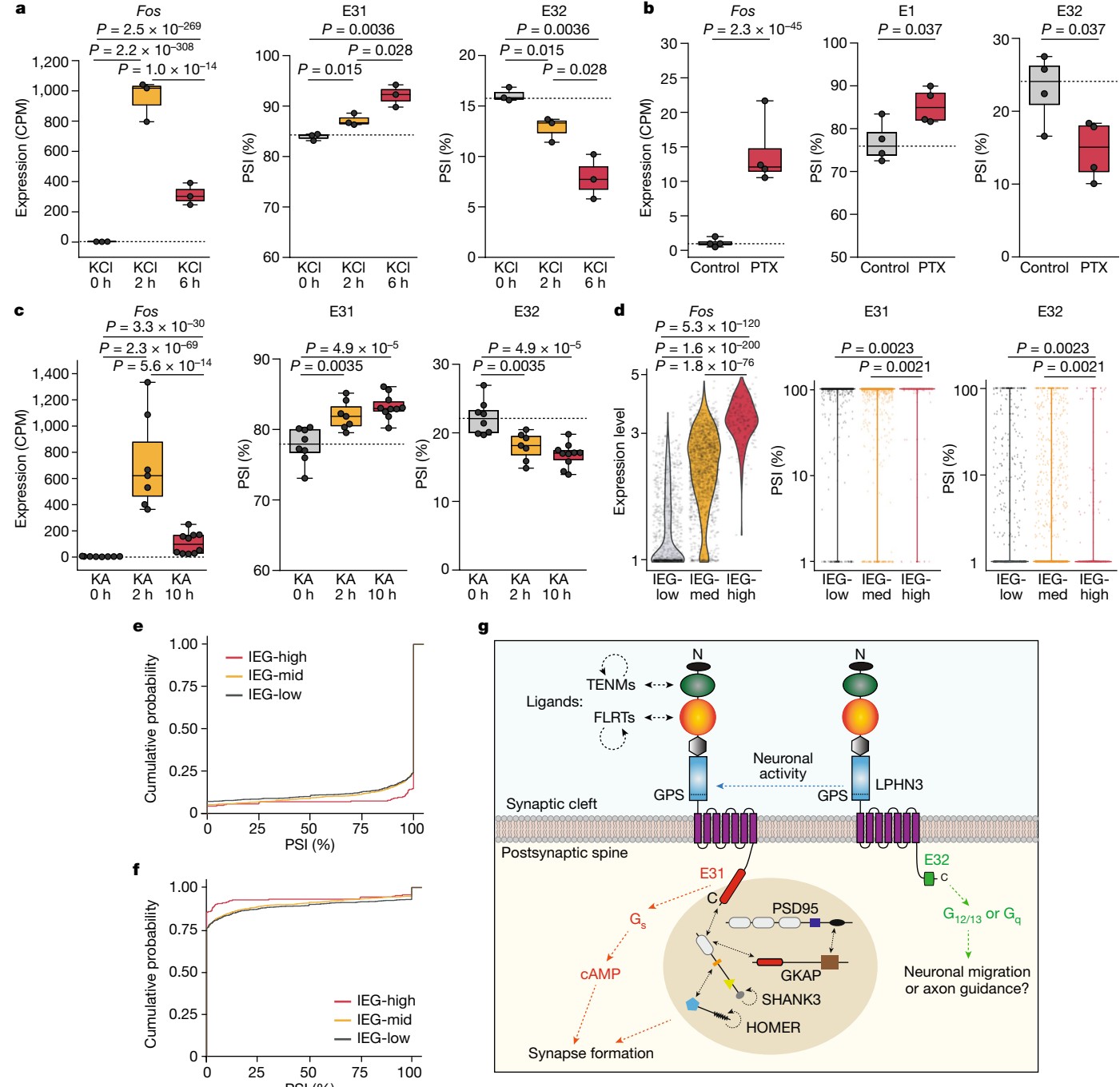

**Fig. 5 | Neuronal activity promotes E31 inclusion and E32 exclusion in *Lphn3* by alternative splicing, leading to increased expression of the synaptogenic LPHN3 E31 variant. a**–**c**, Elevated neuronal activity (on the basis of the expression of the marker *Fos*) was induced by KCl depolarization in cortical cultures[34] (**a**); GABA receptor was blocked using picrotoxin (PTX) in hippocampal cultures[35] (**b**); and hippocampal tissue was analysed after in vivo kainate (KA) injections[34] (**c**). The PSI for E31 and E32 of *Lphn3* is shown for each treatment. Two-sided *t*-tests were used to calculate the statistical significance for PSI values. Two-sided Wald tests (in DEseq2) were used to calculate the statistical significance of total gene expression. *n* = 3 and 4 biologically independent cortical cultures and hippocampal cultures, respectively; *n* = 8, 7 and 10

biologically independent hippocampal tissues after 0 h, 2 h and 10 h kainate treatment, respectively. CPM, counts per million. **d**–**f**, E31 and E32 splicing in natively activated neurons from the mPFC region[37]. Single neurons are classified as IEG-low (*n* = 903 cells), IEG-medium (*n* = 1,526 cells) and IEG-high (*n* = 232 cells) on the basis of the expression level of activity marker genes (Methods and Extended Data Fig. 10d). The PSI of E31 (**d**, middle) or E32 (**d**, right) for individual neurons, and the cumulative probability of all neurons for E31 (**e**) or E32 (**f**) are shown. Statistical analysis was performed using two-sided Wilcoxon rank-sum tests. **g**, Model of the mechanism of action of LPHN3 in synapse formation and the regulation of LPHN3 function by alternative splicing of E31. For **a**–**f**, statistically significant *P* values are shown (*P* < 0.05).

Notably, fluorescence recovery after photobleaching (FRAP) experiments show that LPHN3 exhibits faster recovery kinetics than most scaffold proteins, suggesting that LPHN3 E31 forms a fluidic shell on the surface of the postsynaptic scaffold protein condensates.

The coating by LPHN3 E31 of the condensates, regardless of the TENM2/FLRT3-induced clustering, does not substantially perturb the FRAP recovery kinetics of scaffold proteins (Extended Data Fig. 8f–l). Together, these data suggest E31-containing LPHN3 recruits

postsynaptic scaffold protein complexes to the postsynaptic site. When encountering TENM2/FLRT3 from the axon terminal, the LPHN3-coated postsynaptic scaffold protein complexes assemble into higher-order clusters.

## Synapse formation requires PBM of E31

Owing to the crucial function of the PBM of E31 for LPHN3-dependent postsynaptic scaffold protein assembly, we examined whether the PBM is also important for synapse formation in cultured neurons. Acute CRISPR manipulations in cultured hippocampal neurons with a gRNA targeting the PBM of E31 in LPHN3 (Extended Data Fig. 9a) deleted the PBM in around 70% of *Lphn3* mRNAs (Extended Data Fig. 9b,c). The deletion of the PBM had no effect on the total LPHN3 protein, *Lphn3* E31 mRNA levels or *Lphn3* alternative splicing (Extended Data Fig. 9d,e) but caused a significant decrease in excitatory synapse density (Extended Data Fig. 9f,g). Thus, the PBM of E31 in LPHN3 is important for excitatory synapse formation.

## Activity promotes E31 splicing

Given the cell-type-specific expression of *Lphn3* alternative splice variants and their distinct functions in synapse formation, we examined whether alternative splicing of E31 and E32 is regulated by neuronal activity. To this end, we analysed three recent RNA-seq datasets in which neurons in culture or in vivo were examined after chemical stimulation[34,35]. All treatments increased *Fos* expression, a marker gene for activated neurons (Fig. 5a–c (left)). We observed a significant shift in *Lphn3* alternative splicing from E32 to E31 in all datasets (Fig. 5a–c). By contrast, we found no activity-dependent changes in the inclusion of *Nrxn1* SS4, which is an alternatively spliced exon irresponsive to neuronal activity as expected[36].

To further assess the activity-dependent alternative splicing of *Lphn3* in a more native state, we analysed a high-resolution single-cell RNA-seq (scRNA-seq) dataset from the medial prefrontal cortex (mPFC)[37]. We classified neurons into different activity states on the basis of the expression level of immediate early genes (Fig. 5d (left)) into IEG-low, IEG-medium and IEG-high groups. The IEG-high group exhibited significantly lower levels of E32 and higher levels of E31 compared with the other groups (Fig. 5d–f). We also detected an elevated total LPHN3 expression with certain stimulations (Extended Data Fig. 10e) that was less robust than the switch of alternative splicing from E32-containing non-synaptogenic to E31-containing synaptogenic LPHN3 variants.

## Summary

Here we show that *Lphn3* transcripts are subject to extensive alternative splicing that controls its G-protein coupling specificity and its ability to recruit postsynaptic protein scaffolds. We demonstrate that the Gα$_s$-coupled LPHN3 splice variant that mediates cAMP signalling is required for synapse formation in vivo, and that this synaptogenic splice variant selectively recruits postsynaptic scaffolds by enabling incorporation of LPHN3 onto the surface of phase-transitioned postsynaptic density protein complexes. Furthermore, we found that the phase-transitioned postsynaptic scaffolds recruited by LPHN3 are clustered into larger assemblies by presynaptic teneurin and FLRT ligands of LPHN3. These findings outline a synapse formation mechanism orchestrated by LPHN3 that is mediated by two parallel pathways, localized Gα$_s$/cAMP signalling and recruitment of phase-transitioned postsynaptic protein scaffolds. Finally, we demonstrate that increased neuronal activity enhances the abundance of the synaptogenic splice variant of LPHN3, providing an example for how neurons leverage alternative splicing to precisely control synapse formation.

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

## Methods

### Mouse handling

C57BL/6 (JAX, 000664) mice were used for tissue RT–PCR experiments, and CAG-Cas9 mice (JAX, 024858) were used for all of the other experiments. Mice were weaned at P21 and housed in groups of maximum 5 under a 12 h–12 h light–dark cycle with food and water ad libitum, in the Stanford Veterinary Service Center. All of the procedures conformed to National Institutes of Health Guidelines for the Care and Use of Laboratory Mice and were approved by the Stanford University Administrative Panel on Laboratory Animal Care.

### Plasmids

Plasmids for the TRUPATH assay were from the Roth lab (Addgene, 1000000163). Pink flamindo 2, GFP and PDE7b constructs were obtained from previous study[8]. Mouse *Lphn3* of specified splicing variants (all have the following splicing configuration: E6⁻E9⁺E15⁺E28⁻E29⁻) were cloned into the pCMV vector using In-Fusion HD assembly. For manipulating *Lphn3* KO, *Lphn3* E31 KO or *Lphn3* E31(ΔPBM), gRNAs were cloned into lentiCRISPR v2 (Addgene, 52961), followed by human *SYN1*-promoter driven eGFP (for calcium imaging and RNA-seq) or mScarlet-I (for synapse puncta staining) or Cre recombinase (for monosynaptic rabies tracing), using In-Fusion HD assembly. jRGECO 1a[38] was cloned into the FSW lentiviral vector. *Gkap*, *Homer3*, *Psd95*, *Shank3* coding regions[15] containing N-terminal 6×His and 3C protease cleavage site were cloned into the pCT10 vector. N-terminal Flag-tagged LPHN3 E31 and LPHN3 E32 (with full splicing combination: E31: E6⁻E9⁺E15⁺E24⁺E28⁻E29⁻E30⁻E31⁺; E32: E6⁻E9⁺E15⁺E24⁺E28⁻E29⁻E30⁻E32⁺) were cloned into the lenti_CMV-TetO2 vector. Sequences of all constructs were confirmed by Sanger sequencing at Elim Biopharm or by long-read sequencing at Primordium.

### Genetic CRISPR manipulations

Four gRNAs were designed in this study. The control gRNA (5′-CCGGAA GAGCGAGCTCTACT-3′) was designed to have no target in the mouse genome. The *Lphn3* KO gRNA (5′-GCCCGGACAACGGAGCTCAA-3′) targets the constitutive E7 to induce a frameshift. The *Lphn3* E31 gRNA (5′-TCTTGTAATCTTTTTCAGAG-3′) targets the splicing acceptor site immediately upstream of E31, to disrupt the inclusion of E31. The *Lphn3* E31(ΔPBM) gRNA (5′-AGACTAGTGACCAAGTGCGC-3′) targets the PBM of *Lphn3*. Potential off-target effects were assessed using Cas-OFFinder[41] to ensure specificity.

### Generation of the reference exon list

The exon coordinates of *Lphn3* were extracted from GFF annotation of mouse genome GRCm38/mm10. Non-overlapping exons were named numerically in ascending order from 5′ to 3′ of the transcript. For exons with overlapping regions (mostly due to alternative splicing donor/acceptor site), they were named with the same number but with different letters. For exons at the 5′ and 3′ untranslated region, only the longest annotated exons were used, as the current study focuses on the coding region. This generated the draft of exon list. As the annotated exon list may contain exons that never translate to proteins (mostly due to incomplete splicing/incorrect annotation), the draft exon list was used to map the reads of *Lphn3* from Ribotaq sequencing dataset[17], which is highly enriched for translating mRNAs, during which only exons detected in this dataset were preserved to produce the final reference exon list for this study.

### Analysis of high-throughput sequencing data

This study analysed six datasets from published studies and two datasets generated from this study:

(1) Reads from PacBio long-read mRNA sequencing data[16] were aligned to reference genome (GRCm38/mm10) using gmaps. Reads belonging to *Lphn3* from above five tissue samples (four developmental stages for the retina and P35 for the cortex) were combined to increase the read depth, for analysing the abundance of full-length transcripts.

(2) Cell-type-specific RiboTag sequencing data[18] reads were aligned to reference genome (GRCm38/mm10) using STAR. Reads belonging to *Lphn3* were used for calculating the PSI of exons.

(3) Neuronal activity regulated bulk transcriptome (KCl and kainate treated)[34] and picrotoxin treated[35] datasets were downloaded from the Gene Expression Omnibus (GEO) under accession numbers GSE175965 and GSE104802. Reads were aligned as described above using STAR. Reads belonging to *Lphn3* were used for calculating the PSI of exons.

(4) Analysis of native neuronal activity from the mPFC region used a scRNA-seq dataset[37]. Smartseq reads were mapped to a custom genome, and individual *Lphn3* exons were counted individually. Cells were unbiasedly clustered on the basis of their transcriptomes. Immediate early genes were identified by ranking each genes' correlation to *Fos* expression. An IEG score was calculated by combining the expression of *Fos*, *Ier2*, *Egr1*, *Junb* and *Dusp1*, and this score was used to categorize the activation status of each cell. Single neurons with at least one count for E31 or E32 were used for splicing analysis.

(5) Reads of *Lphn3*-KO and *Lphn3*-E31-KO studies of this work were aligned as above. HTSeq was used to count reads. Only genes with more than 350 reads were used for DEseq2 analysis.

(6) Reads of PBM deletion from the amplicon sequencing dataset were aligned as described above. Paired-read sequences near the edited site were extracted and the length of each read was calculated. Insertions/deletions that caused frameshifts or mutations within the PBM region were classified as PBM-KO events.

(7) From the scRNA-seq analysis in the primary visual cortex and anterior lateral motor cortex dataset from the Allen Institute[42], processed read densities are publicly available.

### Calculation of exon PSI

PSI is the percentage of reads containing the target exon among all reads at the target region. For alternative exons containing both 5′ and 3′ flanking exons (for example, E24 and E30b), only the reads spanning the target exon–exon junctions were used for the calculation. For alternative exons at the 3′ termini (E31 and E32), all reads containing the target exon were used for calculation as the 3′ termini of *Lphn3* ends with either E31 or E32. E31 and E32 reads were normalized to exon length before calculating the PSI.

### Sample preparation of *Lphn3* KO and *Lphn3*-E31 KO neurons for next-generation sequencing

Primary hippocampal culture neurons were infected with lentiviruses expressing gRNAs (control, *Lphn3* KO, *Lphn3*-E31-only KO) at day 3 in vitro (DIV3) and maintained until DIV14. One coverslip of culture was resuspended with 200 µl TRIzol on ice, mixed with 50 µl chloroform and incubated at room temperature for 2 min. The samples were centrifuged at 12,000*g* for 15 min at 4 °C in an Eppendorf 5417C centrifuge. The aqueous layers were added to 100 µl ice-cold isopropanol for thorough mixing before incubation at −80 °C for 1 h. The samples were thawed on ice and centrifuged at 20,817*g* for 20 min at 4 °C in the Eppendorf 5417C centrifuge. Pellets were washed with 0.5 ml ice cold 75% ethanol before being centrifuged at 20,817*g* for 10 min at 4 °C, and subsequently resuspended by 40 µl double-distilled H₂O containing 0.2 U µl⁻¹ SUPERase·In RNase inhibitor. Total RNA samples were converted to a library using Illumina Stranded mRNA kit and sequenced in the NovaSeq (paired-end 150 bp) system with 40 million paired reads at Medgenome.

### Sample preparation of ΔPBM neurons for amplicon sequencing

Primary hippocampal culture neurons were infected with lentiviruses expressing gRNAs (control or ΔPBM) at DIV3 and maintained until

DIV14. Total RNA samples were extracted as above, and converted to cDNA using the PrimeScript RT-PCR Kit. The PBM region of *Lphn3* was amplified using the primers 5′-AAACCTGGGCTCCAGAAACC-3′ and 5′-GGAAAGATTGGGGCACAGGA-3′, and converted to a library using Nextera XT adaptor/indexes, and sequenced in the MiSeq system at the Stanford Functional Genomics Facility.

## TRUPATH G-protein-coupling assay

HEK293T cells were obtained from ATCC and maintained, passaged and transfected in DMEM medium containing 10% FBS, 100 U ml$^{-1}$ penicillin and 100 μg ml$^{-1}$ streptomycin (Gibco-ThermoFisher) in a humidified atmosphere at 37 °C and 5% CO$_2$. After transfection, cells were plated in DMEM containing 1% dialysed FBS, 100 U ml$^{-1}$ penicillin and 100 μg ml$^{-1}$ streptomycin for BRET assays. Constitutive activity of LPHNs was accomplished by using the previously optimized Gα–Rluc8, β-subunit and N-terminally tagged γ-GFP2 subunit pairs described previously[19]. HEK cells were plated in a 12-well plate at a density of 0.3–0.4 × 10$^6$ cells per well with DMEM containing 10% FBS, 100 U ml$^{-1}$ penicillin and 100 μg ml$^{-1}$ streptomycin. Then, 6 h later, cells were transfected with a 1:1:1 ratio of optimized Gα:β:γ pairings at 100 ng and various amounts of receptor (25 ng, 50 ng, 100 ng, 200 ng, 300 ng) using the TransIt-2020 (Mirius Bio) reagent. To establish a baseline for the cells, pcDNA was used at 100 ng and referred to as 0 ng. The next day, cells were removed from the 12-well plate with trypsin and seeded into a 96-well white clear-bottomed plate (Greiner Bio-One) with DMEM containing 1% dialysed FBS at a cell density of 30,000–35,000 cells per well. Cells were incubated overnight to allow for attachment and growth. The next day, the medium was aspirated from the wells. A solution of assay buffer (20 mM HEPES, Hank's balanced salt solution, pH 7.4) and 5 μM of coelentrazine 400a (Nanolight Technology) was prepared and added to each well. Cells were allowed to equilibrate with the coelentrazine 400a in the dark for 10 min. Corresponding BRET data were collected using a Pherastar FSX Microplate Reading with luminescence emission filers of 395 nm (RLuc8-coelentrzine 400a) and 510 nm (GFP2) and an integration time of 1 s per well. BRET ratios were calculated as the ratio of the GFP2:RLuc8 emission. The constitutive coupling (0 ng) was used as the baseline to subtract NET BRET of the experimental conditions for each receptor. Three independent cultures with seven technical replicates in each culture were used in total.

## cAMP reporter assay

HEK293T cells were maintained in DMEM + 10% FBS at 37 °C 5% CO$_2$, and seeded onto a 24-well plate. During calcium transfection of each well, eGFP (0.23 μg), pink flamindo 2 (0.23 μg), Gα$_s$ (0.16 μg), Gβ (0.16 μg) and Gγ (0.16 μg) were used for all conditions. When indicated, additional constructs were co-transfected including PDE7b (0.23 μg) and six isoforms of *Lphn3* (E24$^+$E30b$^-$E31$^+$E32$^-$, E24$^+$E30b$^+$E31$^+$E32$^-$, E24$^+$E30b$^-$E31$^-$E32$^+$, E24$^-$E30b$^-$E31$^+$E32$^-$, E24$^-$E30b$^+$E31$^+$E32$^-$, E24$^-$E30b$^-$E31$^-$E32$^+$, 0.23 μg each). Then, 16 h after transfection, the medium was replaced with 0.5 ml DMEM + 10% FBS. Then, 36–48 h after transfection, medium of all cultures was replaced with 0.5 ml imaging buffer (20 mM Na-HEPES pH 7.4, 1× HBSS (Gibco, 14065056)) and incubated at room temperature for 30 min. When indicated, 2.5 μM forskolin and 5 μM IBMX were added to the culture for 5 min. Imaging was performed under Nikon confocal microscopy under a ×10 objective.

## Primary hippocampal neuron culture

Neonatal P0 mice pups of CAG-Cas9 mice (JAX, 024858) were dissected in ice cold HBS to obtain hippocampi, which were digested in 1% (v/v) papain suspension (Worthington) and 0.1 U μl$^{-1}$ DNase I (Worthington) for 15 min at 37 °C. Hippocampi from two pups were washed with calcium-free 1× HBS (pH 7.3) and dissociated using gentle pipetting in plating medium (MEM containing 5% FBS, 0.6% glucose, 2% Gem21 NeuroPlex Supplement, 2 mM GlutaMAX), filtered through 70 μm cell strainer, and seeded onto Corning Matrigel-coated 12 mm cover

glasses in one 24-well plate, and maintained at 37 °C under 5% CO$_2$. Then, 16 h after seeding (DIV1), 90% of the medium was replaced with maintenance medium (Neurobasal A with 2% Gem21 NeuroPlex Supplement, 2 mM GlutaMAX). At DIV3, 50% of the medium was replaced with fresh maintenance medium supplemented 4 μM Ara-C (cytosine β-D-arabinofuranoside hydrochloride), and lentivirus expressing gRNA. When indicated, lentiviruses expressing jRGECO1a were added at DIV7. At DIV7, 10 and 13, 30% of the medium was replaced with fresh maintenance medium, before analysis at DIV14.

## Virus preparation

Lentiviruses were produced in HEK293T cells using the second-generation packaging system. Per 150 cm$^2$ of cells, 186 μl 2 M CaCl$_2$ containing 5.8 μg of lentivirus shuttle vector, 2.5 μg pVSVG (Addgene, 12259) and 4.2 μg Gag-Pol-Rev-Tat (Addgene, 12260) at a total volume of 1.5 ml was added dropwise to an equal volume of 2× HBS (280 mM NaCl, 10 mM KCl, 1.5 mM Na$_2$HPO$_4$, 12 mM glucose and 50 mM HEPES, pH 7.11) under constant mixing, incubated for 15 min at room temperature and added dropwise to the cells. Then, 8–12 h after transfection, the medium was replaced with DMEM with 10% FBS. Then, 48 h after transfection, the cell medium was cleared by centrifuging in table-top centrifuge at 2,000$g$ for 3 min, and filtered through a 0.45 μm PES membrane. The viral supernatant was loaded onto a 2 ml 30% sucrose cushion in PBS and centrifuged in the Thermo Fisher Scientific SureSpin 630 rotor at 19,000 rpm for 2 h. The viral pellet was resuspended in 30 μl MEM and flash-frozen in liquid nitrogen. AAVs (CAG-DIO-RG and CAG-DIO-TCB-mCherry) in capsid 2.5 and pseudotyped rabies virus RbV-CVS-N2c-deltaG-GFP (EnvA)[43] were prepared at Janelia Farm Viral core facility.

## Monosynaptic retrograde rabies tracing

P0 neonatal mouse pups were anaesthetized on ice for 4 min and were head-fixed using ear bars and a 3D-printed mould. Then, 0.35 μl 1 × 10$^9$ IU ml$^{-1}$ lentiviruses (SYN1-gRNA-NLS-cre) was injected unilaterally to CA1 at the coordinates anteroposterior (AP) +0.95 mm, mediolateral (ML) −0.92 mm, dorsoventral (DV) −1.30 mm (zeroed at Lambda). At P21, mice were anaesthetized by avertin (250 mg per kg) and head-fixed on a stereotaxic injection rig, and 0.2 μl AAVs (CAG-DIO-RG, 3.6 × 10$^{12}$ genome copies per ml; and CAG-DIO-TCB-mCherry, 6.35 × 10$^{12}$ genome copies per ml; 1:1 volume mix) were co-injected to CA1 at coordinates AP −1.80 mm, ML −1.35 mm, DV −1.30 mm (zeroed at bregma). At P35, the same CA1 site was injected with 0.15 μl EnvA-pseudotyped rabies virus RbV-CVS-N2c-deltaG-GFP at 2 × 10$^8$ IU ml$^{-1}$. After the surgery, the incisions of P21 and P35 mice were closed by suture and 3M Vetbond tissue adhesive (1469SB). After all of the injections, the mice were allowed to recover on a heating pad before returning to their home cage. At P42, the mouse brains that had been perfused were fixed in 4% PFA (Electron Microscopy Sciences, EM grade, 15714) in PBS for 4 h at room temperature, subsequently incubated in 30% (w/v) sucrose in PBS at 4 °C overnight and cryopreserved in Tissue-Tek O.C.T. compound (Sakura) on dry ice. Frozen tissue blocks were cut into 20 μm coronal sections on a cryostat and collected on glass slides (Globe Scientific, 1358W). The sections were air dried, stained in 1 μg ml$^{-1}$ DAPI for 10 min, washed once with PBS and sealed in Fluoromount-G (Southern Biotech, 0100-01). The sections were imaged on the Olympus VS200 slide scanner at ×10. A total of 5–6 mice was used per condition for the study.

## RT–PCR analysis of *Lphn3* alternative exons in tissues

C57BL/6 mice at P4, P9, P14, P21 and P35 were euthanized and the brains were dissected to isolate the olfactory bulb, cerebellum, hippocampus, prefrontal cortex, striatum and retina. Tissues were grinded with 500 μl TRIzol on ice, mixed with 125 μl chloroform and incubated at room temperature for 2 min. The samples were centrifuged at 12,000$g$ for 15 min at 4 °C in the Eppendorf 5417C centrifuge. The aqueous layers were added to 250 μl ice-cold isopropanol for thorough mixing before

incubation on ice for 5 min. The samples were centrifuged at 20,817$g$ for 20 min at 4 °C in the Eppendorf 5417C centrifuge. Pellets were washed with 0.5 ml ice cold 75% ethanol before being centrifuged at 20,817$g$ for 10 min at 4 °C, and subsequently resuspended in 40 μl double-distilled H$_2$O containing 0.2 U μl$^{-1}$ SUPERase•In RNase inhibitor. A total of 100 ng of total RNA was used for cDNA conversion using the PrimeScript RT-PCR Kit using random 6-mers. In total, 1 μl cDNA was used for PCR targeting exon–exon junction regions using Ex Taq DNA Polymerase. The following primers were used: *Actb* (5′-TCTACAATGAGCTGCGTGT-3′, 5′-CGAAGTCTAGAGCAACATAG-3′), *Lphn3* E6 (5′-CCACAGCTACTCATCCTCAC-3′, 5′-GCTCTCGATCATGATGACGT-3′), *Lphn3* E15 (5′-GGGGACATCACCTACTCTGT-3′, 5′-TCAGGTCTCTCCAGGCATTC-3′), *Lphn3* E24 (5′-CCTGAATCAGGCTGTCTTGA-3′, 5′-AAATGGTGAAGAGATACGCC-3′), *Lphn3* E31 (5′-TCCAGGACGGTACTCCACA-3′, 5′-GGCATTGTTCAGAAGCCCCT-3′), *Lphn3* E32 (5′-TCCAGGACGGTACTCCACA-3′, 5′-TCCTGTGTCCTGTTTCGGGA-3′). The PCR program was as follows: 94 °C for 1 min; then 31 cycles of 94 °C for 30 s, 55 °C for 30 s, 72 °C for 1 min. PCR products were separated on 2% agarose gel in 1× TAE buffer and imaged using the BioRad Gel Imaging system.

## RT–qPCR analysis of *Lphn3*-KO and *Lphn3*-E31-KO neurons

Total RNA (80 ng) for each culture was used for converting to cDNA using the PrimeScript RT-PCR Kit using random 6-mers. A total of 1 μl cDNA was used for qPCR experiments with the TaqMan Fast Virus 1-Step Master Mix using PrimeTime Std qPCR designed primer-probe sets: *Actb*: 5′-GACTCATCGTACTCCTGCTTG-3′, 5′-GATTACTGCTCTGGCTCCTAG-3′, /56-FAM/CTGGCCTCA/ZEN/CTGTCCACCTTCC/3IABkFQ/; *Lphn3* E27–E31 junction: 5′-CCTTCATCACCGGAGACATAAA-3′, 5′-GTGGTAGAGTATCCATGACACTTG-3′, /56-FAM/CA GCTCAGC/Zen/ATCGCTCAACAGAGA/3IABkFQ/; *Lphn3* E27–E32 junction: 5′-CAGTCAGAGTCGTCCTTCATC-3′, 5′-GTCAGTCTCAGGTCCATA AGTC-3′, /56-FAM/AACAGCTCA/Zen/GCATCGCTCAACAGA/3IABkFQ/. The PCR program was as follows: 95 °C for 20 s; then 41 cycles of 95 °C for 3 s, 60 °C for 30 s. $C_t$ values for the *Lphn3* E31 and E32 sample were subtracted by that of *Actb* from the same sample to get $\Delta C_t$. All $\Delta C_t$ values were normalized to the control gRNA. A total of eight cultures was used.

## RT–PCR analysis of *Lphn3*-KO, *Lphn3*-E31-KO and ΔPBM in neurons

Total RNA (80 ng) for each culture was used for converting to cDNA using the PrimeScript RT-PCR Kit with random 6-mers. A total of 1 μl cDNA was used for PCR targeting exon–exon junction regions using Ex Taq DNA Polymerase. The following primers were used: *Actb*: 5′-TCTACAATGAGCTGCGTGT-3′, 5′-CGAAGTCTAGAGCAACATAG-3′; *Lphn3* E31: 5′-GTCAGAGTCGTCCTTCATCAC-3′, 5′-AGTTGTTCACCAGTTTGTTCATC-3′; *Lphn3* E32: 5′-CGGATTCGGAGAATGTGGAA-3′, 5′-CCACAGATAACGTGTGTGGT-3′. The expression level of E31 and E32 were normalized to *Actb*.

## Immunoblotting analyses

One well of neuron culture from a 24-well plate was lysed in 50 μl lysis buffer (20 mM Tris pH 7.5, 500 mM NaCl, 1% Triton X-100, 0.1% SDS 1× Roche EDTA-free protease inhibitor) at room temperature for 5 min. A total of 20 μl 5× SDS loading buffer was added and the samples were analysed using SDS–PAGE. Gels were transferred to a 0.2 μm nitrocellulose membrane in the Trans-Blot Turbo Transfer system (Bio-Rad) and blocked by western blocking buffer (5% BSA in 1× TBST) at room temperature for 30 min. Mouse anti-LPHN3 (Santa Cruz Biotech, sc-393576, 1:1,000) and mouse anti-actin (Sigma-Aldrich, A1978, 1:3,000) antibodies in western blocking buffer were added and incubated at 4 °C for overnight. The membranes were washed in western blocking buffer three times for 10 min each, and IRDye 800CW donkey anti-mouse IgG secondary antibodies (Li-cor, 926-32212, 1:20,000) in western blocking buffer were added to the membrane, which was incubated at room temperature for 1 h and washed in 1× TBST three times for 10 min each.

The samples were imaged using the Odyssey Imager (Li-Cor). Quantifications of LPHN3 level were normalized to β-actin.

## Calcium imaging

Primary culture neurons were maintained as described above, except they were infected with lentiviruses expressing SYN1-gRNA-EGFP at DIV3, and lentiviruses expressing SYN1-jRGECO1a at DIV7. At DIV14, the coverslips containing neurons were washed once with 37 °C warmed Tyrode buffer (25 mM Na-HEPES pH 7.4, 129 mM NaCl, 5 mM KCl, 2 mM CaCl$_2$, 1 mM MgCl$_2$, 15 mM glucose and transferred to 12-well glass plate (Cellvis, P12-1.5H-N) in Tyrode buffer. After 30 min of incubation in Tyrode buffer at 37 °C under 5% CO$_2$, the cultures were imaged under the Leica microscope at 37 °C under 5% CO$_2$, with 50 ms exposure, 85 ms interval for 1 min for each field of view (FOV). A total of 6–8 fields of view was recorded for each coverglass of culture. For each condition from one batch of culture, 3–5 cover glasses of cultures were imaged. Three batches of culture were used in total.

## Immunohistochemistry and synapse puncta imaging

Primary hippocampal neurons were washed in Tyrode buffer (25 mM Na-HEPES pH 7.4, 129 mM NaCl, 5 mM KCl, 2 mM CaCl$_2$, 1 mM MgCl$_2$, 15 mM glucose) and fixed in 4% PFA and 4% sucrose in 1× DPBS at 37 °C for 15 min. Neurons were next washed three times with 1× DPBS for 5 min each, and permeabilized in 0.1% Triton X-100 in 1× DPBS for 10 min at room temperature without shaking. After blocking with 0.5% fish skin gelatin in 1× DPBS at 37 °C for 1 h, the culture was stained with chicken anti-MAP2 (Encor, CPCA-MAP2, 1:1,000), guinea pig anti-vGluT1 (Milipore, AB5905, 1:1,000), and rabbit anti-HOMER (Milipore, ABN37, 1:1,000) antibodies in blocking buffer at 4 °C overnight. The samples were washed three time with 1× DPBS for 8 min each, and incubated with secondary antibodies (anti-chicken Alexa 405, anti-guinea pig Alexa 647 and anti-rabbit Alexa 488) in blocking buffer at 37 °C for 1 h. Next, the culture coverslips were washed three times with 1× DPBS for 8 min each, once with double-distilled H$_2$O briefly, before being loaded onto glass slides (Globe Scientific, 1358W) in Fluoromount-G (Southern Biotech, 0100-01) and sealed in nail polish (Amazon, B000WQ9VNO). The samples were imaged under the Nikon confocal microscope at ×60, with a 0.35 μm step size and 4–6 $z$ stacks. For each coverglass of the culture, about 20 neurons containing well-isolated dendrites were imaged. For each condition of one batch of culture, two cover glasses of the culture were imaged. Three batches of the culture were used in total.

## Image analyses

Five types of image analyses were performed.

(1) Quantification of excitatory synapse puncta density. Maximum-intensity files were produced from $z$-stacked images. The background was subtracted and the 5–10 well-isolated secondary dendrites were cropped from each neuron in Fiji (v.2.9.0) for processing. Excitatory synapses, especially in mature mushroom spines, are localized -0.5–1 μm away from the dendrite due to the long neck of the spine[3]. Our confocal images have an interpixel unit of 0.20714 μm per pixel. Thus, in our analyses, we include vGluT1/HOMER signals within 5 pixels away from dendrite. The cropped files were converted to binary images using the same threshold for the same channel, for the same batch of experiment. For calculating excitatory synapse puncta, the overlapped region of vGluT1 and HOMER binary images were generated, and the overlapped regions containing more than two neighbouring pixels were considered to be puncta, and were searched and quantified using the scikit-image (v.0.20.0) package[44]. To calculate dendrite length, binary MAP2 channel images were skeletonized by scikit-image to a 1 pixel representation of which the length was measured using FilFinder (v 1.7.3) package[45]. For each cropped file, the puncta number divided by dendrite length produced the puncta density. All of the

imaged regions from one batch of the experiment were averaged to calculate the puncta density for one condition. Three batches of data were plotted in total.

(2) Calcium imaging. Time-lapsed videos of calcium imaging files were processed using the CaImAn package[46] to search for spiking somas and generate corresponding fluorescence intensity ($\Delta F/F$) over time. The key parameters were: decay_time=0.4, p=1, gnb=2, merge_thr=0.85, rf=60, stride_cnmf=6, K=10, gSig=[40,40], method_init='greedy_roi', ssub=1, tsub=1, min_SNR = 200, rval_thr=0.85, cnn_thr=0.99, cnn_lowest=0.1. $\Delta F/F$ traces of all detected spiking somas from one field of view were averaged to produce one synchronized firing trace. SciPy (v1.10.1)[47] algorithm "find_peaks" (height=0.15, width = (2,20), distance=20) was used to detect the spiking number and signal strength ($\Delta F/F$) for each synchronized firing trace. The synchronizing firing rate was calculated by dividing spiking number against total time for each trace. To plot the firing rate (or $\Delta F/F$) for each condition, the median of the firing rate (or $\Delta F/F$) from all traces of one batch was used. Three batches of culture were plotted in total.

(3) Rabies tracing. Coronal sections corresponding to bregma −1.55 to −2.03 mm[48] for hippocampal formation and Bregma −3.8 to −4.1 mm[48] for the LEC were processed in Fiji by background subtraction. Regions of the ipsilateral CA1, ipsilateral CA3, contralateral CA3 and ipsilateral LEC were cropped in Fiji for processing in scikit-image (v.0.20.0). The cropped regions were converted to binary images using the same threshold for the same channel, for the same batch of experiment. Binary regions containing more than 80 neighbouring pixels (red channel for CA1) and 150 neighbouring pixels (green channel for CA3 and LEC) were considered to be neuron soma, and were counted using the scikit-image[44] functions measure. label and measure.regionprops. All counts from one mouse were used to calculate the connectivity strength of ipsilateral CA3–CA1, contralateral CA3–CA1, and LEC–CA1.

(4) cAMP imaging using pink flamindo 2. After background subtraction, the 488 and 546 nm channel signals from one field of view was used to calculate pink flamindo 2/GFP. In total, 3–10 fields of view were imaged per condition per batch of culture. Three batches of cultures were used in total.

(5) Phase-transitioned droplet. After background subtraction, signals from the indicated channels were used for analysis. A 12.86 µm linear region across the diameter of the droplets was used to plot the signal from the edge to the centre of the droplets. To calculate the number of droplets per cluster, contacting droplets were counted as one cluster. The scikit-image (v.0.20.0) package[44] was used to count the size of droplets. Three independent replicates were used for each experiment.

## Protein purification

We used truncated GKAP and SHANK3 to retain essential interaction modules and obtain soluble proteins. 6×His-tagged GKAP, SHANK3, HOMER3 and PSD95 were purified as described previously[15] with slight modifications. Constructs were transformed into BL21 (DE3) pLysS, which were induced at an optical density at 600 nm of 0.6 with 0.25 mM IPTG at 16 °C for 18 h. Cells were lysed in Ni-buffer A (20 mM Tris pH 8, 500 mM NaCl, 5% glycerol, 4 mM BME, 20 mM imidazole, 1× Roche EDTA-free protease inhibitor, 100 U ml$^{-1}$ benzonase) and cleared at SS34 rotor at 14,000 rpm for 30 min at 4 °C. Proteins were loaded onto the Ni-NTA column, washed in Ni-buffer A and eluted in Ni-buffer B (20 mM Tris pH 8, 250 mM NaCl, 5% glycerol, 4 mM BME, 400 mM imidazole). His-tags were removed by 3C protease. Finally, the proteins were purified in a size-exclusion column (SD75 10/300 for GKAP, and SD200 10/300 for others) in SEC buffer (20 mM Tris pH 8, 300 mM NaCl, 2 mM DTT). Lentiviruses containing CMV-TetO$_2$-Flag-Lphn3 E31, E32 and E31(ΔPBM) were used to express proteins in FreeStyle 293-F cells at 37 °C under 8% CO$_2$. Cells were collected 60 h after 5 µg ml$^{-1}$

doxycycline induction, and lysed in lysis buffer (20 mM Na-HEPES pH 7.4, 500 mM NaCl, 1% DDM, 0.1% CHS, 30% glycerol, 1× Roche EDTA-free protease inhibitor cocktail, 100 U ml$^{-1}$ benzonase). The lysate was incubated with 2 mg ml$^{-1}$ iodoacetamide, cleared by centrifugation at SS34 rotor at 16,000 rpm for 30 min. The supernatant was loaded onto anti-Flag M1 Agarose Affinity Gel in wash buffer 1 (20 mM Na-HEPES pH 7.4, 500 mM NaCl, 0.01% LMNG, 0.001% CHS, 2 mM CaCl$_2$, 20 µM leupeptin). Bound protein was washed with wash buffer 1 and wash buffer 2 (20 mM Na-HEPES pH 7.4, 150 mM NaCl, 0.01% LMNG, 0.001% CHS, 2 mM CaCl$_2$), and eluted in elution buffer (20 mM Na-HEPES pH 7.4, 150 mM NaCl, 0.01% LMNG, 0.001% CHS, 5 mM EGTA, 0.2 mg ml$^{-1}$ Flag peptide), and further purified on the SD200 10/300 column in SECL buffer (20 mM Na-HEPES pH 7.4, 150 mM NaCl, 0.002% LMNG, 0.0002% CHS). 6×His-tagged TENM2 and FLRT3 were cloned into the pCMV vector and expressed in Expi293F cells. Then, 4 days after transfection, the medium was collected and loaded onto the Ni-NTA column, washed in Ni-buffer C (20 mM HEPES pH 7.4, 150 mM NaCl, 20 mM imidazole pH 7.6) and eluted in Ni-buffer D (20 mM HEPES pH 7.4, 150 mM NaCl, 250 mM imidazole pH 7.6). His-tags were removed by 3C protease, and the proteins were purified in a size-exclusion column (SD200 10/300) in SEC buffer (20 mM HEPES pH 7.4, 150 mM NaCl).

## Fluorescence labelling of proteins

For HOMER, PSD95 and SHANK, proteins were buffer-exchanged to labelling buffer 1 (100 mM NaHCO$_3$, pH 8.2, 100 mM NaCl) at a final protein concentration of 2–20 µM. NHS-dyes (AAT iFluor NHS-405, AAT iFluor NHS-546, Invitrogen Alexa NHS-647) were added to the protein at 1:1 molar ratio, and the labelling proceeded at room temperature for 1 h. The reaction was quenched by 100 mM Tris pH 8.2. Free dyes were removed using the PD10 desalting column (Cytia). The labelling efficiency was 50–100%. Labelled proteins were mixed with unlabelled proteins so that the labelling efficiency was about 2%, and the sample was concentrated to 200–1,600 µM. The samples were cleared at 14,000 rpm in the Eppendorf 5417C centrifuge for 10 min before freezing in liquid nitrogen. Purified LPHN3 proteins were directly labelled using AAT iFluor NHS-488 as described above.

## Phase-transition imaging and sedimentation assay

Unless otherwise indicated, proteins were added to a final concentration of 10 µM GKAP, 10 µM HOMER3, 10 µM PSD95, 10 µM SHANK3, 6 µM LPHN3 E31, 6 µM LPHN3 E32, 6 µM LPHN3 E31(ΔPBM), 10 µM TEN2, 10 µM FLRT3 in assay buffer (20 mM Na-HEPES pH 7.4, 150 mM NaCl, 0.002% LMNG, 0.0002% CHS). Protein mixtures were incubated at room temperature for 10–20 min. For imaging experiments, 5 µl of sample was loaded onto a channelled slide (ibidi, 80666), which was designed with a cover to minimize evaporation of small-volume samples. For the pelleting experiments, 10 µl of sample was centrifuged at 5,000 rpm in the Eppendorf 5417C centrifuge for 5 min. The supernatant was immediately removed and the pellet was resuspended in 2× SDS loading buffer. All of the samples were analysed using SDS–PAGE and stained in Coomassie G-250 blue. We quantified the LPHN3 pellet percentage using the N-terminal domain, which has the same sequence for all three *Lphn3* constructs.

## FRAP analysis

After phase separation was completed, strong-excitation laser intensities were used to bleach all of the channels of a small area for approximately 10 seconds, after which the fluorescence of the photobleached spot was recorded for 6 min. Recovery traces were fitted with the exponential equation $y = ae^{-bx} + c$ to extrapolate the $t_{1/2} = (\ln 2)/b$. Only the FRAP recovery kinetics were interpreted because the recovery percentage is highly sensitive to the duration of photobleaching, which was not precisely controlled in this experiment.

## Statistics and reproducibility

Most statistical tests were performed using two-sided *t*-tests, as indicated. To control for family-wise error during multiple comparisons, two-sided Tukey's tests were used in parallel and the adjusted *P* values are summarized in Supplementary Tables 1 and 2, and do not change the conclusions drawn from *t*-tests in this work. Gene counts from the high-throughput sequencing dataset were analysed using two-sided Wald test of DESeq2 for bulk RNA-seq datasets, and two-sided Wilcoxon rank-sum tests for the scRNA-seq dataset. For all box plots: the lowest datapoint shows the minimum value; the highest datapoint shows the maximum value; the centre line shows the median; and the box limits show the interquartile range (25th to 75th percentile). Representative experiments were repeated independently the following number of times: Fig. 2b, *n* = 6; Fig. 2e, *n* = 3; Fig. 3a, *n* = 3; Fig. 3d,e, *n* = 5 for control/*Lphn3* KO and 6 for E31 KO; Fig. 4b, *n* = 4; Fig. 4d,f, *n* = 3; Extended Data Fig. 3a, *n* = 3; Extended Data Fig. 5a, *n* = 3; Extended Data Fig. 5e, *n* = 3 (rows 1 and 2), *n* = 5 (rows 3 and 4) and *n* = 6 (row 5); Extended Data Fig. 6a–i, *n* = 2; Extended Data Fig. 7a, *n* = 3; Extended Data Fig. 7c, *n* = 3; Extended Data Fig. 7f, *n* = 3; Extended Data Fig. 7h, *n* = 3; Extended Data Fig. 8d,e, *n* = 3; Extended Data Fig. 8f,g, *n* = 6; Extended Data Fig. 8h, *n* = 10; Extended Data Fig. 9d, *n* = 3; Extended Data Fig. 9f, *n* = 3.

## Reporting summary

Further information on research design is available in the Nature Portfolio Reporting Summary linked to this article.

## Data availability

All raw data supporting the findings of this study have been deposited in the Stanford Data Repository (https://purl.stanford.edu/nj297xj2116), except for the high-throughput sequencing data generated from this study, which were deposited at the GEO under accession code GSE240791. Other public datasets analysed in this work include PacBio long-read mRNA sequencing data (BioProject: PRJNA547800); cell-type-specific sequencing data (GEO: GSE133291 and GSE115746); and neuronal-activity-regulated transcriptome datasets (GEO: GSE175965, GSE104802 and GSE152632).

## Code availability

Codes used for this study were deposited in the Stanford Data Repository (https://purl.stanford.edu/nj297xj2116).

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

**Acknowledgements** We thank J. H. Trotter and Z. Sun for help with imaging; and C. Gui, H. Wang, K. Liakath-Ali, J. Dai and X. Chen for discussions. This work was supported by grants from National Institute of Mental Health to T.C.S. (5R01 MH126929-02); a Stanford Maternal & Child Health Research Institute Postdoctoral Support grant to S.W. (1220319-117-JHACT); R24DK116195, the NIMH Psychoactive Drug Screening Program and the Michael Hooker Distinguished Professorship to B.L.R.; and the Swiss National Science Foundation to W.S. (SNSF 211053). S.R.Q. is a Chan Zuckerberg Investigator.

**Author contributions** S.W. performed all experiments and analysed all data, except the TRUPATH assay, which was performed by C.D. and supervised by B.L.R., and the scRNA-seq data from the mPFC region for neuronal activity analysis, which was performed by W.S. and supervised by S.R.Q.; S.W. and T.C.S. conceptualized the project, designed the experiments and wrote the manuscript with input from all of the authors. All of the authors contributed to data analyses.

**Competing interests** The authors declare no competing interests.

**Additional information**
**Correspondence and requests for materials** should be addressed to Shuai Wang or Thomas C. Südhof.

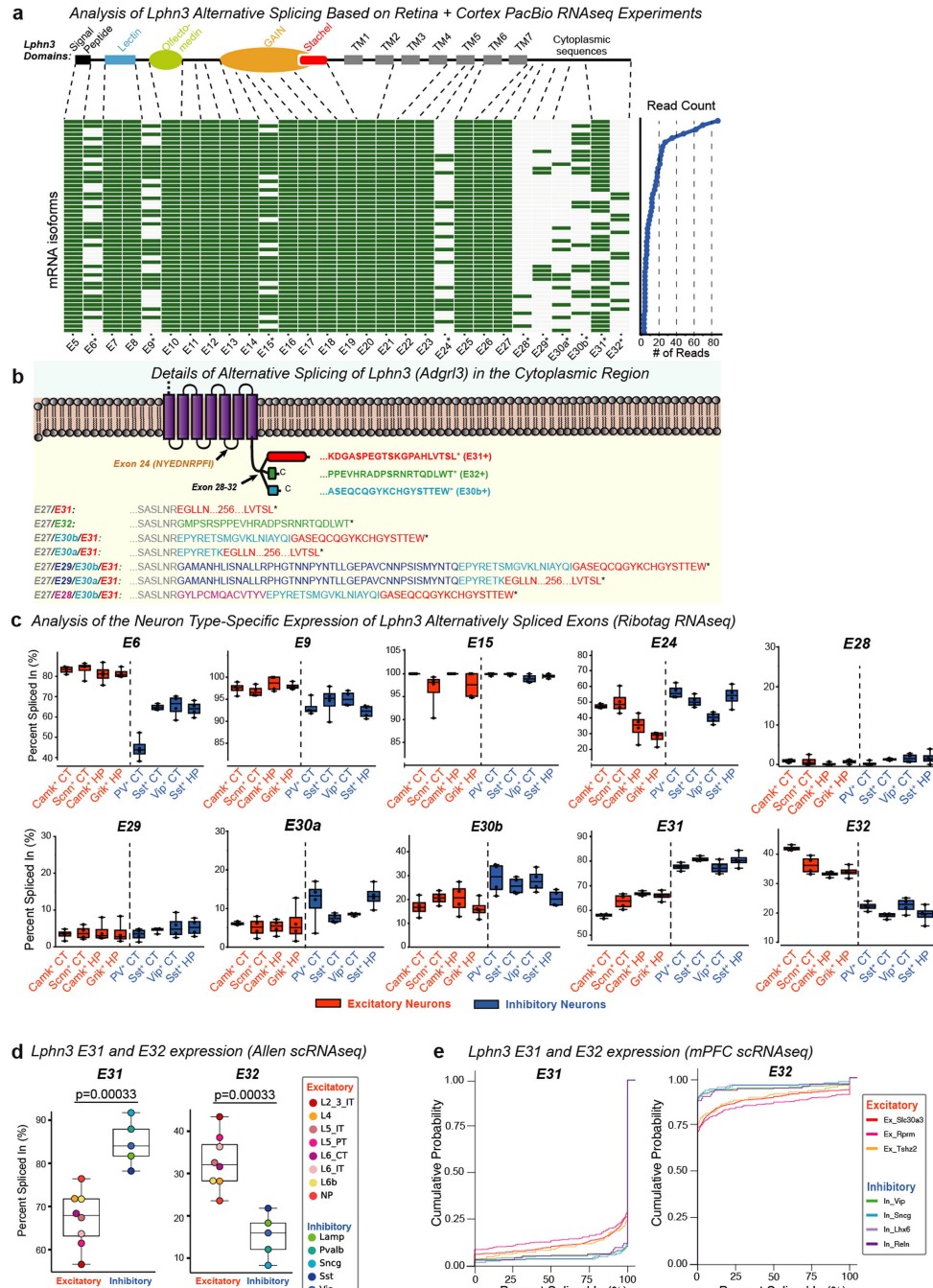

**Extended Data Fig. 1 | Alternative splicing of Lphn3 (*Adgrl3*) transcripts (*a* & *b*) and demonstration that a subset of the sites of alternative splicing of Lphn3 exhibits a high degree of cell type-specific expression as revealed by RNAseq analyses (*c*–*e*). a**, Analysis of a long-read PacBio sequencing[16] uncovers extensive combinatorial alternative splicing of Lphn3. Reads are depicted as heatmaps (green boxes = Included exons; light grey boxes = excluded exons; asterisks = alternatively spliced exon). Each row represents a splice variant combination whose abundance is shown on the right. Note 'a' and 'b' designations (such as '30a' and '30b') are alternative splicing donor or acceptor variants within an exon. For a similar analysis of Lphn1 and Lphn2, see Extended Data Fig. 2a–d. **b**, Details of alternative splicing in the cytoplasmic region of Lphn3. The diagram depicts the amino acid sequences of alternatively spliced variants (asterisk = stop codon; only the 7 TMR region and cytoplasmic sequences of Lphn3 are shown). Note Exons 28 and 29 are in-frame but Exon30 can be present

as Exon30a or Exon30b, of which Exon30b shifts the reading frame of Exon31. **c**, Neuron type-specific alternative splicing of Lphn3 from a ribosome-associated transcriptome study[18]. The abundance of each exon in PSI (percent spliced in) for 8 indicated neuron types from two brain regions (HP: hippocampus, CT: cortex). Each datapoint (n = 4) represents one sample from 1 animal for excitatory neurons, and 2 animals for inhibitory neurons. **d** & **e**, Analyses of Lphn3 Exon31 and Exon32 level in single-cell datasets from primary visual cortex and anterior lateral motor cortex[42] (*d*), or of medial prefrontal cortex[37] (*e*). In *d*, all neurons corresponding to an indicated type were combined as one datapoint. The percent spliced-in for Exon31 and Exon32 of Lphn3 is shown for each type. Two-sided t-test was used to calculate statistical significance (n = 8 biologically independent excitatory neurons and n = 5 for biologically independent inhibitory neurons). In *e*, the cumulative probability of splicing percentage for all neurons of each type were plotted.

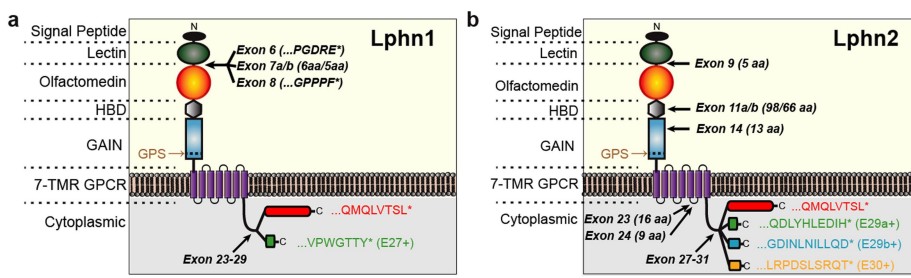

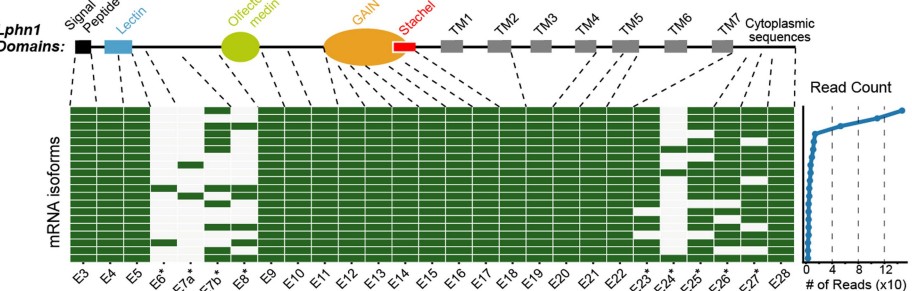

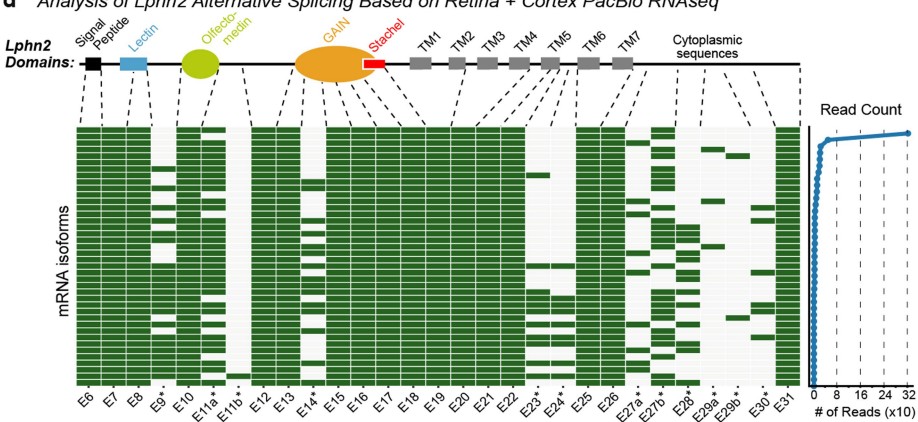

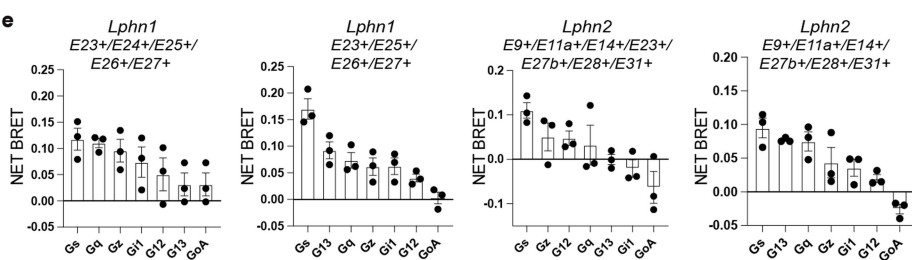

**Extended Data Fig. 2 | Alternative splicing of Lphn1 (*Adgrl1*), Lphn2 (*Adgrl2*), and regulation of Lphn1 and Lphn2 G-protein coupling by alternative splicing. a** & **b**, Schematic of Lphn1 (*Adgrl1*) (a) and Lphn2 (*Adgrl2*) (b) alternative splicing with a depiction of the amino acid sequences of some of the resulting variants (asterisk = stop codon). **c** & **d**, Analysis of a long-read PacBio sequencing dataset[16] also reveals extensive combinatorial alternative splicing of Lphn1 and Lphn2 mRNAs. Reads are depicted as heatmaps (green boxes = included exons; light grey boxes = excluded exons; asterisks = alternatively spliced exon). Each row represents a splice variant combination whose abundance is shown on the right (only the most abundant variants are shown). **e**, G-protein coupling preferences of two Lphn1 and Lphn2 splice variants revealed by TRUPATH analyses. The constitutive G-protein coupling strength (represented by the NET BRET signal) of the indicated Lphn1 and Lphn2 isoform were measured by TRUPATH assay in HEK293 cell. Splice variants with indicated spliced-in exons are shown. BRET signals at 300 ng receptor-transfected condition were normalized to the 0 ng transfected baseline. Graphs show means ± SEM from independent experiments (n = 3).

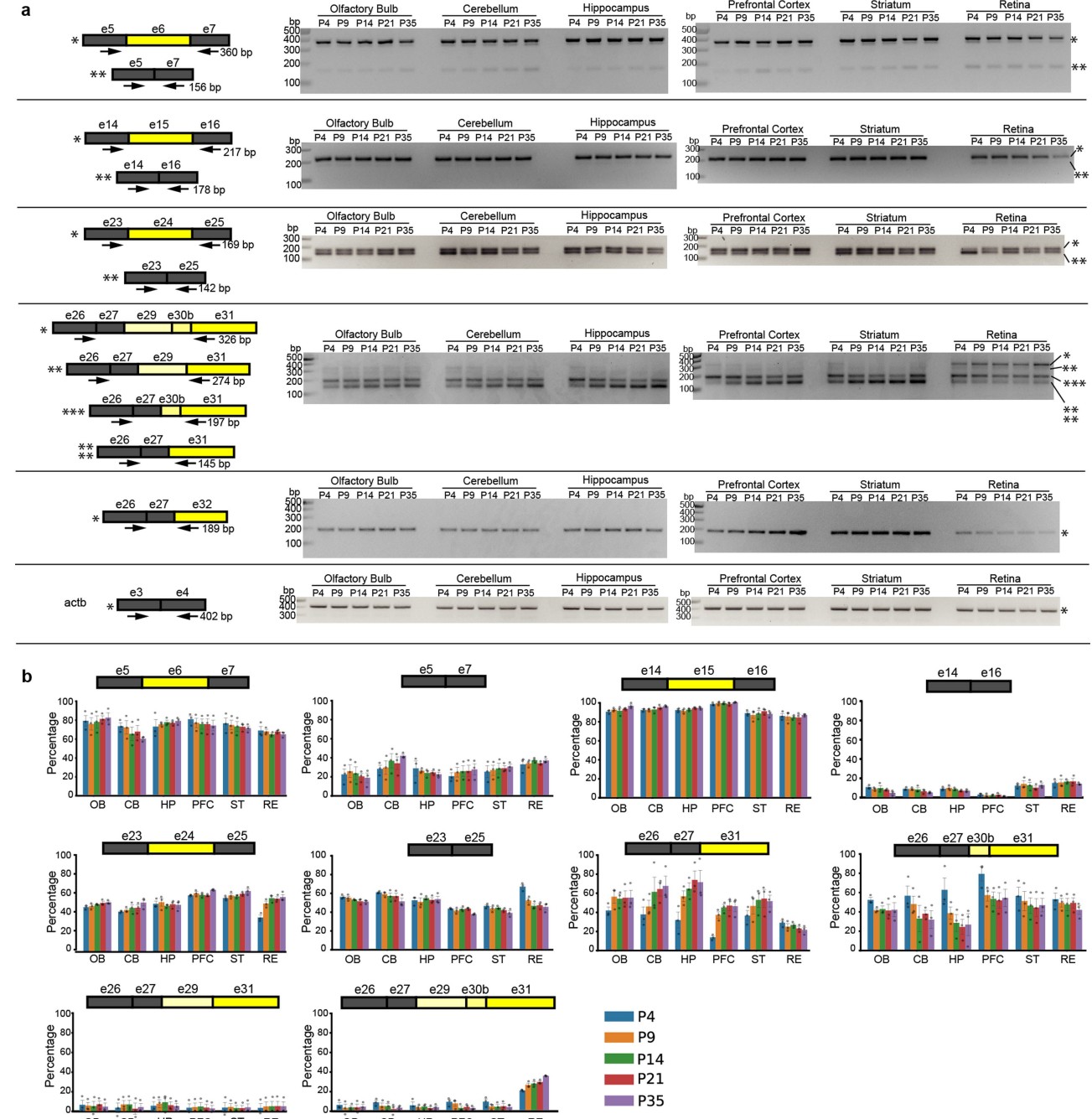

**Extended Data Fig. 3 | Diverse patterns of Lphn3 alternative splicing analysed by RT-PCR in different brain regions and at different times of postnatal development. a**, Total RNA isolated from the indicated brain regions of C57BL/6 mice at the indicated postnatal developmental timepoints were analysed by RT-PCRs using primers (labelled in arrows) at exon-exon junctions. RT-PCR products were separated by 2% agarose gel electrophoresis; bands are marked based on the predicted sizes of the alternatively spliced variants shown above each gel. Raw gels are in Supplementary Figs. 1 and 2. **b**, Quantification of the percentage of multiple variants derived from the same PCR primer pairs. Data are presented as mean values ± SEM (n = 3 biological independent animals). Raw gels are in Supplementary Figs. 1 and 2. Note Exon32 of Lphn3 and β-actin has only one product, therefore were not quantified.

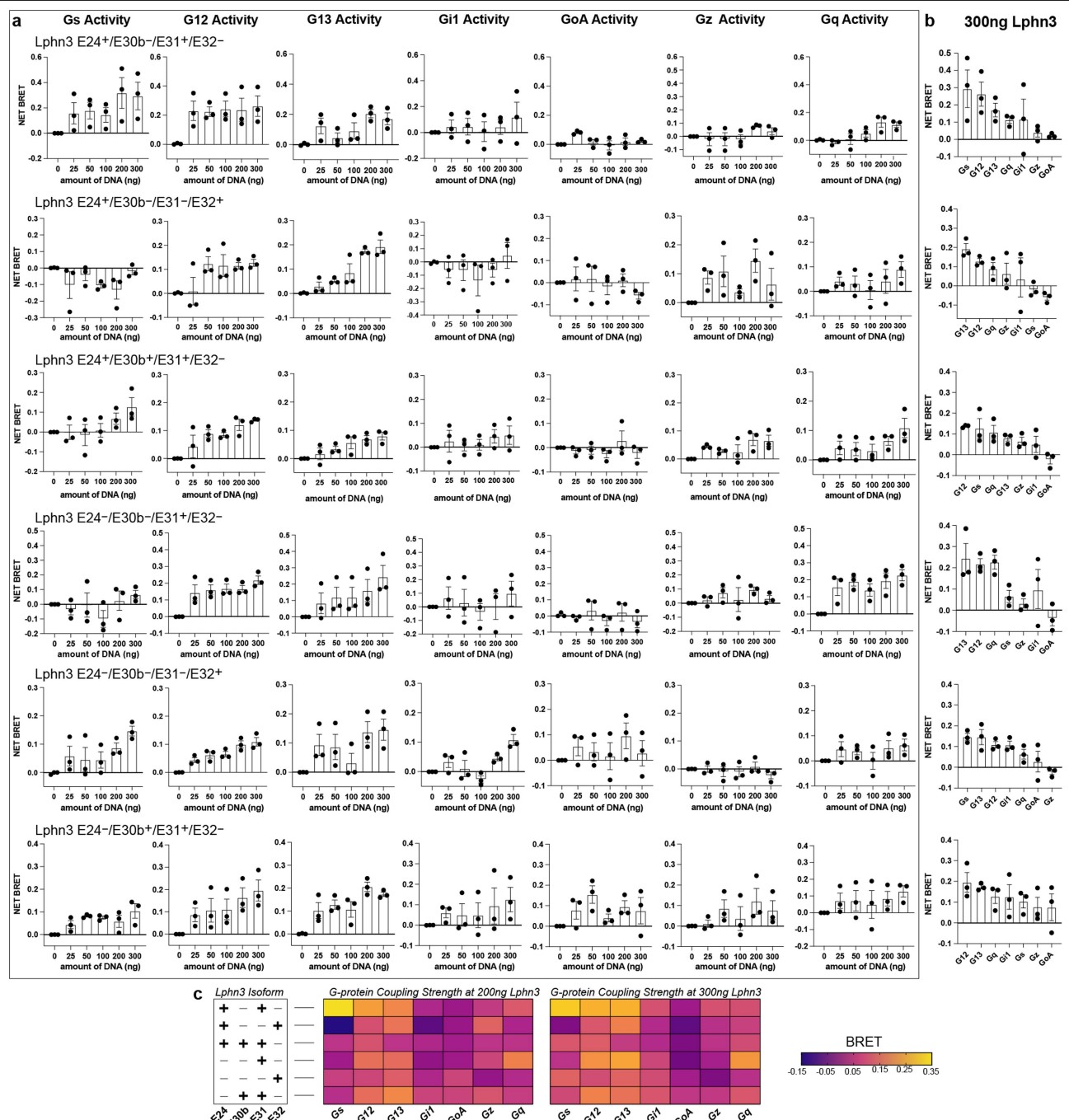

**Extended Data Fig. 4 | Detailed TRUPATH analyses of G-protein coupling mediated by six different Lphn3 splice variants. a**, TRUPATH assays reveal G-protein coupling signals (NET BRET values) mediated by 6 Lphn3 splice variants expressed at different concentrations. HEK293 cells were transfected with Lphn3 expression plasmids at concentrations of 0–300 ng and the TRUPATH signal was monitored as described. Graphs show means ± SEM from independent experiments (n = 3). All signals were normalized to 0 ng transfected baseline. **b**, Summary graph of the TRUPATH signal at the highest Lphn3 concentration (300 ng). Graphs show means ± SEM from independent experiments (n = 3). All signals were normalized to 0 ng transfected baseline. **c**, Heatmap plot of the G-protein coupling strength of the indicated Lphn3 variants at the 200 ng and 300 ng Lphn3 plasmid transfection condition.

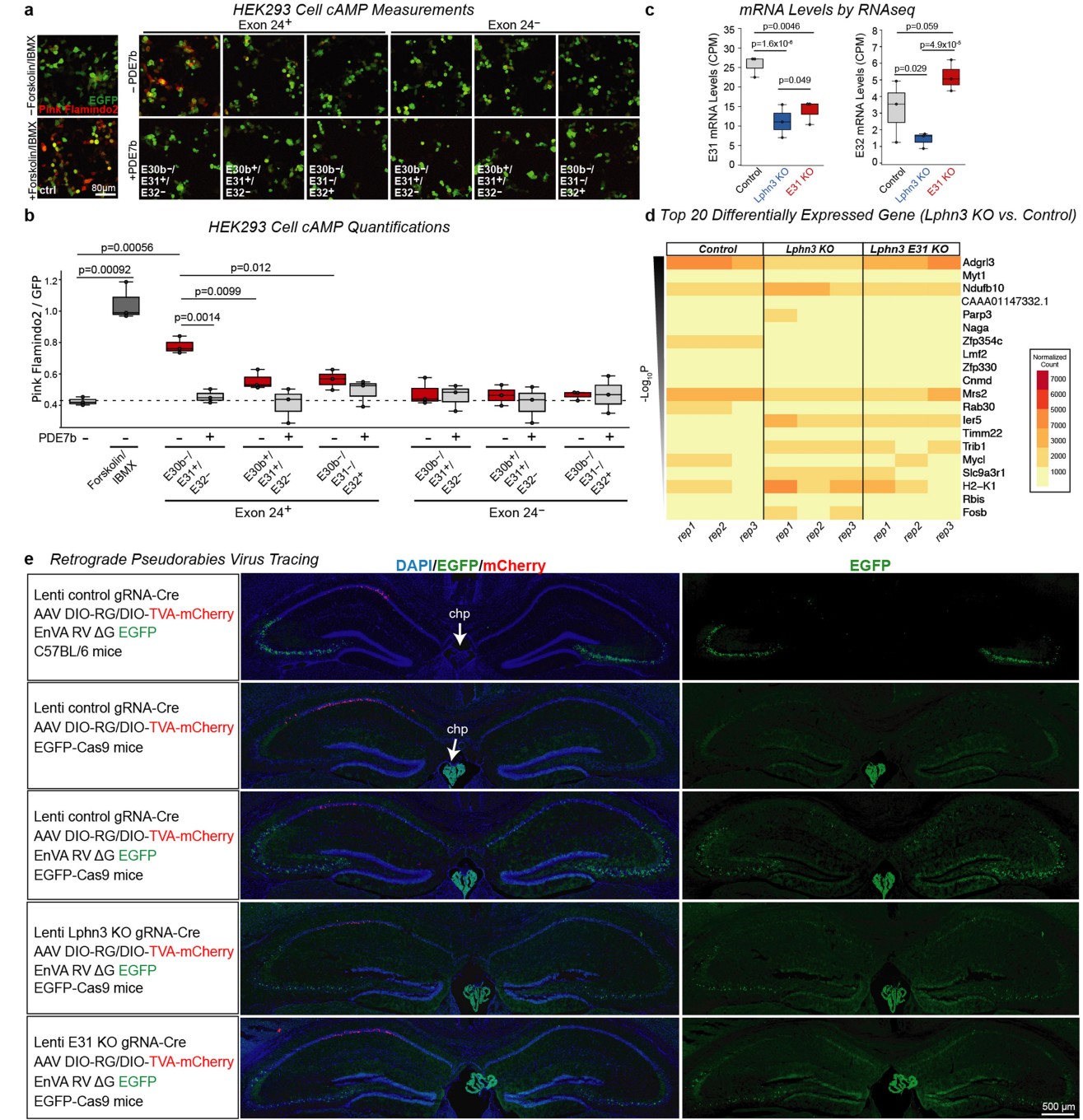

**Extended Data Fig. 5 | Further data characterizing cAMP assays, RNAseq analyses and pseudorabies virus tracing experiments. a,** Representative images of the cAMP-dependent Pink Flamindo2 and the EGFP signal monitored during cAMP assays. Intracellular cAMP levels were monitored in live HEK293T cells that had been transfected with Pink Flamindo2, EGFP and Gs($\alpha$/$\beta$/$\gamma$). Where indicated, Lphn3 variants and/or PDE7b (a cAMP-specific phosphodiesterase that blocks cAMP signalling) were co-transfected. Forskolin/IBMX were only applied to the control group. Imaging was performed by 10x confocal microscopy. **b,** Quantification of cAMP signals. The Pink Flamindo2 signal was normalized by the GFP signal. Each datapoint represents one independent culture (n = 3). Two-sided t-test was used to calculate the statistical significance (only relevant p values < 0.05 are shown). **c,** RNAseq confirms that the E31 KO

and the Lphn3 KO similarly ablate expression of Exon 31-containing Lphn3 mRNAs (left) but have opposite effects on Exon 32-containing Lphn3 mRNAs (right). Statistics used n = 3 biologically independent cultures after removing genes less than 72 reads, with two-sided Wald test by DESeq2 with p values shown. **d,** Heatmap illustrating the most significant changes in gene expression observed in three independent RNAseq experiments (rep1-rep3). The normalized count of each replicate for each condition is shown for the top 20 genes (ranked by the p value of Lphn3 KO vs control). **e,** Additional controls to illustrate the weak GFP signal present in Cas9 mice that is observed in dentate gyrus granule cells and in the choroid plexus (chp) but not in CA3 pyramidal cells (2nd row vs. 1st row).

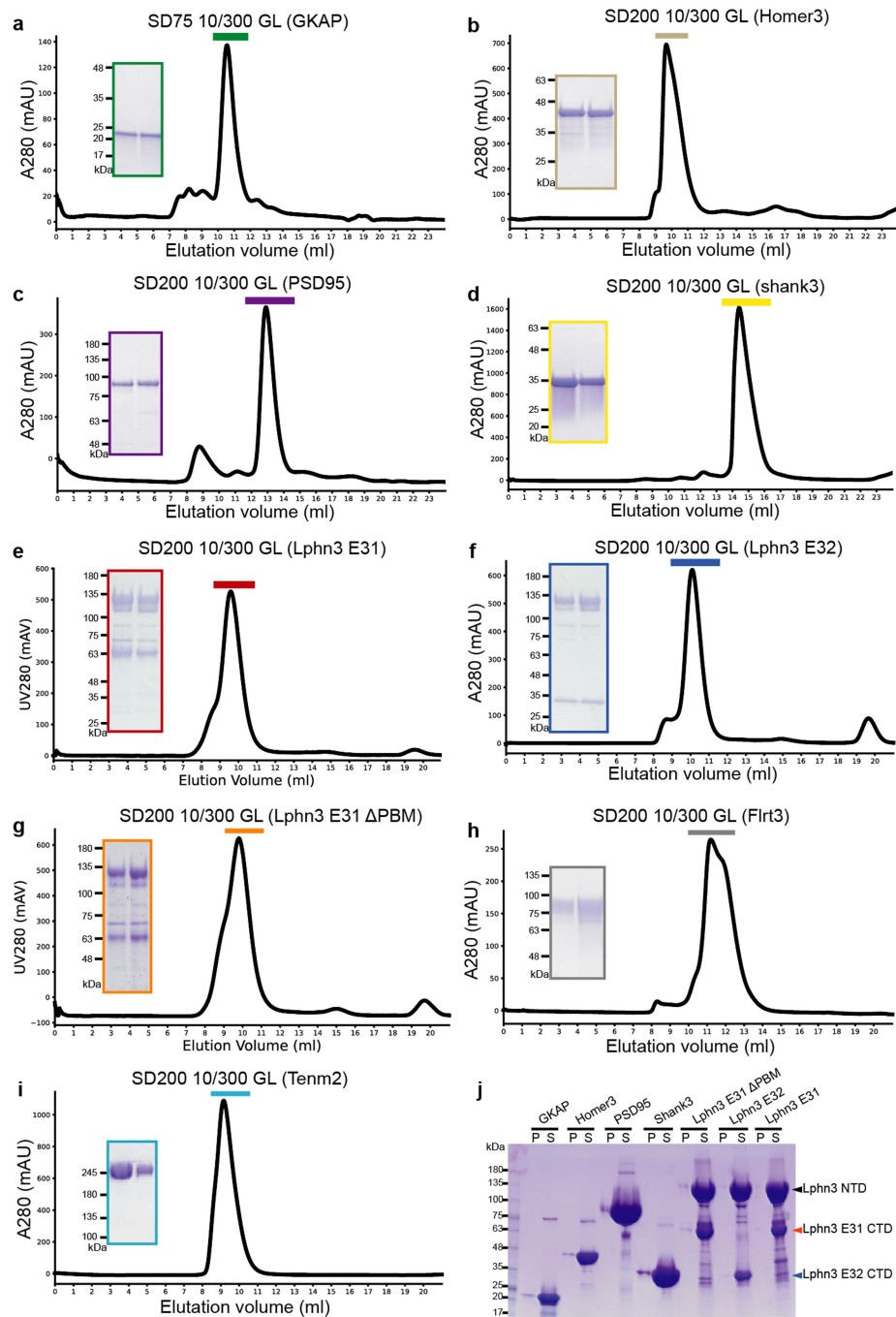

**Extended Data Fig. 6 | Characterization of purified proteins used for phase-transition experiments. a–i,** Chromatographs of proteins on size-exclusion columns during the last step of purification. Peak fractions (horizontal bar) were analysed by SDS-PAGE and Coomassie staining (insets) and used as final purified proteins for phase separation experiments. Note that Lphn3 is autocleaved in the GAIN domain to produce N- (apparent molecular weight ~130 kD) and C-terminal fragments (apparent molecular weight ~65 kD for Lphn3-E31 and Lphn3-E31-ΔPBM, and ~30 kD for Lphn3-E32) which are non-covalently

bound to each other within the GAIN domain[40]. Raw gels are in Supplementary Fig. 3. **j,** Sedimentation behaviour of individual proteins. Purified proteins were centrifuged in the same condition, as used in the phase-transition assay. Supernatant (S) and pellet (P) were subject to SDS-PAGE for analysis. The concentrations of proteins used were: 80 µM GKAP, 100 µM PSD95, 80 µM Homer3, 200 µM Shank3, 40 µM Lphn3-E31 ΔPBM, 40 µM Lphn3-E32, and 40 µM Lphn3-E31.

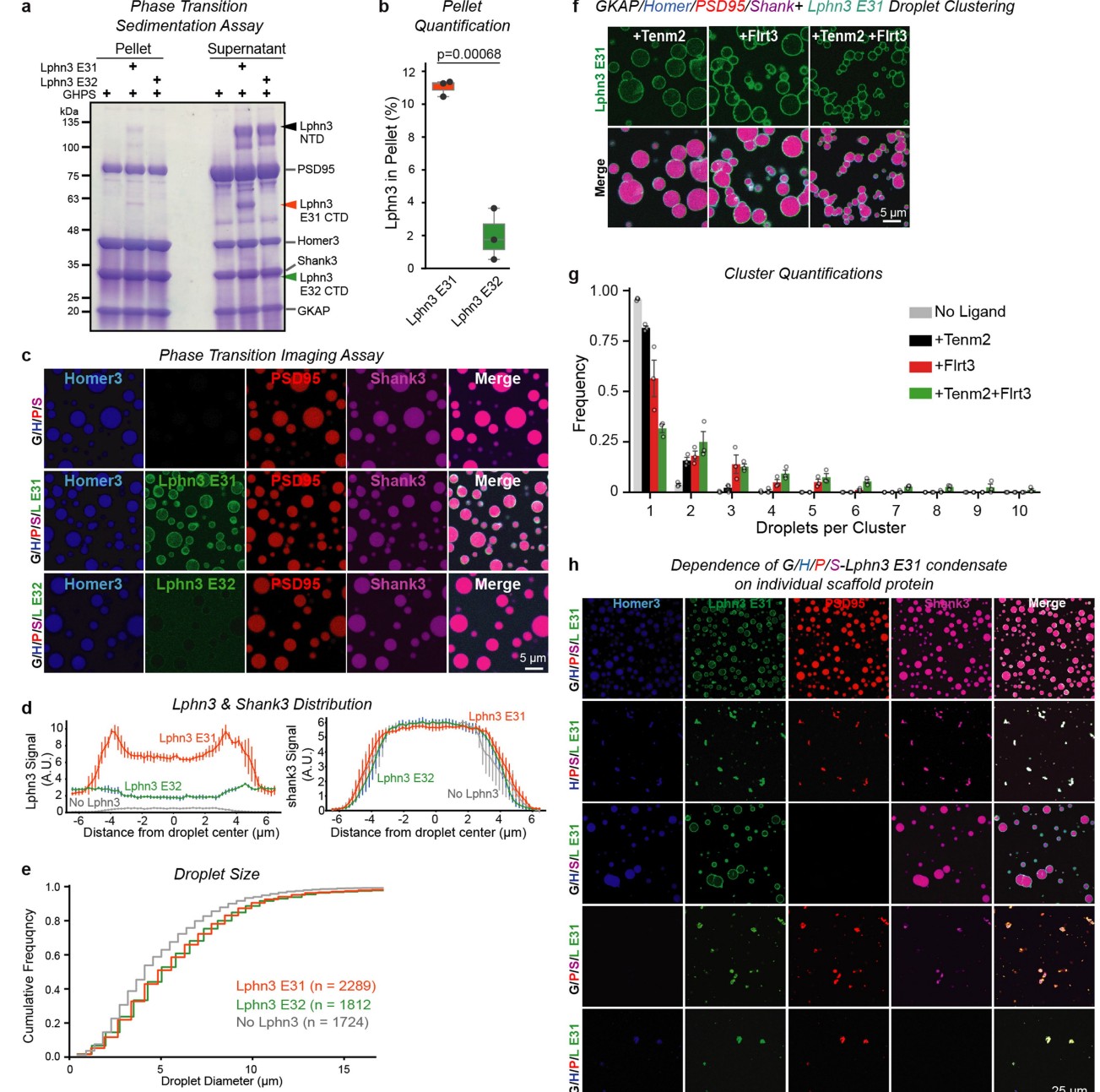

**Extended Data Fig. 7 | Independent replication of the Lphn3-E31 dependent recruitment of phase-separated scaffold protein condensates at a higher concentrations of scaffold proteins (38 μM GKAP, 30 μM Homer3, 19 μM PSD95, 32 μM Shank3). a**, Sedimentation assay of phase transition complexes. The scaffold protein mixture containing GKAP, Homer3, PSD95, and Shank3 (GHPS) was incubated with indicated Lphn3. Pellet and supernatant were separated by centrifugation and analysed by SDS-PAGE. **b**, Quantification of Lphn3 pelleting in the sedimentation assay. The bands corresponding to the Lphn3 NTD were used for analysis. Statistics used n = 3 independent experiments and two-sided t-test for calculating p-values. **c**, Imaging of phase transitioned complexes. Homer3 (H), Lphn3 (E31 and E32), PSD95 (P), and Shank3 (S) were labelled by NHS-ester fluorophore 405, 488, 546, 647, respectively, while GKAP (G) was unlabelled. **d**, Quantification of the Lphn3 (left) and Shank3 (right) fluorescence signal across the phase-separated droplet illustrating the surface localization of Lphn3 E31 on the droplet filled with Shank3. Data are means ± SEM (n = 3 independent experiments). **e**, Quantification of the sizes of phase-transitioned droplets formed by postsynaptic scaffold proteins GKAP/Homer3/PSD95/Shank3 in the absence of Lphn3 or in the presence of Lphn3 E31 or Lphn3 E32. n represents individual droplet. **f**, Representative images of phase-transitioned postsynaptic scaffold protein complexes containing Lphn3 E31 that were clustered by presynaptic ligands Tenm2 and Flrt3. **g**, Quantification of the clustering effect of presynaptic Tenm2 and Flrt3 ligands on Lphn3 E31 coated, phase-transitioned postsynaptic scaffolding protein complexes. Data are means ± SEM (n = 3 independent experiments). **h**, Contribution of individual postsynaptic scaffold protein to phase transitions in the presence of Lphn3 containing Exon 31. Individual scaffold protein was omitted and droplets were subjected to the same imaging condition as above.

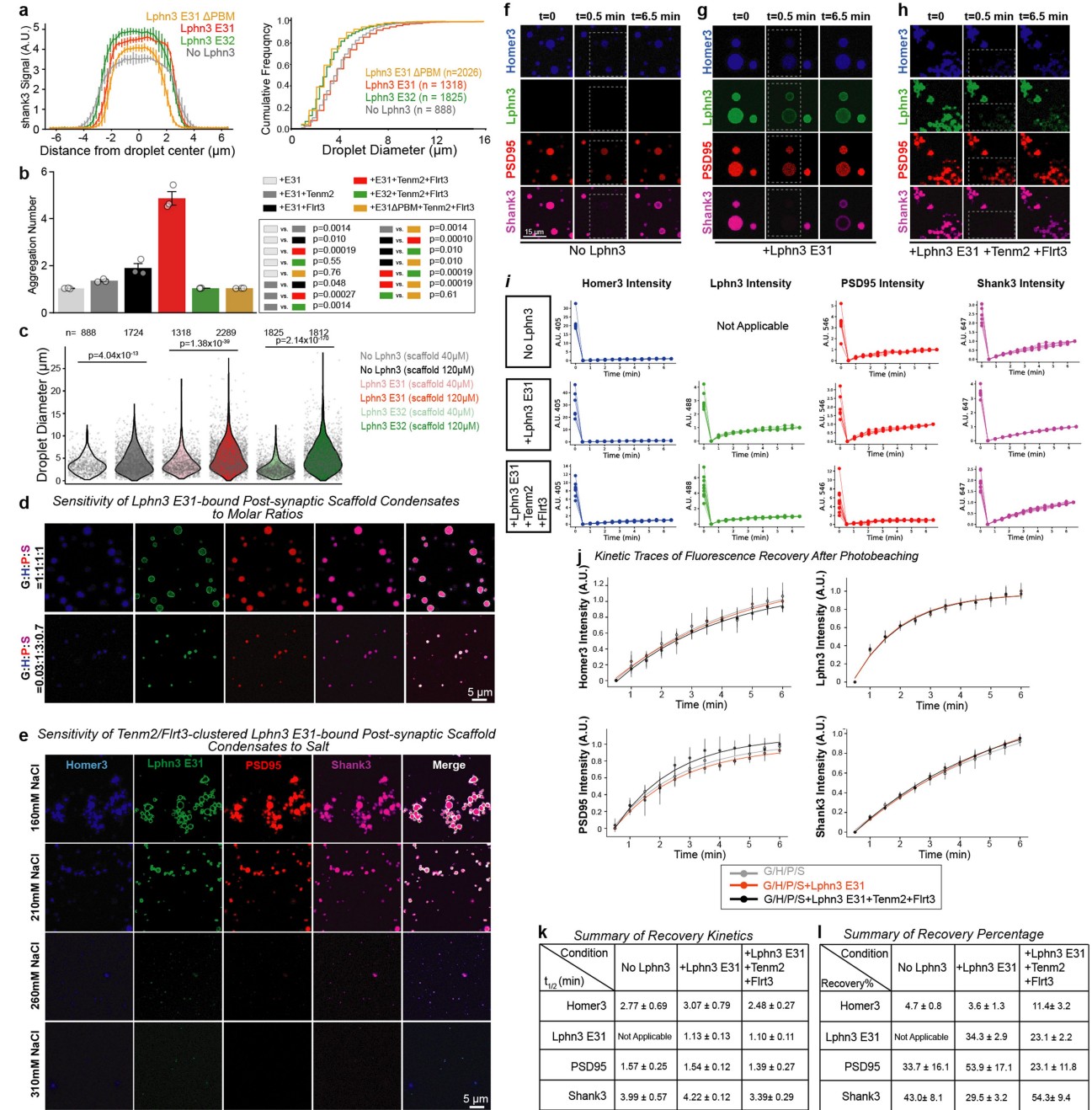

**Extended Data Fig. 8 | Further characterization of phase-separated, Lphn3-E31 coated post-synaptic scaffold condensates and FRAP (fluorescence recovery after photobleaching) experiments. a**, Quantification of the Shank3 fluorescence signal across the phase-separated droplet (left), and cumulative plot of the sizes of phase-transitioned droplets (right), formed by postsynaptic scaffold proteins GKAP/Homer3/PSD95/Shank3 (GHPS) with or without different Lphn3 proteins. Data on the left are means ± SEM (n = 3 independent experiments); n on the right panel represents individual droplet. **b**, Quantifying the aggregation number of phase-transitioned droplets formed by GHPS with or without different Lphn3 proteins, with indicated presynaptic ligands. Aggregation number is calculated by dividing the number of droplets by the number of continuous, phase-separated region. Statistics used n = 3 independent experiments with two-sided t-test showing p-values (inset). Related to Fig. 4g. **c**, Comparison of droplet sizes using 40 μM or 120 μM total scaffold proteins.

Source data from Extended Data Figs. 7e and 8a. Statistics used two-sided t-test with p-values and n (independent droplet) shown in the figure. **d**, Sensitivity of phase separation to the stoichiometry. Physiological molar ratio[49] GKAP: Homer3:PSD95:Shank3 = 0.03125:1:3.09:0.73 at a total concentration of 40 μM was used. **e**, Sensitivity of phase separation to ionic strength. **f-h**, Representative images from FRAP experiments, with the target area (inset) bleached at 0.5 min. scale bar 15 μm applies to all images in f-h. **i**, Fluorescence intensity of all channels during the FRAP experiments. Each trace represents one independent experiment (n = 6, 6, 10 independent experiments for "No Lphn3", "+Lphn3 E31", and "+Lphn3 E31 +Tenm2 +Flrt3", respectively). **j**, Recovery traces showing means ± SD were fitted to exponential equation to extrapolate $t_{1/2}$. Also see method. n = 6, 6, 10 independent experiments for "No Lphn3", "+Lphn3 E31", and "+Lphn3 E31 +Tenm2 +Flrt3", respectively. **k & l**, Summary of recovery kinetics (k) and percentage (l).

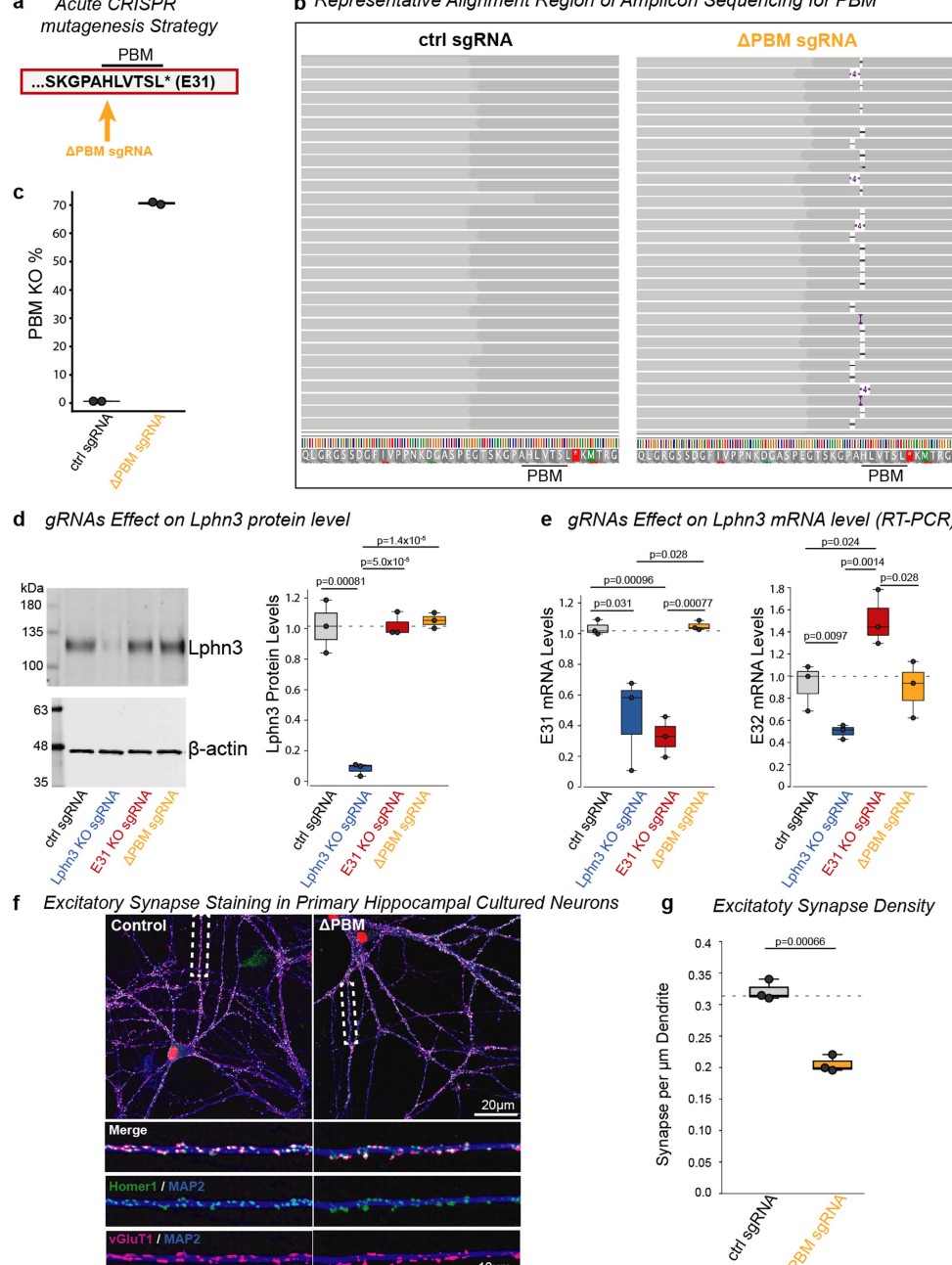

**a** *Acute CRISPR mutagenesis Strategy*

PBM

...SKGPAHLVTSL* (E31)

ΔPBM sgRNA

**c**

PBM KO %

**b** *Representative Alignment Region of Amplicon Sequencing for PBM*

ctrl sgRNA    ΔPBM sgRNA

PBM    PBM

**d** *gRNAs Effect on Lphn3 protein level*

kDa
180
135    Lphn3
100

63
48    β-actin
35

Lphn3 Protein Levels

p=1.4x10⁻⁶
p=5.0x10⁻⁶
p=0.00081

**e** *gRNAs Effect on Lphn3 mRNA level (RT-PCR)*

E31 mRNA Levels

p=0.028
p=0.00096
p=0.031    p=0.00077

E32 mRNA Levels

p=0.024
p=0.0014
p=0.028
p=0.0097

**f** *Excitatory Synapse Staining in Primary Hippocampal Cultured Neurons*

Control    ΔPBM

20μm

Merge

Homer1 / MAP2

vGluT1 / MAP2

10μm

**g** *Excitatoty Synapse Density*

Synapse per μm Dendrite

p=0.00066

**Extended Data Fig. 9 | Selective deletion of the PDZ-domain binding motif (PBM) in Exon31 of the Lphn3 gene decreases synapse numbers.**
**a**, Experimental strategy for the selective deletion of the PBM of Lphn3-E31. A gRNA targeting the genomic DNA that encodes the PBM in Exon31 was used in acute CRISPR experiments (ΔPBM) in hippocampal cultures. **b**, Monitoring the efficiency of CRISPR-mediated deletion of the PBM in cultured neurons. The interactive Genome Viewer (v2.16.2) shows a representative alignment of sequences near the PBM region from neurons subjected to CRISPR with the control and the ΔPBM gRNA. **c**, Quantification of PBM KO efficiency. Insertion/deletion which caused frameshift of the coding region of PBM, and mutation within the PBM region were classified as PBM KO events. n = 2 independent cultures. **d**, Lphn3 immunoblots showing that both the E31 KO and ΔPBM gRNA do not change Lphn3 protein levels, whereas the Lphn3 KO ablates Lphn3

expression. Statistics from n = 3 independent cultures used two-sided t-test for calculating p-values (only showing significant p < 0.05). **e**, RT-PCR demonstrates that the E31 KO and Lphn3 KO similarly ablate expression of Exon 31-containing Lphn3 mRNAs (left) but have opposite effects on Exon 32-containing Lphn3 mRNAs (right), whereas ΔPBM does not change Exon31 or Exon32 mRNA levels. Statistics from n = 3 biologically independent cultured neurons used two-sided t-test showing relevant p-values < 0.05. **f & g**, Selective deletion of the Lphn3 PBM significantly decreases the excitatory synapse density (**f**, representative images of cultured hippocampal neurons stained with antibodies to vGluT1, Homer1 and MAP2; **g**, summary graph of the density of puncta that were positive for both vGluT1 and Homer1). Statistics from n = 3 biologically independent cultured neurons used two-sided t-test for calculating p-values.

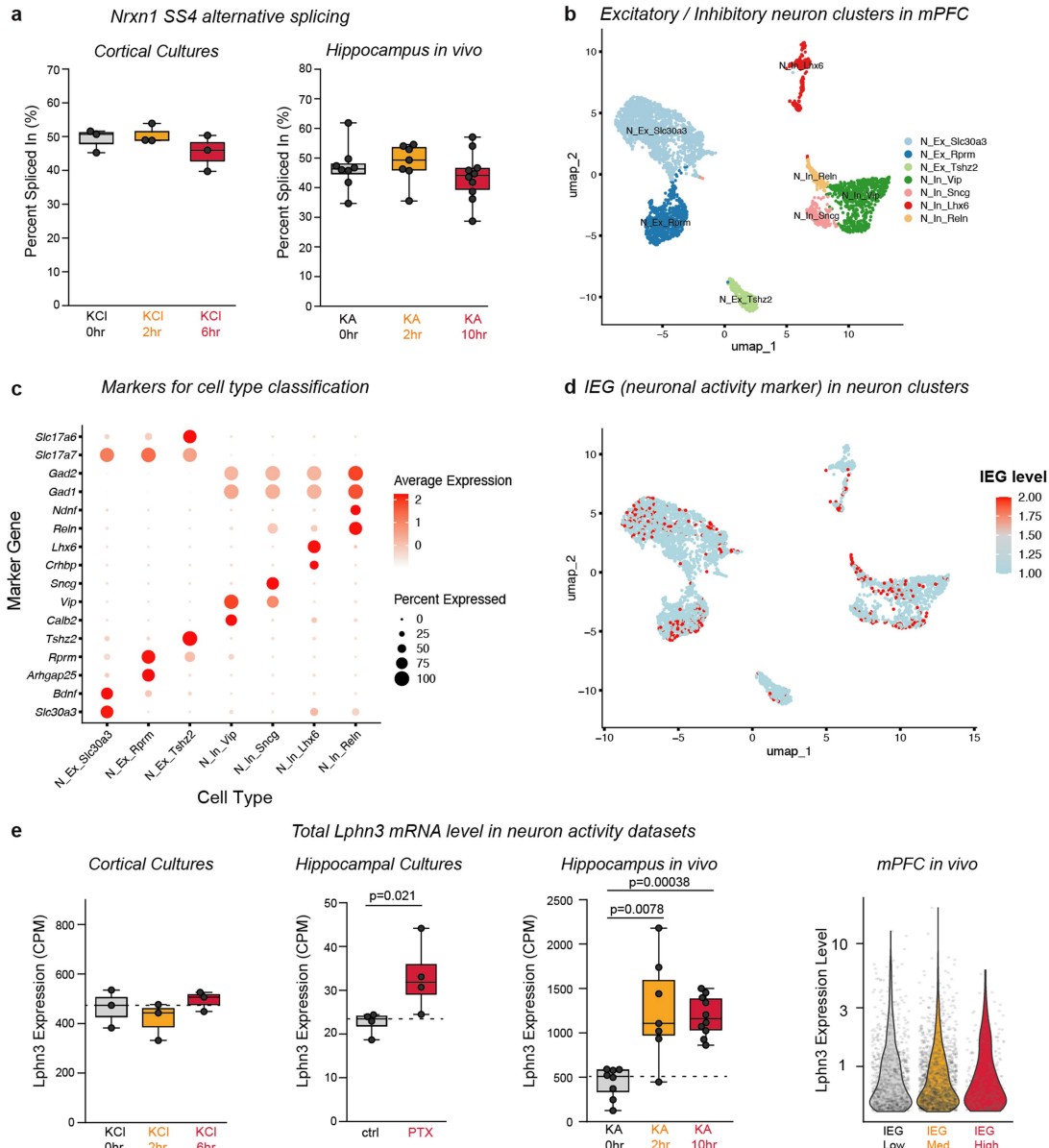

**Extended Data Fig. 10 | Additional analyses of the activity-dependent splicing of Lphn3 Exon31 and Exon32. a**, Neuronal activity does not change the Nrxn1 SS4 splicing level, as reported before[36]. This serves as a negative control to ensure that the strong activation conditions (KCl and kainate) used did not impair neuronal survival. Two-sided t test was used to calculate statistical significance showing only significant p < 0.05. n = 3 biologically independent cortical cultures; n = 8, 7, 10 biologically independent hippocampal tissues after 0 hr, 2 hr, and 10 hr Kainate treatment, respectively. **b**, UMAP plot illustrating the clustering of neurons (NeuN⁺) into subtypes of excitatory and inhibitory groups in a single-cell RNAseq dataset obtained from the mPFC[37]. N, neuron; Ex, excitatory; In, inhibitory. Scl30a3, Rprm, Tshz2 are markers identifying distinct groups of excitatory neurons, while Vip, Sncg, Lhx6 and Reln are markers identifying distinct groups of inhibitory neurons.

**c**, Expression levels of the indicated marker genes in different types of excitatory and inhibitory neurons in the mPFC single-cell RNaseq dataset. **d**, Mapping of the expression levels of immediate early genes (IEG) as activation markers onto the neuron types identified in the UMAP plot of b. **e**, Total Lphn3 levels determined in the various RNAseq datasets used for the analysis of activity-dependent alternative splicing of Lphn3. Two-sided Wilcoxon rank-sum test was used to calculate statistical significance showing only significant p < 0.05. n = 3 biologically independent cortical cultures; n = 4 biologically independent hippocampal cultures; n = 8, 7, 10 biologically independent hippocampal tissues after 0 hr, 2 hr, 10 hr Kainate treatment, respectively; the single cell study analysed n = 903, 1526, 232 independent cells for IEG low, medium, and high, respectively.

# Reporting Summary

## Statistics

For all statistical analyses, confirm that the following items are present in the figure legend, table legend, main text, or Methods section.

| n/a | Confirmed | |
|---|---|---|
| ☐ | ☒ | The exact sample size (*n*) for each experimental group/condition, given as a discrete number and unit of measurement |
| ☐ | ☒ | A statement on whether measurements were taken from distinct samples or whether the same sample was measured repeatedly |
| ☐ | ☒ | The statistical test(s) used AND whether they are one- or two-sided *Only common tests should be described solely by name; describe more complex techniques in the Methods section.* |
| ☒ | ☐ | A description of all covariates tested |
| ☐ | ☒ | A description of any assumptions or corrections, such as tests of normality and adjustment for multiple comparisons |
| ☐ | ☒ | A full description of the statistical parameters including central tendency (e.g. means) or other basic estimates (e.g. regression coefficient) AND variation (e.g. standard deviation) or associated estimates of uncertainty (e.g. confidence intervals) |
| ☐ | ☒ | For null hypothesis testing, the test statistic (e.g. *F*, *t*, *r*) with confidence intervals, effect sizes, degrees of freedom and *P* value noted *Give P values as exact values whenever suitable.* |
| ☒ | ☐ | For Bayesian analysis, information on the choice of priors and Markov chain Monte Carlo settings |
| ☒ | ☐ | For hierarchical and complex designs, identification of the appropriate level for tests and full reporting of outcomes |
| ☒ | ☐ | Estimates of effect sizes (e.g. Cohen's *d*, Pearson's *r*), indicating how they were calculated |

*Our web collection on statistics for biologists contains articles on many of the points above.*

## Software and code

Policy information about availability of computer code

| Data collection | NIS-Elements AR Nikon (5.21.01) Image Studio LI-COR Biosciences (5.2) Leica Application Suite X (3.7.4.23463) QuantStudio (3) ChromLab (3.3) Snapgene (6.0.2) |
|---|---|

| Data analysis | QuantStudio Design and Analysis Software (1.4.1)<br>Python (3.8.13)<br>R (4.2.2)<br>Rstudio (2022.12.0.353)<br>Fiji (2.9.0)<br>STAR aligner (2.7.9a)<br>GMAP aligner (2023-04-28)<br>HTSeq (2.0.2)<br>DESeq2 (3.17)<br>CaImAn (1.9.15)<br>Scikit-image (0.20.0)<br>FilFinder (1.7.3)<br>SciPy (1.10.1)<br>Interactive Genome Viewer (v2.16.2) |
|---|---|

For manuscripts utilizing custom algorithms or software that are central to the research but not yet described in published literature, software must be made available to editors and reviewers. We strongly encourage code deposition in a community repository (e.g. GitHub). See the Nature Portfolio guidelines for submitting code & software for further information.

## Data

Policy information about availability of data

All manuscripts must include a data availability statement. This statement should provide the following information, where applicable:
- Accession codes, unique identifiers, or web links for publicly available datasets
- A description of any restrictions on data availability
- For clinical datasets or third party data, please ensure that the statement adheres to our policy

All raw data supporting the findings of this study are deposited in the Stanford Data Repository (https://purl.stanford.edu/nj297xj2116), except for the high-throughput sequencing data generated from this study which was deposited at Gene Expression Omnibus repository (https://www.ncbi.nlm.nih.gov/geo/query/acc.cgi?acc=gse240791) with accession code GSE240791. Other public datasets analyzed in this work include: Pacbio long-read mRNA sequencing data (accession number PRJNA547800), Cell type-specific sequencing data (accession code: GSE133291, GSE115746), Neuronal activity regulated transcriptome datasets (accession number: GSE175965, GSE104802, GSE152632).

## Research involving human participants, their data, or biological material

Policy information about studies with human participants or human data. See also policy information about sex, gender (identity/presentation), and sexual orientation and race, ethnicity and racism.

| Reporting on sex and gender | This study did not use human participant/data/material |
|---|---|
| Reporting on race, ethnicity, or other socially relevant groupings | This study did not use human participant/data/material |
| Population characteristics | This study did not use human participant/data/material |
| Recruitment | This study did not use human participant/data/material |
| Ethics oversight | This study did not use human participant/data/material |

Note that full information on the approval of the study protocol must also be provided in the manuscript.

# Field-specific reporting

Please select the one below that is the best fit for your research. If you are not sure, read the appropriate sections before making your selection.

☒ Life sciences          ☐ Behavioural & social sciences          ☐ Ecological, evolutionary & environmental sciences

For a reference copy of the document with all sections, see nature.com/documents/nr-reporting-summary-flat.pdf

# Life sciences study design

All studies must disclose on these points even when the disclosure is negative.

| Sample size | No statistical methods were used to predetermine sample sizes, but sample sizes in this study are similar to those generally used in the field (ref 4, 8, 9, 18, 24, 25, 28, 36, 48). |
|---|---|
| Data exclusions | No data were excluded from the analyses. |

| Replication | All results were independently replicated at least 3 times, as described in the figure legends. |
|---|---|
| Randomization | Postnatal day 0 mouse pups were randomly chosen for making primary hippocampal cultures. Different litters of animals receive the same injection of lentivirus (control, Lphn3 KO, or Exon31 KO). In culture experiments, cover slips were assgined randomly. |
| Blinding | Data collection and analysis for TRUPATH/cAMP assay, calcium imaging, synapse puncta and rabies tracing experiments were blinded. All other experiments and analysis were not blinded, because blinding to sample group allocation is not typically relevant to biochemical and high-throughput sequencing analyses. |

# Reporting for specific materials, systems and methods

We require information from authors about some types of materials, experimental systems and methods used in many studies. Here, indicate whether each material, system or method listed is relevant to your study. If you are not sure if a list item applies to your research, read the appropriate section before selecting a response.

### Materials & experimental systems

| n/a | Involved in the study |
|---|---|
| ☐ | ☒ Antibodies |
| ☐ | ☒ Eukaryotic cell lines |
| ☒ | ☐ Palaeontology and archaeology |
| ☐ | ☒ Animals and other organisms |
| ☒ | ☐ Clinical data |
| ☒ | ☐ Dual use research of concern |
| ☒ | ☐ Plants |

### Methods

| n/a | Involved in the study |
|---|---|
| ☒ | ☐ ChIP-seq |
| ☒ | ☐ Flow cytometry |
| ☒ | ☐ MRI-based neuroimaging |

## Antibodies

| Antibodies used | Primary antibodies: anti-MAP2 chicken (1:1000, Encor #CPCA-MAP2), anti-vGluT1 guinea pig (1:1000, Millipore #AB5905), anti-Homer1 rabbit (1:1000, Millipore #ABN37),  anti-actin mouse(1:1000, Sigma #A1978), anti-Lphn3 mouse (1:1000, SCBT #sc-393576). Secondary antibodies: anti-chicken (1:1000 Alexa 405 ThermoFisher #A48260), anti-rabbit (1:1000 Alexa 488 ThermoFisher #A-11034), anti-guinea pig (1:1000 Alexa 647 ThermoFisher #A-21450), anti-mouse (1:20000, IRDye 800CW, Licor) |
|---|---|
| Validation | anti-MAP2 is widely used and sold through many vendors, see for example the results of Google Scholar search for CPCA-MAP2 (https://scholar.google.com/scholar?as_sdt=0%2C10&hl=en&inst=5746887945952177237&q=cpca-map2). anti-vGluT1 guinea pig, anti-Homer 1 rabbit and anti-actin were validated in previous paper (doi: 10.1523/JNEUROSCI.0454-20.2020). anti-Lphn3 was validated by the provider (https://www.scbt.com/p/latrophilin-3-antibody-b-6). |

## Eukaryotic cell lines

Policy information about cell lines and Sex and Gender in Research

| Cell line source(s) | HEK 293T (ATCC #CRL-11268) |
|---|---|
| Authentication | HEK 293T cell was not authenticated since it was directly purchased from ATCC (#CRL-11268) |
| Mycoplasma contamination | Cell lines were tested negative for mycoplasma contamination using PCR. |
| Commonly misidentified lines (See ICLAC register) | None of the cell lines used is listed as commonly misidentified |

## Animals and other research organisms

Policy information about studies involving animals; ARRIVE guidelines recommended for reporting animal research, and Sex and Gender in Research

| Laboratory animals | CRISPR/CAS9 knockin mice (Jax stock no: 02485) and C57BL/6 mice (Jax stock no: 000664) were obtained from Jackson Laboratory. P0 pups were used for hippocampal culture experiments; P0, P21 and P35 mice were injected with viruses for the rabies tracing experiments. |
|---|---|
| Wild animals | This study did not use wild animal. |
| Reporting on sex | Sex was not considered in this study design. |
| Field-collected samples | This study did not use field-collected samples |

Ethics oversight | All mice described in this study were maintained using established procedures according to protocols approved by the Stanford University Administrative Panel on Laboratory Animal Care.

Note that full information on the approval of the study protocol must also be provided in the manuscript.

