## [Peer Review File · Nature]

Manuscript Title: Alternative Splicing of Lathophilin-3 Controls Synapse Formation

Reviewer Comments & Author Rebuttals

Reviewer Reports on the Initial Version:

Referees' comments:

Referee #1 (Remarks to the Author):

In the submitted manuscript, Wang and colleagues address the mechanism by which synaptic adhesion GPCR lathophilin-3 modulates synapse formation. Although lathophilins are known to be central in mediating the synapse assembly and are among the best structurally-characterized GPCRs, it remained enigmatic how switching different (and often opposing) downstream signaling pathways is achieved. In this exciting study, the authors start by detailed dissecting of splice isoforms of lathophilins. It turned out that the cytosolic tails (i.e., encoded by exons 31 and 32) trigger a distinct signaling output via G(α)s or G(α)12/13 cascade. In a set of cellular assays, they determined that deletion of Exon 31 abolished the switching of lathophilin-3 signaling and disrupted the functional assembly of synapses, triggering the question about the underlying mechanism. In a surprising twist in the story, the authors identify that the cytosolic tail of lathophilin-3 encoded by Exon 31, but not by Exon 32, modulates postsynaptic condensates. Furthermore, the switching between Exons 31 and 32 was shown to be responsive to activation signals. To my knowledge, this is the first concrete example where functional input leads to differential signaling outcomes by either permitting or prohibiting phase separation. However, several aspects need to be clarified, and biochemical characterization strengthened prior to publishing in a top scientific journal.

Major comments:

(i) The CRISPR Exon 31-knockout of lathophilin-3 indicates a significant decrease in the total amount of lathophilin-3. Analysis of the submitted immunoblot image in Fig. 2b results in significantly lower signal for lanes 5&6 vs. control lanes. Thus, the splice variant with Exon31 might be important for regulating the total concentration of lathophilin-3 as well. This needs to be amended in the text and subsequent interpretations, such as synapse density and connectivity data (Fig 3) adjusted to account for the change in expression level.

(ii) Phase separation assays have several significant drawbacks—first, excessive protein concentrations. For example, GKAP, Homer-3, PSD95, and Shank-3 have a total concentration of 119 μ M, to which Lphn3 variants are added. Would the Lphn3 accumulate at the condensates of lower concentrations (i.e., a total of 40 μ M)?

(iii) As it is a full-length protein with a detergent around its prominent TMDs, it is unclear whether Lphn3 E31 forms a jammed layer around the condensates or a dynamic, fluid shell. Would there be a recovery upon photobleaching of Lphn3 E31? Similarly, how is the presence of Lphn3 E31 at the surface affecting the internal dynamics of postsynaptic condensates?

(iv) Another issue is that most proteins for assembling the postsynaptic phase are severely truncated. The authors refer to the initial paper from the Zhang lab, in which most of the proteins used are simply short fragments missing major domains responsible for signaling. Given how important this is for the message of the manuscript and the mechanism proposed, this fact should be made clear in the results section, figure 4, and the methods.

(v) Surface properties of condensates are sensitive to their constituents and their molar ratios. In

their reconstitutions, apart from the high total concentration, the authors use unusual molar ratios of the components, which are very different from the ones reported for neurons (PMID: 34168338). Is the accumulation of Lphn3 present when the protein concentrations are closer to physiological? In particular, would Tenm2 and Flrt3 be able to trigger this remarkable clustering of condensates composed of individual proteins at such a high concentration (120 μ M)? How sensitive is the clustering of condensates to ionic strength?

Minor:

(vi) a coomassie gel with the purified proteins—especially lathrophilin-3 E31 and E32, Ten2, Flrt3—needs to be shown.

(vii) The authors then claim to incubate the sample for 10 min and image only 5 μ L total volume. Given this high concentration, the methods section needs to explain how evaporation is controlled in the sample.

(viii) Extended Data Figure 7 was not discussed in the text. It is unclear what protein concentrations are used here nor to what extent these condensates can be related to the situation in the spines (see a major point (v) above).

Referee #2 (Remarks to the Author):

In this paper Sudhof and colleagues demonstrate extensive alternative splicing in the cytoplasmic domain of the Adhesion-GPCR Latrophilin 3 (Lphn3) and explore the functional significance of alternative splicing on synapse formation. They demonstrate that one alternative cytoplasmic domain of Lphn3 generated by alternative splicing selectively signals to $G_{\alpha s}$ thereby increasing cAMP levels and that this form of Lphn3 promotes synapse formation more efficiently than an alternative cytoplasmic domain that signals via an alternative G protein coupled pathway. They present three lines of evidence that this isoform promotes synapse formation including in vivo experiments using rabies virus retrograde labeling. They demonstrate that this form of Lphn3 selectively facilitates complexes with other cytoplasmic components in the postsynaptic membrane and that this activity is further stimulated by binding to ligand. Studies through KCl activation of hippocampal neurons in vitro or kainate treatment in vivo promote inclusion of exons promoting G_{α} coupling. Together, this provides an exciting model of how splicing of synaptic adhesion molecule promotes synapse formation and that this splicing is promoted by neural activity.

The data support the authors' conclusion stated in the title that splicing of Lphn-3 controls synapse formation but it does not support the contention in the second paragraph of the main text that $G_{\alpha s}$ coupling resulting from alternative splicing is essential for synapse formation in the hippocampus. There are other examples in the text where more care should be given to precisely stating the conclusions. It seems to this reviewer that this distinction is an important one. The authors' discoveries reported in this paper provide an exciting perspective on how splicing can regulate intracellular signaling pathways to modify synapse formation by neural activity. This provides yet another way cAMP may contribute to modify synapse formation in response to neural activity and has important implications for the mechanisms underlying the establishment of neural circuits and plasticity. The authors have not shown that this splicing event is essential for synapse formation but rather that splicing provides a mechanism for modulating intracellular signaling pathways to sculpt the connectome, in principle, in myriad ways.

Specific (and minor) comments:

1. A more quantitative representation of the efficacy of different spliced forms of Lphn3 to activate adenylate cyclase should replace the heat map.
2. Figure 2h: Change the Y axis to more accurately represent the decrease. As it is now presented it appears as a much more dramatic change (i.e. start at "0" not "5").

3. The statement "Both the complete loss of Lph3 and exon 31-specific deletion produced a significant decrease in synaptic density" is correct but misleading. This difference appears to be a little more than 20%. It looks more like 50% in the graphical representation in Figure 3b, as the Y axis has been truncated. Please adjust Y axis accordingly.
4. I was not able to find the kainate treatment protocol in the methods.
5. I am not an expert in phase separated assembly. Presumably other reviewers can more critically evaluate these data.

Referee #3 (Remarks to the Author):

A few years ago, Südhof's team showed that the postsynaptic adhesion molecule Latrophilin-3 regulates Schaffer-collateral synapse formation in the hippocampus. However, the intracellular signaling pathway underlying Latrophilin-3 synapse formation was unknown. In this study, following previous findings from the lab, Wang et al. provide evidence that specific G proteins are downstream on Latrophilin-3 driving synapse formation. Also, a preliminary analysis suggests that neuronal activity might promote the specific alternative splicing. Unfortunately, I have several concerns that preclude the publication of this manuscript in Nature. As it is, I do not see any advance in the knowledge to justify its publication in Nature. This study shows the next intracellular protein interacting with Latrophilin-3 for synapse formation. Also, I have some other concerns that the authors need to address.

Figure 1

-Latrophilin-3 drives Schaffer collateral synapse formation, and then different splicing isoforms interact with G-couple proteins. Is this particular to Schaffer Collaterals, or within Schaffer collaterals, are there different splicing isoforms regulated depending on context? The authors conducted a nice analysis of the cell-type specificity expression but did not use this data.

-Fig.1d. The authors claim: "The most abundant Lphn3 splice variant in the hippocampus (E24+/E30b-/E31+/E32-) preferentially couples to Gas and less strongly to Ga12/13". However, from the data in the TRUPATH plots, it is difficult to understand why the authors say that the couple is less strongly to Ga12/13. Gas Ga12 seem to have identical G-protein coupling strength (same intensity).

-The authors claim that Lphn3 alternative splicing is cell-specific (see lines 94-104; Extended data Fig.1 and Fig.3). However, it is unclear why they focus only on E31, which is more expressed in inhibitory neurons rather than, for example, E32 which has a strong preference for excitatory neurons.

-Extended Figure 3 shows the alternative splicing of Lphn3 in different brain regions and developmental time points. Sometimes, the same isoform presents different developmental profiles depending on the brain region analyzed. This figure would benefit from some quantifications. For example, some blots have distinct light exposure contrast within the blots.

-Can the specific G-protein coupling function be mediated through a synergistic role of the Exon 24 and Exon31?

Figure 2

-This method figure does not describe a significant analytical/technical discovery or a main finding but confirms that the deletions are well done. It can be transferred to supplementary information.

-Fig.2c. Why is there an increase of E32 mRNA levels in the E31 KO? Could this be a compensatory

mechanism?

-Fig.2e-h. The image in panel e needs to be clarified. What is the true Calcium transient signal when it fills the entire soma or the "dot" pattern? A better image or a better explanation may need it here.

-Why do Latrophilin-3 KO and E31 KO show the same firing rate phenotype? Does it mean that the ability of E31 to recruit postsynaptic proteins is sufficient to drive the phenotype or that the method does not have the resolution to distinguish between the complete lack of Latrophilin-3 signalling and the absence of one splicing isoform?

Figure 3

-Fig.3a. Why do the authors use vGluT1 and Homer to label the synapses? Bassoon is often more used with Homer since they are both at the synaptic cleft. Is there any reason why PSD95 is not used? This is important considering that in Figure 4, the authors describe the possibility of interaction of the PDZ E31 with PSD95.

-In Figure 3a, the overlay colocalization of vGluT1 and Homer are often outside of the dendrite "floating" puncta. What are the criteria for colocalization? Do they quantify as positive even if it is not on the top of the dendrite?

-Rabies show connectivity between cells, which in this case, could indicate a decrease in synaptic connectivity due to a reduction in density or strength. To confirm the loss of synapses in vivo, a better approach will be to quantify the density of synapses in vivo.

Figure 3d. Why are there more connections from the dentate gyrus in the E31 KO compared to Lphn3 KO and control?

Figure 4

-Fig. 4b. It is difficult to distinguish the presence of Lphn3-Exon32 in the supernatant.

-Fig. 4h. The authors conclude in line 216: "These data suggest that alternative splicing of Lphn3 at the C-terminus determines its ability to recruit postsynaptic scaffold proteins, presumably via interactions between the PDZ-binding motif in Exon 31 of Lphn3 and the PDZ domain in Shank3 or/and PSD95". However, in Figure 4d, Shank3 and PSD95 show a similar pattern in the present E31 or E32. It seems that this E31 isoform is part of the postsynaptic density complex rather than involved in any type of recruitment. If the authors want to go further in their statement, they need to add more controls to these experiments. For example, "a non-binding" partner to Lphn3, also Lphn3 32, that seems to be doing nothing (Figure 4d).

Figure 5

This is the weaker figure of the paper, and the link with activity is in the title of this manuscript. The authors use previous data from Greenberg's lab where the activity-dependent genes are screened using depolarization-induced stimulation in cultured neurons or Kainate-induced hippocampus activation in vivo. Both approaches are rather extremes and not physiological. If they want to show an activity-dependency of splicing isoforms, they must corroborate the in-silico analysis with at least some more physiological approaches in vivo (e.g. DREADDs). Also, if they want to link this to a synaptic change, they need to show that the change is prevented in the KO experiments.

Minor

-Why do the authors use Retina databases since the manuscript is focused on the hippocampus?

- Would it be possible to confirm the cell-type specific expression pattern of the Lphn3-Exon 31 and Lphn3-Exon 32?

Author Rebuttals to Initial Comments:

Authors' Response to the Reviewers' Comments for Shuai Wang et al. "..."

We thank the reviewers for their careful evaluation of our manuscript, their enthusiasm, and their insightful suggestions. In response to the reviewers' comments, we have performed extensive additional experiments that strengthen the conclusions of our paper as described in detail below, and that also provide further controls and broaden our data's implications. Specifically, the following additional experiments were included in the revised paper:

1. Further confirmation of the cell type-specific alternative splicing of Exon31 and Exon32 of Lphn3 (*Adgrl3*) using two additional independent datasets in which neurons are classified into distinct excitatory and inhibitory subtypes: (1) Allen Brain Institute single-cell RNAseq data (PMID: 30382198) from the mouse primary visual cortex and the anterior lateral motor cortex (new Extended Data Figure 1d) and (2) single-cell RNAseq data from the mouse mPFC (PMID: 33177708) (new Extended Data Figure 1e). Cell types were classified by specific marker genes, with the results confirming the analyses presented in the original submission demonstrating the cell type-specific alternative splicing of Lphn3 (in response to comment #19 of referee #3; new Extended Data Figure 10b, 10c).
2. Further analyses of the activity-dependent alternative splicing of Lphn3 (*Adgrl3*) Exon31 and Exon32 in two additional independent datasets: (1) hippocampal neuron cultures treated with picrotoxin from the Futai lab (PMID: 29898393), and (2) in vivo natively activated medial prefrontal cortex (mPFC) neurons from the Quake and Südhof labs (PMID: 33177708) (in response to comment #17 of referee #3; new Figure 5b, 5d-5f, new Extended Data Figure 10b-10d).
3. Quantification of the developmental region-specific alternative splicing of Lphn3 using qRT-PCR in 3 independent batches of experiments documenting the regulated alternative splicing of Lphn3 (in response to comment #5 of referee #3; new Extended Data Figure 3b).
4. Additional experimental analyses of the G-protein coupling of Lphn3 splice variants using the TRUPATH platform with a series of Lphn3 concentrations (in response to comment #3 of referee #3; new Extended Data Figure 4a, 4c).
5. Quantification of total Lphn3 protein levels using quantitative immunoblotting from 6 independent culture experiments to firmly establish that the acute CRISPR deletion of Lphn3 effectively abolishes Lphn3 protein expression whereas the CRISPR-mediated exclusion of the alternatively spliced Exon31 does not change Lphn3 protein levels (in response to comment #1 of referee #1; new Figure 2b, Supplementary Figure 1).
6. An additional control experiment for the pseudorabies virus tracing results demonstrating that the weak GFP signal in the dentate gyrus area was due to Cas9 (in response to comment #14 of referee #3; new Extended Data Figure 5e).
7. Further experiments demonstrating that with lower concentrations of postsynaptic scaffold proteins (10 μ M GKAP, 10 μ M Homer, 10 μ M PSD95, and 10 μ M Shank3) phase separation is maintained as effectively as with the original higher

concentrations, supporting the specificity of the observed Lphn3-anchored phase separation (in response to comment #2 of referee #1; new Figure 4).

8. A new phase-separation experiment showing that the PDZ-domain binding motif (PBM) of Lphn3 that binds to Shanks is essential for Lphn3 enrichment on the phase separated postsynaptic scaffolds and for Tenm2/Flrt3-induced clustering of the condensates. This new experiment was carried out with a repetition of the original experiment but at lower scaffold protein concentrations (see new experiment #6 above) and is now part of Figure 4 (in response to comment #5 of referee #1).
9. A new analysis of the functional significance of the Shank-interacting C-terminal PBM of Lphn3 by developing a CRISPR-based strategy that enables selective deletion of the PBM sequence in the endogenous Lphn3 gene. Analyses of Lphn3 mRNAs and protein confirms that this strategy made it possible to delete the PBM without changing Lphn3 alternative splicing or total protein level. Quantification of synapses revealed that mutant Lphn3 lacking the PBM did not support synapse formation. These experiments thus establish the specificity of Exon31 that contains the PBM in mediating synapse formation and provide a functional correlate for the new phase separation experiments (see new experiment #7 above), demonstrating that the PBM is essential for the Lphn3-anchored phase separation of postsynaptic scaffolds (in response to comment #10 and #16 of referee #3; new Extended Data Figure 9).
10. An additional new phase-separation experiment using Fluorescence Recovery After Photobleaching (FRAP) to study the dynamic properties of Lphn3 and of scaffold proteins. The FRAP experiment documents that Lphn3 forms a dynamic shell on the post-synaptic condensates and that the condensates are further clustered by presynaptic ligands Tenm2/Flrt3 without perturbing the dynamics of scaffold proteins (in response to comment #3 of referee #1; new Extended Data Figure 8f-8l).
11. A new control experiments for the phase separation results demonstrating that phase-separation is sensitive to high ionic strength (in response to comment #5 of referee #1; new Extended Data Figure 8e).
12. Further new control experiments for the phase separation results documenting the importance of the scaffold protein stoichiometry for phase separation (in response to comment #5 of referee #1; new Extended Data Figure 8d).
13. Addition of size exclusion chromatography profiles and of images of Coomassie-stained gels of the peak fractions obtained during the last step of protein purification for all 9 proteins used in the phase separation experiments (in response to comment #6 of referee #1; new Extended Data Figure 6a-6i).

We hope that with these additional data and the changes and corrections instituted in our paper, the paper may be deemed acceptable for publication. In the following, we will cite the full reviewers' comments in regular black typeface and provide our response in blue typeface.

Referee #1

In the submitted manuscript, Wang and colleagues address the mechanism by which synaptic adhesion GPCR lathrophilin-3 modulates synapse formation. Although lathrophilins are known to be central in mediating the synapse assembly and are among the best structurally-characterized GPCRs, it remained enigmatic how switching different (and often opposing) downstream signaling pathways is achieved. In this exciting study, the authors start by detailed dissecting of splice isoforms of lathrophilins. It turned out that the cytosolic tails (i.e., encoded by exons 31 and 32) trigger a distinct signaling output via G(alpha)s or G(alpha)12/13 cascade. In a set of cellular assays, they determined that deletion of Exon 31 abolished the switching of lathrophilin-3 signaling and disrupted the functional assembly of synapses, triggering the question about the underlying mechanism. In a surprising twist in the story, the authors identify that the cytosolic tail of lathrophilin-3 encoded by Exon 31, but not by Exon 32, modulates postsynaptic condensates. Furthermore, the switching between Exons 31 and 32 was shown to be responsive to activation signals. To my knowledge, this is the first concrete example where functional input leads to differential signaling outcomes by either permitting or prohibiting phase separation. However, several aspects need to be clarified, and biochemical characterization strengthened prior to publishing in a top scientific journal.

We greatly appreciate the reviewer's excitement about our finding that activity-regulated alternative splicing of Lphn3 controls synapse formation via differential G protein signaling and phase separation, and especially thank the reviewer for providing insightful suggestions to our phase separation experiments. In addition to addressing the reviewer's comments as described in detail below, we have expanded the phase separation experiments to increase their significance and impact by analyzing the role of the PDZ-binding motif (PBM) of Lphn3 that binds to Shanks. Specifically, we generated and purified a full-length control Lphn3 protein that lacks the last 3 amino acids of Lphn3 (which constitute the PBM; the mutant is referred to as ' Δ PBM'). We then performed a new set of phase separation experiments with full-length wild-type and mutant Lphn3 proteins. These demonstrate that, whereas wild-type Lphn3 is robustly recruited to the surface of the phase-separated condensates, mutant Lphn3 is only very weakly enriched on the surface of the phase-separated scaffold protein condensates. Moreover, only wild-type but not mutant Lphn3 triggered clustering of condensates upon addition of the Lphn3 ligands Tenm2 and Flrt3 (new Figure 4). Thus the C-terminal PBM of Lphn3 is essential for recruiting post-synaptic scaffold protein networks. Moreover, in additional new experiments we now show that genetically deleting the PBM from the endogenous Lphn3 gene using acute CRISPR manipulations results in a decrease in synapse density (new Extended Data Figure 9), suggesting that the recruitment of post-synaptic scaffold protein networks via phase separation is crucial for synapse formation.

Major comments:

(i) The CRISPR Exon 31-knockout of lathrophilin-3 indicates a significant decrease in the total amount of lathrophilin-3. Analysis of the submitted immunoblot image in Fig. 2b results in significantly lower signal for lanes 5&6 vs. control lanes. Thus, the splice variant with Exon31 might be important for regulating the total concentration of

lathrophilin-3 as well. This needs to be amended in the text and subsequent interpretations, such as synapse density and connectivity data (Fig 3) adjusted to account for the change in expression level.

To address this important point, we have now quantified the immunoblots from 6 independent cultures (raw blots in Supplementary Figure 1). The results demonstrate that the E31-specific KO does not alter total Lphn3 protein levels (new Figure 2b, new Extended Data Figure 9d). Moreover, our transcriptome-level RNAseq experiment (Figure 2d) unbiasedly showed that the Lphn3 KO, but not the Exon31-specific KO, suppressed the Lphn3 mRNA. These data strongly support the specificity of our genetic manipulations, consistent with the finding that the Exon31-specific KO led to an increase in the levels of Exon32-containing Lphn3 mRNAs without perturbing the total mRNA or protein level of Lphn3. We have replaced the old Figure 2b immunoblot with a more representative image as suggested and added the new quantifications.

(ii) Phase separation assays have several significant drawbacks—first, excessive protein concentrations. For example, GKAP, Homer-3, PSD95, and Shank-3 have a total concentration of 119 μM , to which Lphn3 variants are added. Would the Lphr3 accumulate at the condensates of lower concentrations (i.e., a total of 40 μM)?

We agree and have followed the reviewer's suggestion. We repeated all phase separation assays in the previous Figure 4 by using 10 μM of GKAP, Homer-3, PSD95, and Shank-3 each (total scaffold concentration 40 μM as suggested) with the same concentrations of Lphn3, Tenm2, Flrt3 as before. The lower concentration dataset resulted in the same conclusions as the higher concentration dataset. A few differences, however, were observed. (1) Lphn3-E31 co-sedimented with scaffold proteins at a slightly lower efficiency: 7.5% now (new Figure 4c) vs. 11% before (old Figure 4c, now shown in new Extended Data Figure 7b); (2) the phase-separated liquid condensates were overall smaller than before (new Extended Data Figure 8c; new Figure 4d-4f and new Extended Data Figure 8a, 8b vs. new Extended Data Figure 7c-7f); (3) the formation of the scaffold protein condensates was kinetically slower, especially when adding the Lphn3 ligands Tenm2 and Flrt3 to induce clusters of condensate which took ~ 20 min now vs. ~ 10 min before. These minor differences are expected since less scaffold proteins have slower on-rates for forming complexes (for example, bimolecular complex formation $A+B \rightleftharpoons A \bullet B$ has an on-rate of $k_{on}[A][B]$, which is proportional to the concentration of each reactant). Due to the lower amount of individual scaffold proteins before mixing, at steady state there are less scaffold proteins complexes thus smaller condensates. Because Lphn3-E31 depends on scaffold proteins to sediment (new Extended Figure 6j), less scaffold condensates would result in less Lphn3-E31 co-sedimented with the scaffold complexes to the pellet fraction when the total amount of Lphn3-E31 remains the same.

(iii) As it is a full-length protein with a detergent around its prominent TMDs, it is unclear whether Lphn3 E31 forms a jammed layer around the condensates or a dynamic, fluid shell. Would there be a recovery upon photobleaching of Lphn3 E31? Similarly, how is the presence of Lphn3 E31 at the surface affecting the internal dynamics of postsynaptic condensates?

We agree. Analyzing the dynamics of Lphn3 in phase separation using Fluorescence Recovery After Photobleaching (FRAP) experiments is an excellent suggestion that we have followed in new experiments. We now performed FRAP experiments using 10 μ M individual scaffold proteins, 6 μ M Lphn3 E31, 10 μ M Tenm2, and 10 μ M Flrt3 (new Extended Data Figure 8f-8l). In the FRAP experiments, we photo-bleached a small area and recorded the fluorescence recovery for all channels for 6 min post-bleaching. We found that Lphn3-E31 and PSD95 recovered slightly faster than Homer3 and Shank3. We reason that the slower Homer3 and Shank3 recovery rate reflects a stronger interaction within the condensates because: (1) the known interaction network (Figure 4a) shows that Shank3 and Homer3 have multiple domains to enable multivalent interactions, whereas PSD95 and Lphn3 have only one domain participating into this network; (2) the phase separation of scaffold proteins is highly dependent on Shank3 and Homer3 and less dependent on PSD95 (new Extended Data Figure 7h; see also the pioneering work of Mingjie Zhang's laboratory PMID: 30078712), but naturally independent of Lphn3 (Figure 4d first two rows). These results support the conclusion that Lphn3-E31 forms a dynamic fluid shell on the scaffold protein condensates. Consistent with the fluidic nature of Lphn3-E31, the addition of Lphn3-E31, alone or in the presence of Tenm2/Flrt3 that induced clusters of condensates, does not affect the kinetics of scaffold proteins. Therefore, the function of Lphn3-E31 is not to perturb the kinetic stability of scaffold proteins. Instead, it forms a fluidic shell on the scaffold proteins condensates, by which priming the scaffold protein droplets to be clustered by presynaptic ligands Tenm2 and Flrt3. We have discussed these important results in the main text.

(iv) Another issue is that most proteins for assembling the postsynaptic phase are severely truncated. The authors refer to the initial paper from the Zhang lab, in which most of the proteins used are simply short fragments missing major domains responsible for signaling. Given how important this is for the message of the manuscript and the mechanism proposed, this fact should be made clear in the results section, figure 4, and the methods.

We agree with the reviewer that this feature of phase separation experiments needs to be discussed. Note, however, that our constructs of PSD95 and Homer3 are full-length proteins. The reason of using truncated Shank3 and GKAP is to obtain soluble proteins, which serves as a clean background for the phase separation experiment. Only when individual scaffold proteins do not aggregate into phase separated structures (new Extended Data Figure 6j) can we study the combined effect of multiple scaffolds in vitro. In fact, full-length recombinant Shank3 potently forms large sheets composed of helical fibers stacked side by side as revealed by negative stain electron microscopy (PMID: 16439662). We have added a discussion of these points in the main text, Figure 4 legend, and method.

(v) Surface properties of condensates are sensitive to their constituents and their molar ratios. In their reconstitutions, apart from the high total concentration, the authors use unusual molar ratios of the components, which are very different from the ones reported for neurons (PMID: 34168338). Is the accumulation of Lphn3 present when the protein concentrations are closer to physiological?

We agree. To address this point – again a very important issue – we have looked at the data on the stoichiometry of scaffold proteins in vivo in the suggested reference (PMID: 34168338, Supplementary Table 2). These data show that the copy number of the scaffold proteins in the spines are as followed:

Protein	Spine copy number	Molar ratio
DLGAP1 (GKAP)	5.3 ± 2.3	0.03125
Homer3	169.7 ± 41.2	1
PSD95	525.2 ± 76.7	3.09
Shank3	124.4 ± 20.1	0.73

When we analyzed phase separation with scaffold proteins present in the in vivo stoichiometric (molar ratio GKAP:Homer3:PSD95:Shank3=0.03125:1:3.09:0.73) at a total concentration of 40 μM we still observed phase-separated protein condensates, albeit of a smaller size, and we still detected efficient enrichment of Lphn3 (new Extended Data Figure 8d). Because GKAP is critical for assembling the scaffold condensate (new Extended Data Figure 7h), the low level of GKAP significantly decreased the droplet size as expected. Importantly, as mentioned in (iv), the in vivo synapse has full-length Shank3 which is more prone to form large assemblies than the Shank3 construct used in this work. Also synapses contain additional scaffold proteins (such as Homer1, Homer2, Shank1, Shank2) which can participate into the network. Finally, the scaffold proteins are probably differentially distributed in spines and their total vs. local concentrations may differ. We speculate that synapses may have evolved an optimal stoichiometry for the complete repertoire of scaffold proteins. Our in vitro assay is a reductionist approach which reveals the minimal protein/domains sufficient to drive the phase separation but can only approximate an in vivo condition.

In particular, would Tenm2 and Flrt3 be able to trigger this remarkable clustering of condensates composed of individual proteins at such a high concentration (120 μM)?

We also find the clustering effect of Tenm2 and Flrt3 remarkable, and agree that it is important to have negative controls for these clustering effects. Following up on the reviewer's comment, we found that individual scaffold proteins at high concentrations (using 120 μM Homer3, 120 μM PSD95, or 120 μM Shank3) do not by themselves phase separate with Tenm2/Flrt3 (merged images of all channels shown below):

(Note: leftmost positive control used 38 μM GKAP, 30 μM Homer3, 19 μM PSD95, 32 μM Shank3)

Basically, individual scaffold protein with Tenm2/Flrt3 produced no signal as viewed by confocal microscopy because they are freely diffusing under these conditions. We have provided two more stringent controls by the analysis of Lphn3-E32 and mutant Lphn3-E31 with a deletion of the PDZ-binding motif (ΔPBM). Neither of these proteins showed clustering upon addition of Tenm2/Flrt3 (new Figure 4f, 4g). These data demonstrate that the clustering induced by Tenm2/Flrt3 is highly dependent on the PBM of Lphn3 interaction with phase-separated protein scaffold (presumably mediated by Shank3).

How sensitive is the clustering of condensates to ionic strength?

This is an excellent question that we have tested in the revised paper. We show that the sensitivity of Tenm2/Flrt3-induced Lphn3 E31 clustered scaffold condensates to increasing salt concentration (new Extended Figure 8e). Thus increasing ionic strength beyond physiological range dissipates the phase separation.

Minor:

(vi) a coomassie gel with the purified proteins—especially lathrophilin-3 E31 and E32, Ten2, Flrt3—needs to be shown.

Agreed. We show the quality of our purified proteins in three parts: (1) We demonstrate the protein elution patterns during the last step of protein purification on size-exclusion chromatography for all proteins; (2) we show Coomassie gels of the peak fractions from the size-exclusion chromatography, which together with the elution pattern document the purity, homogeneity and the in-solution size of the purified proteins (new Extended Data Figure 6a-6i); and (3) we include sedimentation assays for individual protein with supernatants and pellets analyzed by Coomassie-stained SDS gels (New Extended Data Figures 6j). These results document the solubility of the purified proteins at high concentrations (80 μM GKAP, 100 μM PSD95, 80 μM Homer3, 200 μM Shank3, 40 μM Lphn3 E31 ΔPBM , 40 μM Lphn3 E32, 40 μM Lphn3 E31).

(vii) The authors then claim to incubate the sample for 10 min and image only 5 μL total volume. Given this high concentration, the methods section needs to explain how evaporation is controlled in the sample.

Again, we agree that this is an important technical detail. We used channeled slides (ibidi #80666) that are designed for imaging small sample volumes. The slides contain a channel that holds 1.7 μl of sample and the channel connects two reservoirs. After protein samples were mixed, the mix was pipetted into the reservoir (reservoir and channel needs to be pre-wetted by buffer) which send the samples into the channel. The slide needs to be sealed by its cover which minimizes evaporation. When used at room temperature, we found the sample in the channel does not have significant evaporation within 40 min. After that, evaporation began to drive the movement of droplets which is not ideal for imaging experiments. We have added the above description to the methods section of the paper.

(viii) Extended Data Figure 7 was not discussed in the text. It is unclear what protein concentrations are used here nor to what extent these condensates can be related to the situation in the spines (see a major point (v) above).

We apologize for the omission that we have remedied in the revised manuscript. Old Extended Data Figure 7 (new Extended Data Figure 6j and 7h) aims to show two results. (6j) Documentation that individual scaffold proteins do not sediment. The concentrations of proteins used were: 80 μ M GKAP, 100 μ M PSD95, 80 μ M Homer3, 200 μ M Shank3, 40 μ M Lphn3 E31 Δ PBM, 40 μ M Lphn3 E32, 40 μ M Lphn3 E31. (7h) Description of the degree of phase separation when an individual scaffold protein is removed. Concentration of proteins used here were: 38 μ M GKAP, 30 μ M Homer3, 19 μ M PSD95, 32 μ M Shank3, 6 μ M Lphn3 E31.

Referee #2:

In this paper Sudhof and colleagues demonstrate extensive alternative splicing in the cytoplasmic domain of the Adhesion-PCR Latrophilin 3 (Lphn3) and explore the functional significance of alternative splicing on synapse formation. They demonstrate that one alternative cytoplasmic domain of Lphn3 generated by alternative splicing selectively signals to $G\alpha_s$ thereby increasing cAMP levels and that this form of Lphn3 promotes synapse formation more efficiently than an alternative cytoplasmic domain that signals via an alternative G protein coupled pathway. They present three lines of evidence that this isoform promotes synapse formation including in vivo experiments using rabies virus retrograde labeling. They demonstrate that this form of Lphn3 selectively facilitates complexes with other cytoplasmic components in the postsynaptic membrane and that this activity is further stimulated by binding to ligand. Studies through KCl activation of hippocampal neurons in vitro or kainate treatment in vivo promote inclusion of exons promoting $G\alpha$ coupling. Together, this provides an exciting model of how splicing of synaptic adhesion molecule promotes synapse formation and that this splicing is promoted by neural activity.

The data support the authors' conclusion stated in the title that splicing of Lphn-3 controls synapse formation but it does not support the contention in the second paragraph of the main text that $G\alpha_s$ coupling resulting from alternative splicing is essential for synapse formation in the hippocampus. There are other examples in the text where more care should be given to precisely stating the conclusions. It seems to this reviewer that this distinction is an important one. The authors discoveries reported in this paper provide an exciting perspective on how splicing can regulate intracellular signaling pathways to modify synapse formation by neural activity. This provides yet another way cAMP may contribute to modify synapse formation in response to neural activity and has important implications for the mechanisms underlying the establishment of neural circuits and plasticity. The authors have not shown that this splicing event is essential for synapse formation but rather that splicing provides a mechanism for modulating intracellular signaling pathways to sculpt the connectome, in principle, in myriad ways.

We thank the reviewer for her/his enthusiasm, helpful comments, and careful attention to the complexity of GαS signaling. Indeed, our current study did not specifically perturb synaptic GαS signaling to study the functional consequences in synapse formation. Previous work from our lab (PMID: 33646123) inserted T4 lysozyme into the Exon24 of Lphn3 (isoform: E24+/E30b-/E31+/E32-), which abolished cAMP signaling, and decreased synapse numbers and CA3-CA1 connectivity. These previous data suggest GαS signaling is functionally important for synapse formation. Our current study builds on these previous data to show that activity-dependent alternative splicing of Lphn3 regulates its ability to mediate GαS signaling and to recruit phase-separated postsynaptic protein scaffolds, providing a novel mechanistic model for synapse formation. In earlier studies such as the one cited above we only used a single splice variant since we were not aware of the alternative splicing of Lphn3 that we now describe, but luckily the variant we used turned out to be the active one. It is for us remarkable that switching Lphn3 alternative splicing from Exon31 to Exon32 abolishes coupling to GαS and recruitment of phase-separated protein scaffolds and simultaneously suppresses synaptic connectivity, consistent with the notion that alternative splicing also regulates synapse formation by a dual parallel mechanism: GαS signaling and recruitment of phase-separated postsynaptic protein scaffolds.

Specific (and minor) comments:

1. A more quantitative representation of the efficacy of different spliced forms of Lphn3 to activate adenylate cyclase should replace the heat map.

The quantitative results of the cAMP measurements from 3 independent batches were included in the Extended Data 4b, 4c (new Extended Data Figure 5a,5b), but we would be happy to move them to the main figures if this improves the manuscript. We felt that a simple heatmap as used in the main Figure 1d might be easier to appreciate by a general audience since the focus is on the main message of G-protein coupling for each splice isoform, but we will follow whatever advice we are given.

2. Figure 2h: Change the Y axis to more accurately represent the decrease. As it is now presented it appears as a much more dramatic change (i.e. start at “0” not “5”).

Agreed – we have changed the Y axis of Figure 2h as suggested.

3. The statement “Both the complete loss of Lph3 and exon 31-specific deletion produced a significant decrease in synaptic density” is correct but misleading. This difference appears to be a little more than 20%. It looks more like 50% in the graphical representation in Figure 3b, as the Y axis has been truncated. Please adjust Y axis accordingly.

We again agree and have changed the Y axis of Figure 3b as suggested.

4. I was not able to find the kainate treatment protocol in the methods.

We agree that this is an important detail but we have not provided this information since we used public data reported by others, as described in the manuscript. The original paper

from the Greenberg lab (PMID: 36792830) documented this protocol in the “Experimental Procedures-Collection time points”.

5. I am not an expert in phase separated assembly. Presumably other reviewers can more critically evaluate these data.

Thanks. This was extensively discussed by reviewer #1.

Referee #3:

A few years ago, Südhof’s team showed that the postsynaptic adhesion molecule Latrophilin-3 regulates Schaffer-collateral synapse formation in the hippocampus. However, the intracellular signaling pathway underlying Latrophilin-3 synapse formation was unknown. In this study, following previous findings from the lab, Wang et al. provide evidence that specific G proteins are downstream on Latrophilin-3 driving synapse formation. Also, a preliminary analysis suggests that neuronal activity might promote the specific alternative splicing. Unfortunately, I have several concerns that preclude the publication of this manuscript in Nature. As it is, I do not see any advance in the knowledge to justify its publication in Nature. This study shows the next intracellular protein interacting with Latrophilin-3 for synapse formation. Also, I have some other concerns that the authors need to address.

The reviewer’s main criticism seems to be that our “*study shows the next intracellular protein interacting with Latrophilin-3 for synapse formation*”. We would like to note, however, that our study does not report a new protein that interacts with Lphn3 for synapse formation. We agree that if this was the main point of our paper, our study would indeed not be justified for publication in *Nature*, as the reviewer states.

Instead, the key advance we report in our paper is that active-dependent alternative splicing of Lphn3 regulates its coupling to specific G proteins and its recruitment of phase-separated postsynaptic protein scaffolds, leading to a new unified mechanism for synapse formation. The new observations described in our study build on a large number of previous results to achieve this advance. Scaffold proteins are well-known to be important for synapse organization and to form phase-separated condensates, and we now suggest a mechanism that recruits these to actual trans-synaptic junctions. Many synaptic adhesion molecules are known to localize pre- or post-synaptically but no coherent concept existed on how they build synapses. One key adhesion molecule, Lphn3, is an adhesion-GPCR but it was unclear why a synaptic adhesion molecule is also a GPCR. Our work provides a new conceptual framework that addresses these questions by suggesting a simple mechanism: Lphn3 recruits a post-synaptic scaffold protein network proximal to the membrane and simultaneously activates G-protein signaling. The axon terminal contains the pre-synaptic ligands Tenm2/Flrt3. When axons are close enough to the Lphn3-containing dendrite, presynaptic Tenm2/Flrt3 interact with post-synaptic Lphn3, and because Tenm2 and Flrt3 are dimers, such adhesion causes higher order oligomerizations of Lphn3-bound post-synaptic scaffold networks, thereby triggering the accumulation and stabilization of the well-known electron-dense postsynaptic scaffold network at the synaptic cleft. Meanwhile, Lphn3 triggers GαS signaling which may be amplified within the above condensates. This proposed mechanism is regulated by the

activity-dependent alternative splicing of Lphn3, especially at its C-terminus by two mutually exclusive exons: exon31 and exon32. When neurons are activated, alternative splicing of exon31 is favored, leading to more synapse formation. Among others, such activity-dependent alternative splicing offers an explanation for how neurons “fire together, wire together”. Together, these findings suggest a new paradigm of how Lphn3 -and possibly other synaptic adhesion molecules- promote synapse formation.

Figure 1

(1) -Latrophilin-3 drives Schaffer collateral synapse formation, and then different splicing isoforms interact with G-couple proteins. Is this particular to Schaffer Collaterals, or within Schaffer collaterals, are there different splicing isoforms regulated depending on context?

We agree this is an interesting question but have not addressed this question in the present manuscript because our study focuses on fundamental mechanisms involved in synapse formation. We have not yet examined different types of synapses beyond looking at Schaffer collateral synapses. However, it seems likely to us that the same mechanisms apply to other synapses given the widespread expression of Lphn3 in brain, which would be a fertile subject of future studies especially in the mPFC given the role of Lphn3 in attention deficit-hyperactivity disorder (ADHD). As regards Schaffer-collateral synapses, we show that different Lphn3 splice variants are expressed postsynaptically in CA1 pyramidal neurons (new Extended Data Figure 1c, CamK+ Hippocampus sample), but due to the lack of splice-variant specific antibodies we cannot tell whether these variants are co-localized at the same synapses. Our current and previous data only show that Lphn3 is generally enriched in Schaffer-collateral synapses and that only the Lphn3 splice variant (Lphn3-E31) that mediates G α S signaling and phase-separation with scaffold proteins can support synapse formation.

(2) The authors conducted a nice analysis of the cell-type specificity expression but did not use this data.

We agree with the reviewer that the cell type-specific expression of Lphn3 is fascinating, again consistent with a broad function of Lphn3 in multiple synapses as indicated by the reviewer’s previous comment. This finding actually originally motivated us to study the function of alternative splicing of Lphn3 and its regulation by activity! The aim of the current study was to unravel the biological significance and molecular mechanism of alternatively spliced Lphn3 variants, not to test the role of such variants in different types of neurons.

(3) -Fig.1d. The authors claim: “The most abundant Lphn3 splice variant in the hippocampus (E24+/E30b-/E31+/E32-) preferentially couples to G α s and less strongly to G α 12/13”. However, from the data in the TRUPATH plots, it is difficult to understand why the authors say that the couple is less strongly to G α 12/13. G α s G α 12 seem to have identical G-protein coupling strength (same intensity).

We agree that this point needs to be clarified. Our conclusion that (E24+/E30b-/E31+/E32-) preferentially couples to $G_{\alpha s}$ and less strongly to $G_{\alpha 12/13}$ is based on the BRET signal of Lphn3 (E24+/E30b-/E31+/E32-) in Gs vs. G12 and G13 (old Extended Data Figure 4a, new Extended Data Figure 4b), using 300 ng Lphn3 for the transfection. In fact, we titrated the amount of Lphn3 DNA (0, 25, 50, 100, 200, 300 ng) and assayed G protein coupling for all 6 isoforms (new Extended Data Figure 4a). At DNA concentrations of 200 and 300 ng, the cells express saturating amount of Lphn3 to bind G proteins. The TRUPATH data from both 200 and 300 ng concentrations (new Extended Data Figure 4c) supported our conclusion that the most abundant Lphn3 splice variant in the hippocampus (E24+/E30b-/E31+/E32-) preferentially couples to $G_{\alpha s}$ and less strongly to $G_{\alpha 12/13}$, although the reviewer is correct that coupling to $G_{\alpha 12/13}$ is also observed and could be physiologically relevant. In the new Figure 1d, we chose to use the data from 200 ng Lphn3, to be more consistent with the concentration of Lphn3 (230 ng) used in our cAMP assays.

(4) -The authors claim that Lphn3 alternative splicing is cell-specific (see lines 94-104; Extended data Fig. 1 and Fig. 3). However, it is unclear why they focus only on E31, which is more expressed in inhibitory neurons rather than, for example, E32 which has a strong preference for excitatory neurons.

Since Exon31 and Exon32 are mutually exclusive and all Lphn3 mRNAs contain either Exon31 or Exon32, we do not actually focus on one or the other. Our functional experiments, however, show that Exon32 variants of Lphn3 are inactive at the synapse (but could possibly be doing something else). As a result, we do focus on the mechanism of Exon31 since we don't have a functional interpretation for Exon32.

(5) -Extended Figure 3 shows the alternative splicing of Lphn3 in different brain regions and developmental time points. Sometimes, the same isoform presents different developmental profiles depending on the brain region analyzed. This figure would benefit from some quantifications. For example, some blots have distinct light exposure contrast within the blots.

Agreed. We have now quantified the data as suggested (new Extended Data Figure 3b).

(6) -Can the specific G-protein coupling function be mediated through a synergistic role of the Exon 24 and Exon31?

Yes. This is shown in the paper and discussed in the text as follows: "If Exon 31 is replaced by Exon 32 (E24+/E30b-/E31-/E32+), Lphn3 predominantly couples to $G_{\alpha 12/13}$. Inclusion of Exon 30b, or exclusion of Exon 24 also shifts Lphn3 G_{α} coupling from $G_{\alpha s}$ to $G_{\alpha 12/13}$ ". In other words, $G_{\alpha s}$ coupling needs the inclusion of both Exon24 and Exon31.

Figure 2

(7) -This method figure does not describe a significant analytical/technical discovery or a main finding but confirms that the deletions are well done. It can be transferred to supplementary information.

Figure 2 is more than a method figure, because panel e-h presents a key finding that Lphn3 alternative splicing is important for neuronal network activity. For panel a-d, we extensively validated our method of acute CRISPR-mediated manipulations to delete all Lphn3 (Lphn3 KO), and importantly, to switch alternative splicing from Exon31 to Exon32 (Exon31 KO). To the best of our knowledge the latter goal has not been achieved before, with both high efficiency and high specificity in primary cultured neurons. Thus, we think it is crucial to benchmark how well it was done, so that others could utilize this method to study the function of alternative splicing of other genes in neurons.

(8) -Fig.2c. Why is there an increase of E32 mRNA levels in the E31 KO? Could this be a compensatory mechanism?

It is possible to view the increase in E32 mRNA levels as compensatory as the reviewer suggests. Because E31 and E32 are mutually exclusive during splicing (as described in main text; see Extended Data Figure 1a), if one deletes E31, E32 has to go up. Pre-mRNAs are processed in spliceosomes by joining the 5' splice donor proximal to the upstream exon to the 3' splice acceptor site proximal to the downstream exon. So when the constitutive exon27 cannot be spliced into exon31 (because our gRNA destroys the splice acceptor site of exon31, Figure 2a), the spliceosome will guide exon27 to find the alternative downstream acceptor site that corresponds to exon32. Because the total Lphn3 protein/mRNA level does not change with Exon31 KO (Figure 2b, 2d), the same amount of total Lphn3 would naturally contain more Exon32 than in the control group.

(9) -Fig.2e-h. The image in panel e needs to be clarified. What is the true Calcium transient signal when it fills the entire soma or the "dot" pattern? A better image or a better explanation may need it here.

Following the reviewer's suggestion, we now present a higher resolution and higher magnification image in the new Figure 2. Green dots mark the nuclei of cells that were infected with the lentiviruses expressing the gRNA together with nuclear EGFP. The red signal of the soma corresponds to the jRGECO1a peak signal during a recording period. We have added the above explanation to the Figure legend.

(10) -Why do Latrophilin-3 KO and E31 KO show the same firing rate phenotype? Does it mean that the ability of E31 to recruit postsynaptic proteins is sufficient to drive the phenotype or that the method does not have the resolution to distinguish between the complete lack of Latrophilin-3 signalling and the absence of one splicing isoform?

The neuronal firing rate as revealed by calcium imaging is the convergent result of many cellular signaling processes, including synaptic events. The Exon31 KO phenocopies the effect of the total Lphn3 KO, which is the key observation in all three assays (calcium imaging, synapse density, connectivity in rabies tracing) – none of these assays reveal a function for the Exon32 variant of Lphn3. This observation indicates that Exon31 contains essential sequences for the function of Lphn3. In Figure 4 and the new Extended Data Figure 9, we showed that part of this essential sequence is the PDZ binding motif (PBM) at the very end of Exon 31 that is lacking from Exon 32. The loss of the PBM causes: (1)

loss of excitatory synapses (New Extended Data Figure 9f, 9g), (2) loss of Lphn3 enrichment on the post-synaptic scaffold condensates (New Figure 4d, 4e), (3) and loss of tenm2/Flrt3-induced clustering of the post-synaptic scaffold condensates (New Figure 4f, 4g).

Figure 3

(11) -Fig.3a. Why do the authors use vGluT1 and Homer to label the synapses? Bassoon is often more used with Homer since they are both at the synaptic cleft. Is there any reason why PSD95 is not used? This is important considering that in Figure 4, the authors describe the possibility of interaction of the PDZ E31 with PSD95.

We agree that the choice of correct antibodies for studies of synapses is important. We chose vGluT1 as a presynaptic marker instead of Bassoon, and Homer1 as a postsynaptic marker instead of PSD95 because of excellent previous results (PMID: 32973045). The reviewer is correct that Bassoon is a great presynaptic marker but our intent was to specifically label excitatory synapses, for which vGluT1 is the only presynaptic marker since Bassoon is also present in inhibitory synapses. As regards postsynaptic markers, Homer1 antibodies are generally superior to PSD95 antibodies for high-resolution imaging.

(12) -In Figure 3a, the overlay colocalization of vGluT1 and Homer are often outside of the dendrite “floating” puncta. What are the criteria for colocalization? Do they quantify as positive even if it is not on the top of the dendrite?

Again this is an excellent technical question. Excitatory synapses, especially in mature mushroom spines, are localized ~0.5-1 μ m away from the dendrite due to the long neck of the spine (PMID: 34168338, PMID: 34086051). Our confocal images have interpixel unit of 0.20714 μ m/pixel. As a result, in our analyses we include vGluT1/Homer signals within 5 pixels away from dendrite for analysis, even if the signal is not directly on top of the dendrite. This is now described in detail in our Methods section.

(13) -Rabies show connectivity between cells, which in this case, could indicate a decrease in synaptic connectivity due to a reduction in density or strength. To confirm the loss of synapses in vivo, a better approach will be to quantify the density of synapses in vivo.

We agree with the reviewer that pseudorabies virus tracing assesses the synaptic connectivity between neurons, which is how we interpreted our result in the main text (“We used monosynaptic retrograde tracing by pseudo-typed rabies virus to map the connectivity of genetically manipulated starter neurons in the hippocampal CA1 region...”, “...These data demonstrate that the Exon 31-containing Lphn3 isoform coupled to G α S is essential for Lphn3-mediated synaptic connectivity”). We also agree that additional synapse density measurements in vivo could be performed. However, given the strong synaptic connectivity phenotype we observed, and given the previous description from our lab of an in vivo synapse loss after genetic deletion of Lphn3 that correlates with a

decrease in synaptic connectivity as assessed by pseudorabies virus tracing (PMID: 30792275), we felt that additional in vivo synapse density measurements were redundant.

(14) Figure 3d. Why are there more connections from the dentate gyrus in the E31 KO compared to Lphn3 KO and control?

We apologize for the confusion but there actually are no major connections from dentate gyrus to CA1 region neurons. We provided sub-optimal representative images in the original version of this paper that misled the reviewer to the interpretation of dentate gyrus connections. The weak green fluorescence in the dentate gyrus is derived from the Cas9 mice used in the rabies tracing experiments that constitutively express Cas9 and weakly express GFP in some brain regions (JAX #024858). We thank the reviewer for making us aware of this problem and have now used more representative images in the revised paper (new Figure 3d, 3e). Moreover, we document this background GFP effect of Cas9 mice in the new Extended Data Figure 5e, in which we show that C57BL/6 mice lack this weak GFP signal in the dentate gyrus, whereas control Cas9 mice contain it. Nonetheless, the background GFP signal does not affect our interpretation of presynaptic input from CA3 and lateral Entorhinal cortex region.

Figure 4

(15) -Fig. 4b. It is difficult to distinguish the presence of Lphn3-Exon32 in the supernatant.

We agree with the reviewer that the Lphn3 Exon32 C-terminal domain (CTD, 42.4 kDa) migrates similarly to Shank3 (31.7 kDa), which is more clearly seen in the new Extended Data Figure 6j. We have clarified this issue in Figure 4b legend as follows: “the Lphn3 Exon32 CTD fragments migrates similarly to Shank3. See New Extended Data Figure 6j. Thus we quantify the Lphn3 pellet % using the NTD which has the same sequence for all three Lphn3 constructs.”

(16) -Fig. 4h. The authors conclude in line 216: “These data suggest that alternative splicing of Lphn3 at the C-terminus determines its ability to recruit postsynaptic scaffold proteins, presumably via interactions between the PDZ-binding motif in Exon 31 of Lphn3 and the PDZ domain in Shank3 or/and PSD95”. However, in Figure 4d, Shank3 and PSD95 show a similar pattern in the present E31 or E32. It seems that this E31 isoform is part of the postsynaptic density complex rather than involved in any type of recruitment. If the authors want to go further in their statement, they need to add more controls to these experiments. For example, “a non-binding” partner to Lphn3, also Lphn3 32, that seems to be doing nothing (Figure 4d).

The reviewer is correct that Shank3, PSD95 and Homer3 form phase-separated droplets independent of Lphn3-E31, as originally shown in classical studies from Mingjie Zhang’s lab (PMID: 30078712). Our data reveal that Lphn3-E31 serves to coat the surface of postsynaptic scaffold condensates but doesn’t promote such condensates. Importantly, binding of the presynaptic Lphn3 ligands Tenm2 and Flrt3 to Lphn3-E31 coated postsynaptic condensates clusters these into larger assemblies. Lphn3-E32 is unable to do this. As a further control, we also now show that the action of Lphn3-E31 in phase-

separated condensates requires its C-terminal PBM that binds to Shanks. Thus, our data demonstrate that Lphn3-E31 but not Lphn3-E32 recruits postsynaptic scaffolds that are present in phase-separated condensates and that this action of Lphn3 is amplified by binding of presynaptic Lphn3 ligands. Our finding suggests a mechanism of how postsynaptic scaffold networks are assembled, at the synapses.

Figure 5

(17) This is the weaker figure of the paper, and the link with activity is in the title of this manuscript. The authors use previous data from Greenberg's lab where the activity-dependent genes are screened using depolarization-induced stimulation in cultured neurons or Kainate-induced hippocampus activation in vivo. Both approaches are rather extremes and not physiological. If they want to show an activity-dependency of splicing isoforms, they must corroborate the in-silico analysis with at least some more physiological approaches in vivo (e.g. DREADDs). Also, if they want to link this to a synaptic change, they need to show that the change is prevented in the KO experiments.

We agree with the reviewer that additional data on the activity-dependence of Lphn3 alternative splicing would be helpful and we now provide such data. To corroborate the observations in KCl and kainate-activated neurons of the original manuscript, we have now added the following data:

- (1) We analyzed another public RNAseq dataset (PMID: 29898393) in which hippocampal cultures at DIV14 were treated with 100 μ M picrotoxin for 15 hr to elevate neuronal activity (picrotoxin is generally thought to be a mild method to increase neuron activity). Compared with control neurons treated with DMSO (solvent of picrotoxin), the activated neurons exhibited significantly higher Exon31 inclusion and less Exon32 inclusion.
- (2) We analyzed a single cell RNAseq dataset (PMID: 33177708) from the medial prefrontal cortex (mPFC) region of mice, which had not subjected to any drug/virus treatment. We classified neurons as activated and non-activated by the expression level of immediate early genes (IEG), including *fos*, *arc*, *Junb* (see Methods, also new extended Data Figure 10d). The IEG high group (a proxy to "activated" neurons) express significantly less Exon32 and more Exon31 than in IEG low/med group (non-activated neurons). This dataset again shows activated neurons have higher inclusion of Exon31, less inclusion of Exon32. Since this dataset captures the natively activated neurons in vivo, we think it provides strong support for our conclusion that neuronal activity promotes the splicing of the more synaptogenic Lphn3 variant (Exon31-containing) of Lphn3.

We have added the above result to the Figure 5b and 5d-5f and main text accordingly.

Minor

(18) -Why do the authors use Retina databases since the manuscript is focused on the hippocampus?

The only publicly available dataset that examines long-read mRNA sequences (Pacbio) for Lphn3 was performed using retina and cortex tissues. We used this dataset to assess

the pattern of Lphn3 (also Lphn1 and Lphn2) alternative splicing, which is more intuitive for the broader audience to appreciate which exons are alternatively spliced, and whether an exon is either facultatively deleted (e.g., Exon24 of Lphn3), or whether two different exons are used in a mutually exclusive manner (e.g., Exon31 and Exon32 of Lphn3). We did not use the long-read mRNAs sequencing data for other interpretations. The other datasets we analyzed are all based on next-generation sequencing (NGS) that generates short-read fragments which are more informative/quantitative for analysis of expression levels but do not provide insight into the overall splicing patterns of a gene that long-read sequences offer.

(19) - Would it be possible to confirm the cell-type specific expression pattern of the Lphn3-Exon 31 and Lphn3-Exon 32?

We have performed further analyses to confirm the cell type-specific expression of Exon31 vs Exon32 splice variants of Lphn3:

- (1) We analyzed a highly cited public single cell RNAseq dataset from Allen Institute (PMID: 30382198), which used 23822 cells from primary visual cortex and the anterior lateral motor cortex. By sorting into different cell types, this dataset also shows strong preference of Exon31 for inhibitory neurons and Exon32 for excitatory neurons (New extended Data Figure 1d). This dataset has expression count available for the public to browse, without the need of bioinformatic expertise (<https://mouse-v1-alm-rna-hub.s3-us-west-2.amazonaws.com/MouseV1ALMRNAHub.html>).
- (2) We analyzed the mPFC dataset as mentioned above (PMID: 33177708), by sorting neurons into excitatory and inhibitory groups (New Extended Data Figure 10b, 10c). Again, Exon31 prefers to express in inhibitory groups, and Exon32 prefers to express in excitatory groups (New Extended Data Figure 1e).

We hope that with the additions and changes made, the reviewers will continue to share our excitement about our findings and endorse publication of our paper. Again, we thank the reviewers for their insightful and detailed evaluation of our study.

Reviewer Reports on the First Revision:

Referees' comments:

Referee #1 (Remarks to the Author):

The authors performed a thorough revision, incorporating both the suggested experiments and going one step beyond by providing a cause-effect link between Lphn3/Shank binding and the targeting of postsynaptic scaffold proteins to the interface of condensates. Regarding the new FRAP analysis (Ext. Data Fig. 8), there is an apparent immobile fraction for Homer3 (i.e., ~90%) irrespectively of other components, strongly speaking against its diffusibility. Interestingly, it seems that PSD95 transitions from a larger mobile fraction when Lphn3 is present to nearly becoming immobile once presynaptic ligands Tenm2 and Flrt3 are in the mixture. Both of these observations should be discussed, especially as this indicates that surface recruitment of Lphn3-E31 directly modulates the viscoelastic properties of these condensates. Overall, this study comprehensively documents that activity-dependent switching of Lphn3 isoforms leads to differential functional outcomes by prohibiting or stimulating phase separation. Given the novelty of the message and the clarity of experiments, this study promises to stimulate both fields of synaptic biology and functional signalling mediated by phase separation; therefore, I strongly endorse its publication in Nature.

Signed review,
Drago Milovanovic

Referee #3 (Remarks to the Author):

The authors provide a relatively convincing argument for why this manuscript does not show a next-intracellular protein interacting with Latrophilin-3 for synapse formation in their rebuttal. For a general audience, in my opinion, the way the Introduction is written does not reflect this step forward (e.g. "...but the molecular mechanisms by which Lphn3 induces synapse formation remain elusive., ...However, it is unknown which Ga protein physiologically mediates Lphn3- dependent synapse formation..."). The manuscript would benefit from strengthening the message as it is written in the revision. The authors have addressed most of the comments by doing new experiments or clarifying my concerns.

Minor

Point 11 of my previous revision (11) -Fig.3a. Why do the authors use VGluT1 and Homer to label the synapses? Bassoon is often more used with Homer since they are both at the synaptic cleft. Is there any reason why PSD95 is not used? This is important considering that in Figure 4, the authors describe the possibility of interaction of the PDZ E31 with PSD95.

This point is not addressed accurately. I agree that Homer antibodies are superior in quality compared to PSD95. However, although Bassoon is also in inhibitory synapses, Homer is not. Therefore, the "synapses" can be localized using Bassoon and Homer and can be excitatory. I still do not understand why they did not use Bassoon and Homer, but I do not consider this a major point.

Author Rebuttals to First Revision:

Referees' comments:

Referee #1 (Remarks to the Author):

The authors performed a thorough revision, incorporating both the suggested experiments and going one step beyond by providing a cause-effect link between Lphn3/Shank binding and the targeting of postsynaptic scaffold proteins to the interface of condensates.

Regarding the new FRAP analysis (Ext. Data Fig. 8), there is an apparent immobile fraction for Homer3 (i.e., ~90%) irrespectively of other components, strongly speaking against its diffusibility. Interestingly, it seems that PSD95 transitions from a larger mobile fraction when Lphn3 is present to nearly becoming immobile once presynaptic ligands Tenm2 and Flrt3 are in the mixture. Both of these observations should be discussed, especially as this indicates that surface recruitment of Lphn3-E31 directly modulates the viscoelastic properties of these condensates.

Overall, this study comprehensively documents that activity-dependent switching of Lphn3 isoforms leads to differential functional outcomes by prohibiting or stimulating phase separation. Given the novelty of the message and the clarity of experiments, this study promises to stimulate both fields of synaptic biology and functional signalling mediated by phase separation; therefore, I strongly endorse its publication in Nature.

Signed review,
Drago Milovanovic

Response to Referee #1:

We greatly appreciate Dr. Milovanovic's enthusiasm and insightful comments to our manuscript. We agree that two parameters are measured in the FRAP experiment: recovery kinetics (which reflects how fast the diffusive molecules exchange with those in phase condensate) and recovery percentage (which reflects the fraction of mobile molecules in the phase separated condensate). Fast recovery kinetics and high recovery percentages indicate highly dynamic condensates.

Practically, the recovery percentage is also affected by the duration of the bleaching. Each recording in our experiment employed ~10 seconds bleaching, which was manually controlled. The small variations in bleaching duration in our recordings preclude unequivocal deduction of the mobile fraction from the recovery percentage. Nonetheless, we did notice that the recovery percentage of Homer3 (3.6-11.4%) is consistently lower than that of other molecules (23-54%), which we think is at least partially due to the ~10 seconds bleaching duration that bleached the 405 nm dye (used for label Homer3) more than the other dyes of longer wavelengths (488, 546, 647 nm for labeling Lphn3, PSD95 and shank3, respectively), as suggested by previous FRAP recordings using the same 4 scaffold proteins (PMID: 30078712, Fig1D) that showed much higher recovery percentages (at least ~40%) for Homer3 labeled with the 488 nm dye. Note that these referenced data showed recovery kinetics (PSD95 faster than Homer3 and shank3) that are in agreement with our measurements (Ext. Data Fig. 8j,8k).

As regards the Tenm2/Flrt3 effect on PSD95, we also did not draw any conclusion from the recovery percentage data for the reasons explained above. In addition, if there were real changes induced by the ligands in the dynamics of scaffold proteins, we would expect to see a consistent effect of ligands on the recovery percentage of both PSD95 (a potential interactor of Lphn3) and Shank3 (a known interactor of Lphn3), which is inconsistent with the data in Ext. Data. Fig 8i.

Our reported recovery kinetics not only reproduced previous measurements (PMID: 30078712, Fig1D) but also are intrinsically consistent (Ext. Data Fig 8k) in the sense that all scaffold proteins show a similar trend.

Therefore, we propose to only interpret the recovery kinetics of the FRAP experiment in the main text and to specify the reasons/limitations in the methods section to explain why we did not interpret the recovery percentages. The text is given below with the edited sentences underlined:

Main text: “fluorescence recovery after photobleaching (FRAP) experiments show that Lphn3 exhibits a faster recovery kinetics than most scaffold proteins, suggesting that Lphn3-Exon31 forms a fluidic shell on the surface of the scaffold protein condensates. The coating by Lphn3-Exon31 of the condensates, regardless of the Tenm2/Flrt3-induced clustering, does not significantly perturb the recovery kinetics of scaffold proteins”.

Methods: “Fluorescence Recovery After Photobleaching (FRAP) experiments. After phase separation is completed, strong excitation laser intensities were used to bleach all channels of a small area for approximately 10 seconds, after which the fluorescence of the photobleached spot was recorded for 6 min. Recovery traces were fitted with the exponential equation $y = ae^{-bx} + c$ to extrapolate the $t_{1/2} = \frac{\ln 2}{b}$. Only the FRAP recovery kinetics were interpreted because the recovery percentage is highly sensitive to the duration of photobleaching which was not precisely controlled in this experiment.”

Referee #3 (Remarks to the Author):

The authors provide a relatively convincing argument for why this manuscript does not show a next-intracellular protein interacting with Latrophilin-3 for synapse formation in their rebuttal. For a general audience, in my opinion, the way the Introduction is written does not reflect this step forward (e.g. “...but the molecular mechanisms by which Lphn3 induces synapse formation remain elusive..., ...However, it is unknown which G α protein physiologically mediates Lphn3- dependent synapse formation...”). The manuscript would benefit from strengthening the message as it is written in the revision. The authors have addressed most of the comments by doing new experiments or clarifying my concerns.

We greatly appreciate the reviewer’s careful assessment of our manuscript, and the suggestion of clarifying the significance statements in the introduction. We have revised our introduction as shown below, with the changed text underlined, to better describe the significance of our study:

“Multiple synaptic adhesion molecules are known to localize pre- or postsynaptically, but no coherent concept exists on how synaptic adhesion molecules assemble synapses. Among various synaptic adhesion molecules, Latrophilin-3 (Lphn3; gene symbol Adgrl3) plays a prominent role in establishing Schaffer-collateral synapses formed by CA3-region axons on CA1-region pyramidal neurons in the hippocampus⁴. Lphn3 belongs to a family of postsynaptic adhesion-GPCRs (aGPCRs) that bind to the teneurin and FLRT presynaptic adhesion molecules^{5,6,7}. Lphn3’s function in synapse formation is known to require both its extracellular FLRT/tenurin-binding sequences and its intracellular regions, including its G α protein-binding sequences^{4,8}, but it is unclear how Lphn3 functions in synapse formation as a key synaptic adhesion molecule that is also a GPCR. In cell-signaling assays, multiple G α proteins were reported to couple to Lphn3^{9,10,11,12} as confirmed by cryo-EM structures^{13,14}. However, it is unknown which G α protein physiologically mediates Lphn3-dependent synapse assembly, whether G α protein signaling on its own constitutes the core mechanism of synapse formation, and how presynaptic ligand-binding to postsynaptic Lphn3 induces synapse formation. Moreover, synaptic scaffold proteins have well-established functions in synapse organization via formation of phase-separated condensates²⁸, but their relation to trans-synaptic adhesion complexes remains poorly understood.”

Minor

Point 11 of my previous revision (11) -Fig.3a. Why do the authors use vGluT1 and Homer to label the synapses? Bassoon is often more used with Homer since they are both at the synaptic cleft. Is there any reason why PSD95 is not used? This is important considering that in Figure 4, the authors describe the possibility of interaction of the PDZ E31 with PSD95.

This point is not addressed accurately. I agree that Homer antibodies are superior in quality compared to PSD95. However, although Bassoon is also in inhibitory synapses, Homer is not. Therefore, the “synapses” can be localized using Bassoon and Homer and can be excitatory. I still do not understand why they did not use Bassoon and Homer, but I do not consider this a major point.

We agree with the reviewer that the localization of PSD95 might be very informative, particularly if a super-resolution approach had been pursued. However, we use Homer only as a marker without a claim for precise nanometer localizations. Since Homer and PSD95 are both known to be part of the overall PSD scaffold complex, we feel that the superior quality of Homer antibodies (PMID: 33083769, Fig2) justifies their choice for immunocytochemistry. Moreover, using PSD95 for synapse labeling in culture neurons would not provide insight into a possible interaction between the PDZ binding motif (PBM) of Lphn3 E31 and PSD95 because of the limited resolution of microscopy – even super-resolution microscopy would not resolve this question. Indeed it is possible that besides Shanks, PSD95 and its paralogs interact with Lphn3-Exon31 but this could only be tested by more sophisticated methods, such as FLIM of tagged proteins. We agree this would be an interesting future direction to pursue.

We also agree that bassoon antibodies combined with Homer antibodies can label excitatory synapses by counting only puncta that are positive for both. On the other hand, vGluT1 is also an excellent pre-synaptic marker for excitatory synapses, and we had been successfully using it in many of our previous studies (PMID: 32973045 Fig2, PMID: 31262725 FigS3, PMID: 34913963 FigS3, PMID: 33646123 Fig3, PMID: 30792275 Fig2). As a result, we think there is no reason not to use vGluT1 antibodies, especially since they have the added benefit of being specific for excitatory synapses. In sum, we feel that both bassoon/Homer colocalization and vGluT1/Homer colocalization are great ways to label excitatory synapses but there is no inherent advantage of the bassoon/Homer or bassoon/PSD95 combination of the vGluT1/Homer combination.